# Regulating protein corona on nanovesicles by glycosylated polyhydroxy polymer modification for efficient drug delivery

Yunqiu Miao[1,2], Lijun Li[1,3], Ying Wang[1,3], Jiangyue Wang[1], Yihan Zhou[1], Linmiao Guo[1], Yanqi Zhao[1,3], Di Nie[1], Yang Zhang[2], Xinxin Zhang ●[1,3,4] ✉ & Yong Gan ●[1,3,5] ✉

The dynamic protein corona formed on nanocarriers has been revealed to strongly affect their in vivo behaviors. Precisely manipulating the formation of protein corona on nanocarriers may provide an alternative impetus for specific drug delivery. Herein, we explore the role of glycosylated polyhydroxy polymer-modified nanovesicles (CP-LVs) with different amino/hydroxyl ratios in protein corona formation and evolution. CP-LVs with an amino/hydroxyl ratio of approximately 0.4 (CP₁-LVs) are found to efficiently suppress immunoglobulin adsorption in blood and livers, resulting in prolonged circulation. Moreover, CP₁-LVs adsorb abundant tumor distinctive proteins, such as CD44 and osteopontin in tumor interstitial fluids, mediating selective tumor cell internalization. The proteins corona transformation specific to the environment appears to be affected by the electrostatic interaction between CP-LVs and proteins with diverse isoelectric points. Benefiting from surface modification-mediated protein corona regulation, paclitaxel-loaded CP₁-LVs demonstrate superior antitumor efficacy to PEGylated liposomes. Our work offers a perspective on rational surface-design of nanocarriers to modulate the protein corona formation for efficient drug delivery.

When nanocarriers are exposed to the human body, they inevitably adsorb hundreds of proteins in bodily fluids[1]. It has been reported that many plasma proteins (i.e., serum albumin, immunoglobulin, adhesion mediators, and complement components) are adsorbed on the nanocarrier surface, forming a protein corona, and confer a changed biological identity for nanocarriers that determines their biological behaviors[2,3]. Moreover, the protein corona is not static, dynamically exchanging as the nanocarriers are transported in different physiological environments over time[4–6]. The dynamic protein corona induces recognition and clearance of nanocarriers by the immune system, and hinders uptake in desired target cell populations[7], presenting a major obstacle to the bench-to-bedside translation of nanocarriers. Although much effort has been devoted to determine the factors affecting the formation of protein corona, a longstanding question regarding how to modulate the interaction modes of functional proteins on the nanosurface remains unresolved.

Recent studies have shown that protein adsorption is significantly influenced by the surface modification of nanocarriers[8]. Polyhydroxy polymers, such as polyethylene glycol (PEG) and block copolymers, are the most widely applied biomaterials modified on liposomes, micelles, and other nanocarriers[9,10]. Polymer modification could minimize protein adsorption to evade recognition by phagocytes of the innate

[1]State Key Laboratory of Drug Research, Shanghai Institute of Materia Medica, Chinese Academy of Sciences, Shanghai 201203, China. [2]Shanghai Tenth People's Hospital, School of Medicine, Tongji University, Shanghai 200072, China. [3]University of Chinese Academy of Sciences, Beijing 100049, China. [4]Shandong Laboratory of Yantai Drug Discovery, Bohai Rim Advanced Research Institute for Drug Discovery, Yantai, Shandong 264117, China. [5]NMPA Key Laboratory for Quality Research and Evaluation of Pharmaceutical Excipients, National Institutes for Food and Drug Control, Beijing 100050, China. ✉e-mail: xinxinzhang@simm.ac.cn; ygan@simm.ac.cn

immune system and prolong the blood circulation time of nanocarriers[11]. However, recent research has demonstrated that PEGylated nanocarriers easily trigger the production of anti-PEG antibodies and result in accelerated blood clearance (ABC) with repeated administration. The half-life of the second injected PEGylated liposomes at day 4 post-first injection dramatically decreased more than 90% of that of the first injected liposomes[12]. Thus far, most PEG modification strategies[13–15] cannot completely eliminate the protein corona-induced immune clearance[16]. Hence, seeking an alternative polymer modification approach remains extremely urgent to reduce the formation of protein coronas and extend blood circulation time.

Meanwhile, hydrophilic polymer modification has been reported to limit the cellular uptake of nanocarriers in target cells, significantly reducing the potential of the delivery system[17]. Inspired by nanocarrier-protein interactions influencing cellular internalization, researchers have attempted to regulate the formation of the protein corona to improve the cellular uptake of nanocarriers in tumor cells[18,19]. Chen et al. have developed a target strategy of nanoparticles functionalized with penicillamine to adsorb transferrin and interact with transferrin receptor which is overexpressed on various cancer cells[20]. Gao et al. found that precoating transferrin-modified nanoparticles with protein coronas from healthy mice significantly enhanced active targeting capacities to tumor cells[21]. Cationic polymers, such as polyethyleneimine[22] containing amino groups that interact with the negatively charged domain of the protein on the cell surface, are advantageous for passive uptake in tumor cells[23,24]. Additionally, hydroxyl and amino groups are the key components of hydrogen bonds that link protein-receptor or drug-target protein, and could endow nanocarriers with hydrophilicity to reduce nonspecific protein adsorption, which may provide an alternative approach to modify polymers to regulate the interaction of nanocarriers and proteins[25–27]. For example, amino acid modification influenced the protein corona formation of nanocarriers and subsequent biological behaviors in vivo[28,29]. Hence, seeking suitable modification of nanocarriers to selectively avoid immune-associated protein binding and promote target cell-related protein adsorption may enhance the delivery efficiency of nanocarriers through the improved blood circulation and target cell internalization.

In this study, inspired by the functional groups of amino acid modification affecting protein adsorption and biological effects, glycosylated polyhydroxy polymers (CSO-g-TCP), which could endow nanocarrier surfaces with different amino/hydroxyl ratios, are synthesized through the grafting of chitosan oligosaccharide (CSO) and PEO-PPO-PEO triblock copolymers (TCP), and are then modified on lipid nanovesicles (CP-LVs). CP-LVs are used to investigate the protein corona composition affected by functional group differences and the nanovesicle transportation influenced by protein coronas (Fig. 1). CP-LVs with an appropriate amino/hydroxyl ratio (CP$_1$-LVs) are expected to decrease the formation of dynamic protein coronas in plasma and livers, preventing immune activation and the ABC effect. Meanwhile, during the transport of nanovesicles from blood to tumors, CP$_1$-LVs could dynamically adsorb a large number of tumor-associated proteins, such as CD44 and osteopontin (OPN) with low isoelectric point which could bind to the amino groups of CP$_1$-LVs, resulting in improved selective uptake in tumor cells. Due to their strong ability to prolong blood circulation and promote tumor cell internalization, CP$_1$-LVs loaded with paclitaxel (PTX) are expected to exert superior antitumor efficacy in tumor-bearing mice. The present study emphasizes the promising application of protein corona regulation to improve delivery efficiency of nanocarriers, and may promote the rational design of functional groups modified nanocarriers.

## Results

### Characterization of CP-LVs
The amino-modified hydrophilic polymers were synthesized by grafting PEO$_{20}$-PPO$_{70}$-PEO$_{20}$ triblock copolymers to the backbone of CSO through amidation (Fig. 2a and Supplementary Fig. 1). The molar ratios of TCP and CSO in polymers were 4:1, 2:1 and 1:1, which were calculated based on proton nuclear magnetic resonance ($^1$H-NMR), and named as CSO-g-TCP$_4$, CSO-g-TCP$_2$ and CSO-g-TCP$_1$ (Supplementary Fig. 2), respectively. CSO-g-TCP and DPPC were hydrated together to obtain a series of lipid nanovesicles, possessing the structure of a spherical vesicle and an average size of approximately 100 nm, called CP$_4$-LVs, CP$_2$-LVs and CP$_1$-LVs (Fig. 2b). The ratio of amino and hydroxyl groups was an important characteristic of the obtained CP-LVs, which was determined by Boehm titration. As shown in Fig. 2c, the average molar ratios of amino/hydroxyl groups in CP$_4$-LVs, CP$_2$-LVs, and CP$_1$-LVs were 0.2, 0.3, and 0.4, respectively, consistent with the amino/hydroxyl density in CSO-g-TCP polymers.

Surface modification of amino and hydroxyl groups determines the interaction of nanovesicles and proteins[30,31]. To further investigate the spatial arrangement of functional groups, time-of-flight secondary ion mass spectrometry (Tof-SIMS) was used to scan the signals of the two functional groups (Fig. 2d). The images clearly illustrated the presence of amino and hydroxyl ions on CP-LVs at the top section (at the beginning of the scan) and the middle section (after a certain period of sputtering), respectively. CP-LVs exhibited a higher intensity of amino ions at the top section than that at the middle section, while hydroxyl ions showed a similar intensity at the top and middle sections (Fig. 2e). The depth profiles of selected ions showed that the signal intensity of amino in CP-LVs decreased to varying degrees during the scan (Fig. 2f). These results indicated that hydroxyl groups were distributed throughout the modified layers and amino groups were exposed to the outside of the PEO layers, which was consistent with the previous literature reporting that the more voluminous coronal chains preferentially segregated to the outer leaflet of the membrane for the free energy and local curvature effects[32]. Additionally, the signal intensity of amino on CP$_1$-LVs was higher than that on CP$_2$-LVs and CP$_4$-LVs, similar to the result of Boehm titration. Meanwhile, the relative intensity of the amino signal in CP$_4$-LVs, CP$_2$-LVs, and CP$_1$-LVs decreased at a similar rate (Fig. 2g), suggesting a similar distribution pattern of amino ions on CP-LVs. These results demonstrated that CP$_1$-LVs have the highest amino intensity covering the surface. In addition, the zeta potential of CP-LVs was similar and nearly neutral (Fig. 2h), in consistent with literatures reported previously[33,34]. So far, we synthesized a series of stable and biosafe nanovesicles with similar size, zeta potential, and morphology, leaving the ratio of amino and hydroxyl groups as the only variable for the subsequent study (Supplementary Figs. 3 and 4). In addition, PEGylated liposomes (PEG-Lips) with similar mass of poly(ethylene glycol) to CP-LVs, TCP-modified lipid nanovesicles (P-LVs) and conventional lipid nanovesicles (Lips) without polymer modification, which showed similar size to CP-LVs (Supplementary Fig. 5), were fabricated as control groups to explore how the amino/hydroxyl ratio of CP-LVs differentially regulates the protein corona compositions and their biological behaviors.

### The composition of protein coronas on CP-LVs influenced by surface modification
The different surface modifications may affect the composition of protein coronas on nanovesicles after entering a physiological environment[35,36]. Bicinchoninic acid (BCA) assay and sodium dodecyl sulfate-polyacrylamide gel electrophoresis (SDS-PAGE) were used to analyze the protein corona formed on CP-LVs incubated with plasma. PEG-Lips and Lips were utilized as positive and negative controls; meanwhile, P-LVs were set as a control group compared with CP-LVs to investigate the protein corona composition influenced by nanovesicle surface function groups. As shown in Fig. 3a, Lips adsorbed a large number of plasma proteins, resulting in obvious differences in nanovesicle size changes before and after plasma protein adsorption (Supplementary Fig. 6), and the adsorption of plasma proteins was significantly reduced by surface functionalization with polyhydroxy polymers. The amount of protein corona formed on PEG-Lips and

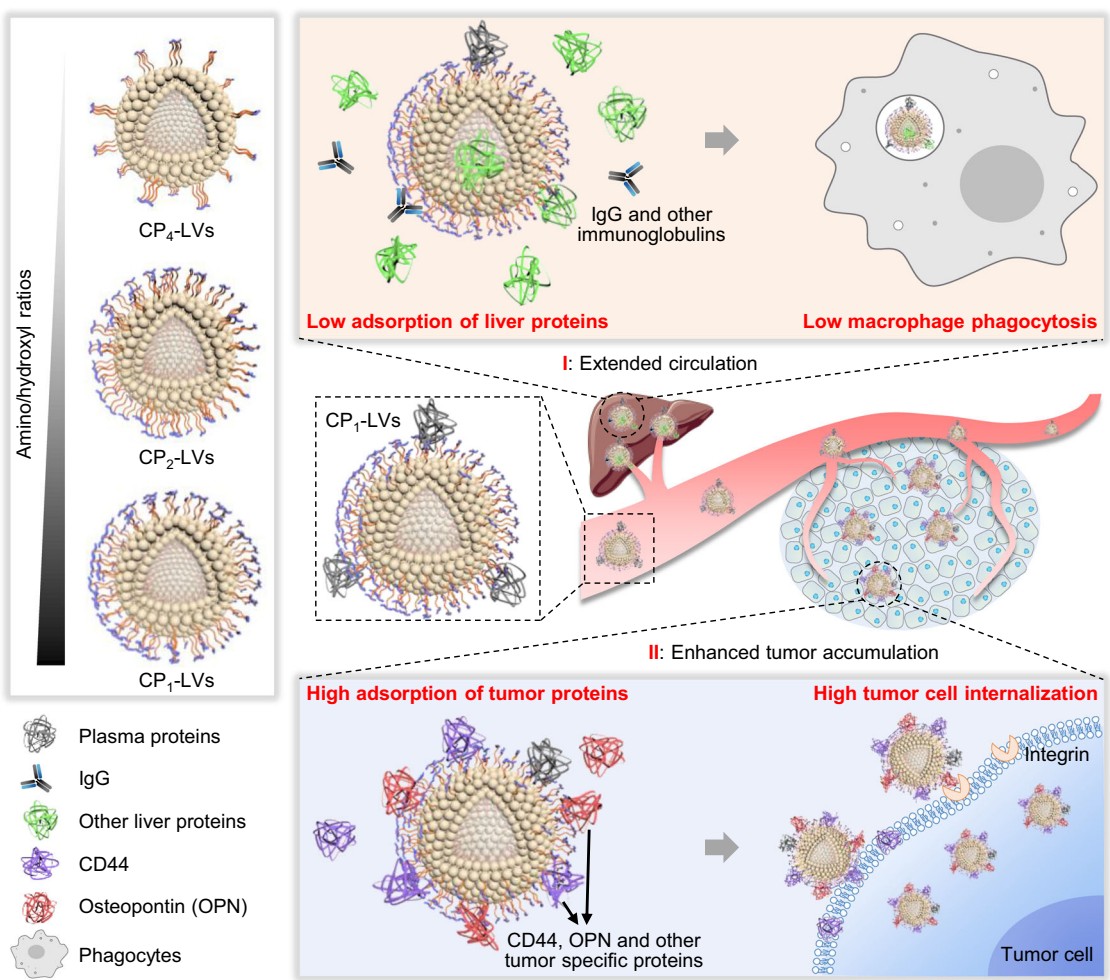

**Fig. 1 | Schematic diagram describing the design of nanovesicles modified with different ratios of amino and hydroxyl groups to explore the dynamic protein corona compositions on nanovesicle surfaces and protein corona-affected in vivo behaviors.** CP$_1$-LVs with an appropriate amino/hydroxyl ratio are expected to display the extended circulation by evading protein adsorption to avoid macrophage phagocytosis, and enhance internalization in tumor cells through binding tumor-related proteins.

P-LVs decreased by more than 50% compared with that formed on Lips. Most intriguingly, increasing the density of amino groups on the surface of CP-LVs reduced total protein adsorption. Analysis of the protein corona by SDS-PAGE was consistent with the result of the amount of proteins adsorbed on the nanovesicles (Fig. 3b). CP$_1$-LVs showed the lowest protein adsorption among all groups and weaker intensity of most bands compared with CP$_2$-LVs and CP$_4$-LVs.

To gain a more in-depth understanding of the protein corona composition, we digested adsorbed protein isolates with trypsin and analyzed the resulting peptide mixtures by liquid chromatography-tandem mass spectrometry (LC-MS/MS). Grouping the identified proteins according to their functions showed that multitudinous proteins were adsorbed on nanovesicles, including serum albumin, lipoproteins, immunoglobulins and complement proteins (Fig. 3c). Compared to Lips group, few lipoproteins including apolipoprotein A-I (ApoA1) attracted to the hydrophilic surface of PEG-Lips, P-LVs, and CP-LVs, resulting in negligible effect on macrophage phagocytosis (Supplementary Fig. 7). PEG-Lips and P-LVs mainly adsorbed albumin, fibrinogens and various immune-associated proteins. Proteins adsorbed on CP-LVs exhibited similar species to those on PEG-Lip but differed in content. Furthermore, we chose the top 20 proteins with the highest protein composition and amounts for comparison to the proteomic fingerprints (Fig. 3d). Albumin, as one of the most abundant proteins in plasma, was observed to be enriched on the nanovesicle surfaces and exhibited a reduced adsorption amount on CP-LVs compared with that on PEG-Lips

and P-LVs. Meanwhile, CP-LVs with amino modification adsorbed fewer immune response opsonins than PEG-Lips (Fig. 3d). To our knowledge, complement proteins, immunoglobulins, and macroglobulins, as immune response opsonins, are involved in a multiplicity of biological functions, including the recognition and clearance of nanocarriers. As reported, the enrichment of specific proteins (including IgG and IgM) in the protein corona induces the interaction between nanocarriers and immune cells, and activates the immune system, leading to the clearance of nanocarriers[37,38]. The expression levels of IgG and IgM were further evaluated by a Western blot assay. PEG-Lips and P-LVs adsorbed a large amount of IgG and IgM, while the adsorption of IgG and IgM on CP-LVs decreased as the amino modification increased. CP$_1$-LVs with the highest amino density exhibited the lowest adsorption of IgG and IgM compared with CP$_2$-LVs and CP$_4$-LVs (Fig. 3e, f).

The formation of protein corona is also affected by the biological environment. The liver is the largest reticuloendothelial system (RES) organ and it takes up a significant portion of the nonefficient accumulation of nanocarriers, hindering the long blood circulation and tumor delivery. The liver contains thousands of proteins differing from plasma, which may obviously influence the protein corona compositions. Compared to the plasma protein coronas, albumin and IgG were still present in the protein corona formed in the liver tissue interstitial fluid (TIF), while IgM was almost absent (Fig. 3g, h). Additionally, metabolic enzymes as abundant proteins in livers, were adsorbed to the nanovesicle surface in large quantities (Fig. 3g, i), which may cooperate with

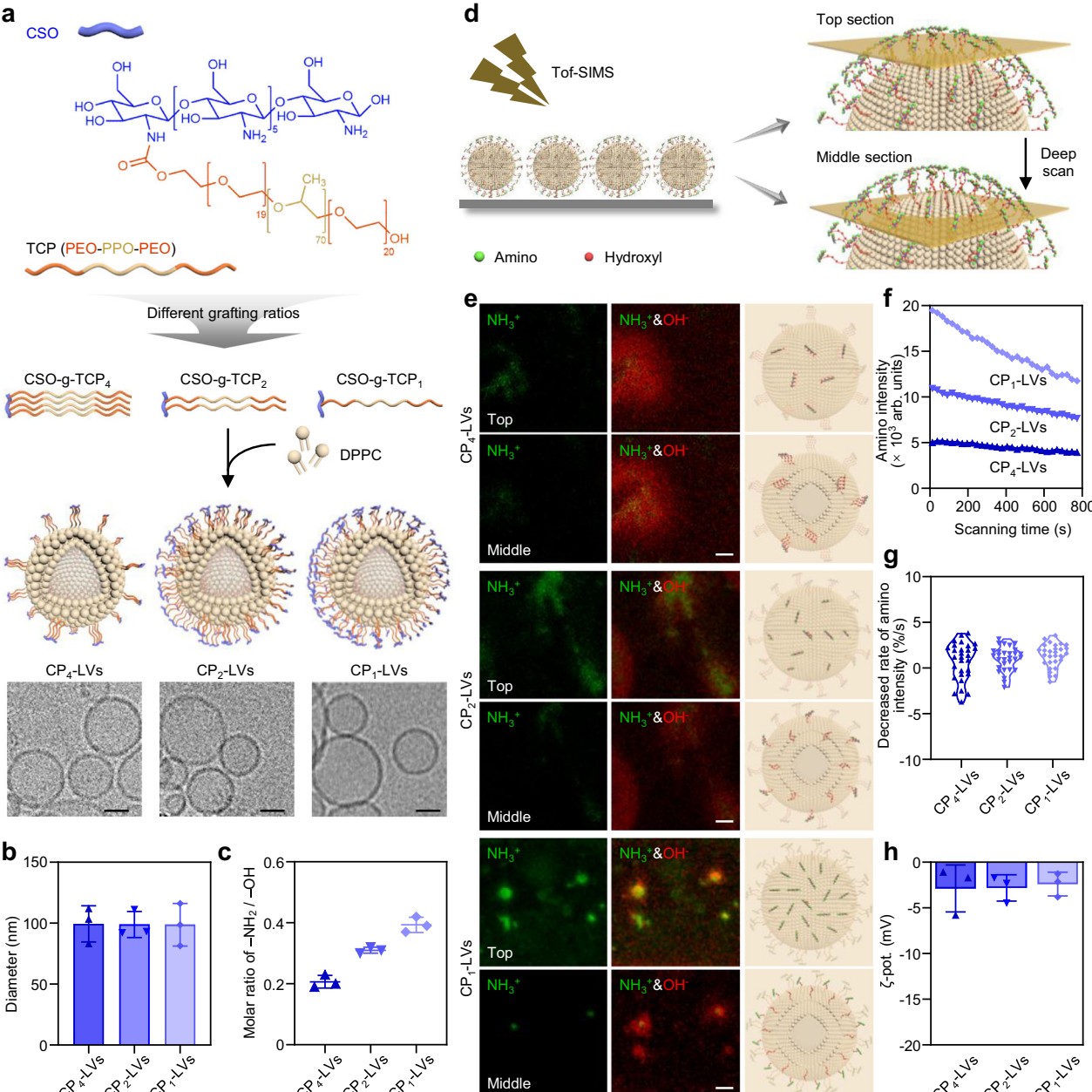

**Fig. 2 | Characterization of CP-LVs. a** Schematic diagram of the structure of CP-LVs and representative TEM images of CP₄-LVs, CP₂-LVs, and CP₁-LVs. Scale bar: 50 nm. **b** The average size of CP-LVs determined by a Malvern Zetasizer Nano ZS analyzer. The data are displayed as the mean ± SD ($n = 3$ independent samples). **c** The molar ratio of amino and hydroxyl groups in CP-LVs analyzed through Boehm titration. The data are displayed as the mean ± SD ($n = 3$ independent samples). **d** Schematic diagram describing the spatial distribution of amino and hydroxyl groups in CP-LVs determined by Tof-SIMS. **e** Representative images of amino and hydroxyl distribution at the top and middle sections observed by Tof-SIMS. CP-LVs were coated on silicon substrates and freeze-dried to form a thin film. The signals of amino and hydroxyl groups were detected by Tof-SIMS at different depths in an area of $35 \times 35\ \mu m^2$ with the pixel size of about 120 nm. Scale bar: 2 μm. **f** Real-time signal of amino during a scanning time of 800 s. **g** The decreased rate of relative intensity of amino signal during scanning. **h** The zeta potential of CP₄-LVs, CP₂-LVs, and CP₁-LVs. The data are displayed as the mean ± SD ($n = 3$ independent samples). Source data are provided as a Source Data file.

immunoglobulins with catalase activity to metabolize nanovesicles[39,40]. Similar to the plasma protein coronas, CP₁-LVs adsorbed the lowest IgG and liver metabolic enzymes among all nanovesicles, possibly avoiding the immune response activated by protein coronas.

## Dynamic protein coronas on nanovesicles in the transportation from blood to liver

The protein corona composition on the surface of nanocarriers has been reported to change with their trafficking in vivo[6,41]. To investigate the dynamic protein adsorption on nanovesicles in different physiological environments, we analyzed the composition changes of the protein corona during the transportation of nanovesicles from plasma to liver. The nanovesicles were incubated in plasma, PBS, and liver TIF in turn to clearly observe the dynamic change process of protein corona on nanovesicles[6], including adsorption, desorption, and readsorption (Fig. 4a). As shown in Fig. 4b, the amount of plasma protein on the nanovesicles was reduced after PBS washing and increased after liver TIF incubation. Intense bands of proteins of approximately 70 kDa, including albumin, in the samples with plasma and PBS incubation displayed a remarkable decrease after liver TIF

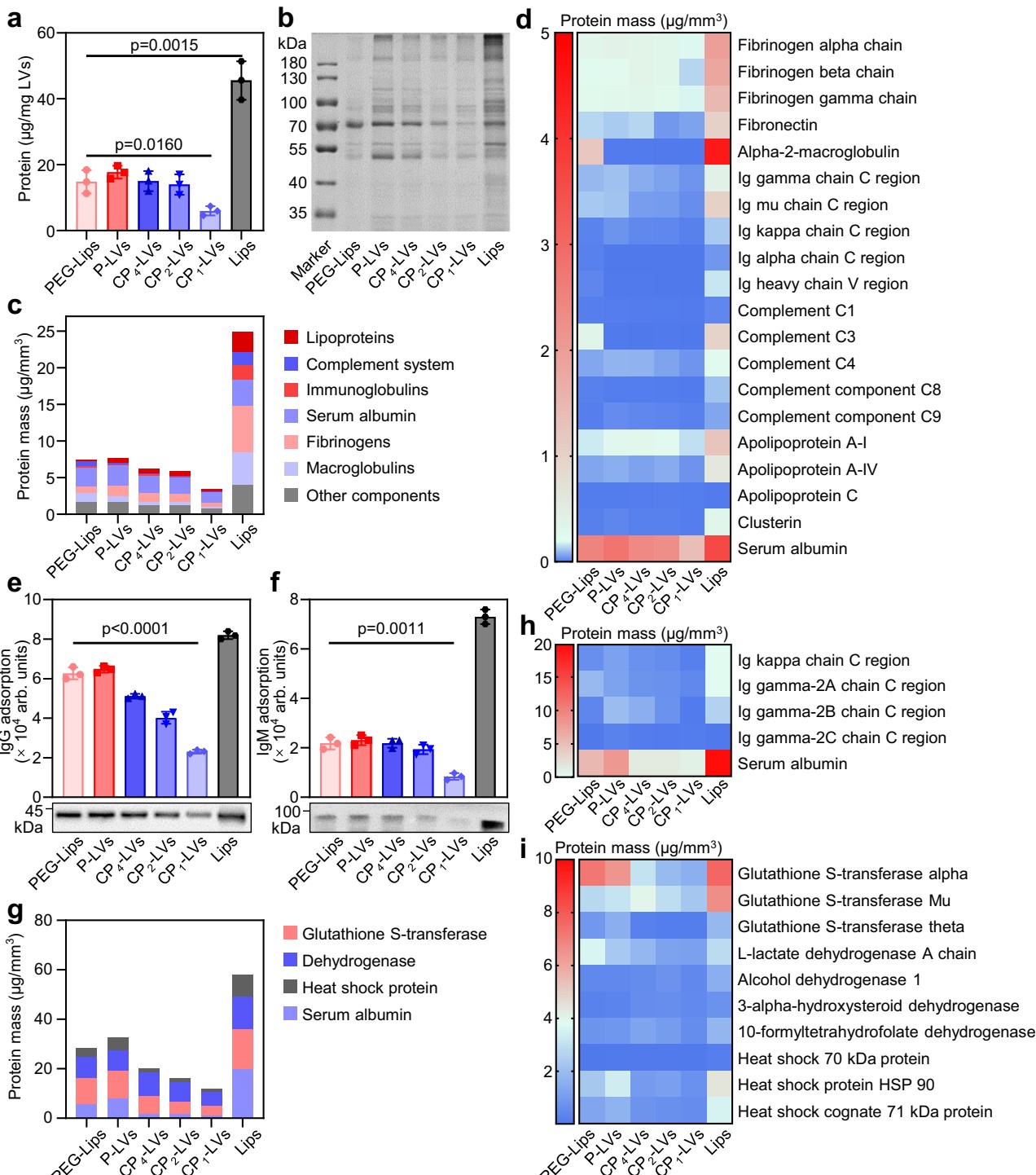

**Fig. 3 | The composition of protein coronas on CP-LVs influenced by surface modification. a** Quantification of plasma proteins adsorbed to the nanovesicle surface determined by BCA assay. The data are displayed as the mean ± SD (two-tailed Student's *t*-test. *n* = 3 independent samples). **b** Qualitative molecular composition of the adsorbed protein layer on the nanovesicles by SDS-PAGE. (*n* = 3 independent samples.) **c** Classification of plasma protein corona components identified by quantitative LC-MS/MS. **d** Heatmap of the most abundant proteins (top 20) in the protein corona of the nanovesicles with plasma incubation determined by proteomic mass spectrometry. **e** Western blot assay of IgG and **f** IgM adsorbed in the protein corona of the nanovesicles. The intensity of the IgG and IgM bands was quantified by ImageJ software. The data are displayed as the mean ± SD (two-tailed Student's *t*-test. *n* = 3 independent samples). **g** Classification of protein corona components formed in the liver TIF identified by quantitative LC-MS/MS. **h** Heatmap of immunoglobulin and albumin in the protein corona of the nanovesicles with liver TIF incubation. **i** Heatmap of the most abundant proteins (top 10) in the adsorbed metabolic enzymes and heat shock proteins on nanovesicles with liver TIF incubation. Source data are provided as a Source Data file.

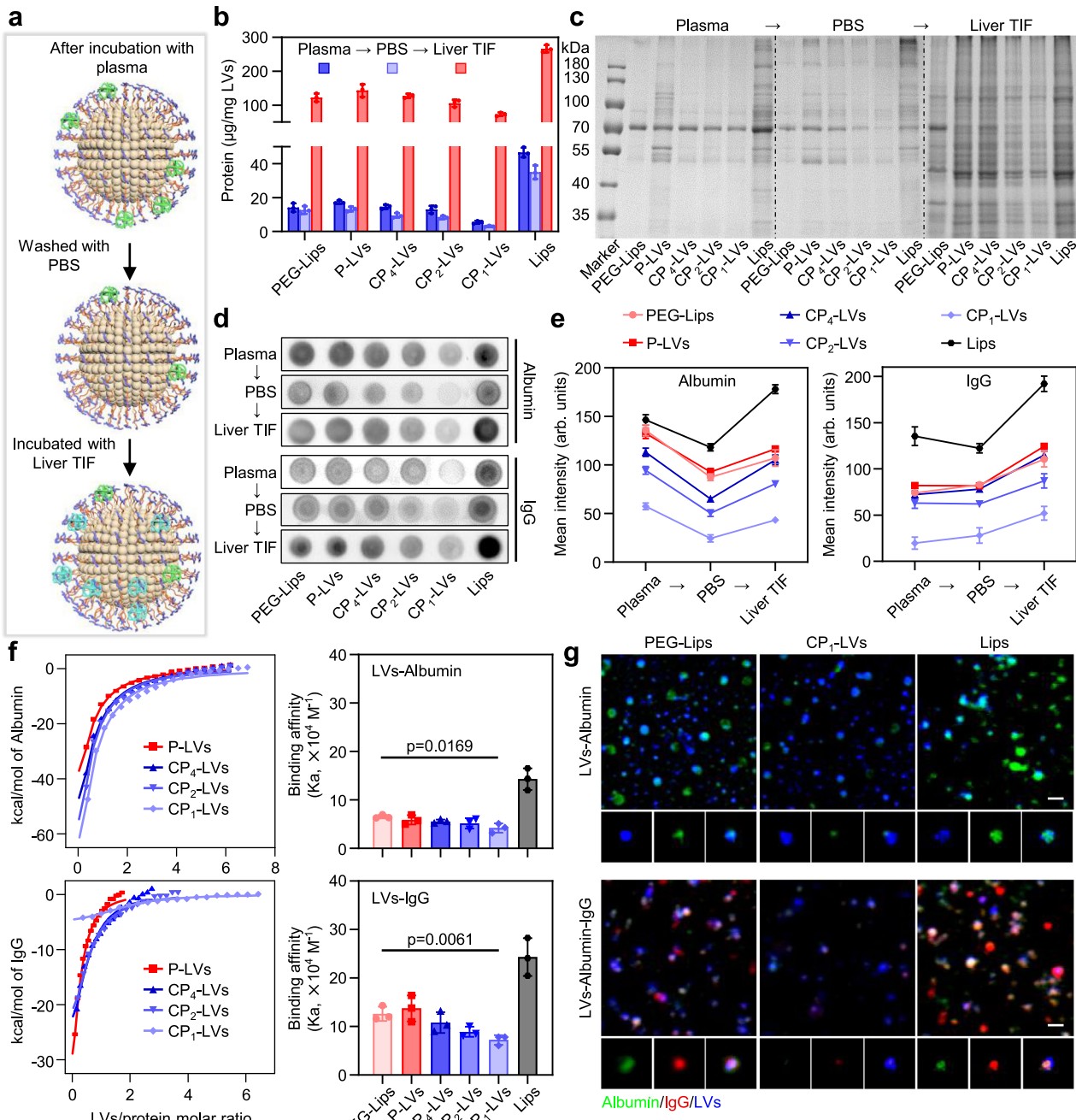

**Fig. 4 | Characterization and mechanism study of dynamic protein adsorption on nanovesicles. a** Schematic diagram of nanovesicles dynamically adsorbing proteins. Nanovesicles were incubated with plasma, PBS, and liver TIF in turn. **b** Quantification of proteins adsorbed to the nanovesicles (PEG-Lips, P-LVs, CP$_4$-LVs, CP$_2$-LVs, CP$_1$-LVs, Lips) with incubation of plasma, PBS, and liver TIF in turn. The data are displayed as the mean ± SD ($n$ = 3 independent samples). **c** Qualitative molecular composition of the adsorbed protein layer on nanovesicles by SDS-PAGE in (**b**). **d** Adsorption level of albumin and IgG on nanovesicles was determined by dot blotting assay. **e** The intensity of albumin and IgG was analyzed by ImageJ software in (**d**). The data are displayed as the mean ± SD ($n$ = 3 independent samples). **f** ITC profiles and calculated binding affinity (Ka) of LVs-albumin and LVs-IgG. The data are displayed as the mean ± SD (two-tailed Student's $t$-test. $n$ = 3 independent samples). **g** Representative confocal images of nanovesicles incubated with albumin and IgG solution. Blue: nanovesicles; Green: albumin; Red: IgG. The magnified images of single nanovesicle were shown under the confocal images. Scale bar: 100 nm. Source data are provided as a Source Data file.

incubation, while many protein bands were obviously increased on the liver TIF-incubated nanovesicles, especially proteins between 40 and 55 kDa, such as IgG (Fig. 4c). This suggested that the plasma protein corona destabilized during the transfer from plasma into liver TIF, and some plasma proteins consequently dissociated from the nanovesicles and were replaced by other proteins in liver TIF. During the incubation process, Lips adsorbed a dense layer of proteins, while PEG-Lips and P-LVs reduced protein adsorption both in plasma and liver TIF due to

the steric repulsion of PEG and PEO chains. As the ratio of amino to hydroxyl groups increased, the protein adsorption on CP-LVs obviously decreased. CP$_1$-LVs with the highest amino/hydroxyl ratio exhibited the lowest protein adsorption in plasma and liver TIF, with approximately 2.0-fold protein amount lower than that of PEG-Lips.

In accordance with the composition analysis of protein coronas in Figs. 3 and 4c, albumin and IgG, as important components that are dynamically adsorbed on nanovesicles in different physiological

environments, were chosen as the representative proteins for further study. Dot blotting was then used to analyze the dynamic adsorption of albumin and IgG on nanovesicles during incubation with plasma, PBS, and liver TIF in turn. Albumin adsorption decreased after PBS incubation and increased after liver TIF. In contrast, the adsorption level of IgG on nanovesicles in plasma showed negligible changes after PBS washing and obviously increased on nanovesicles with liver TIF incubation (Fig. 4d, e and Supplementary Fig. 8). These results indicated that albumin underwent the process of adsorption, desorption, and readsorption during incubation, while IgG was stably adsorbed in plasma and PBS and largely bound to nanovesicles in liver TIF. CP-LVs with different amino/hydroxyl ratios adsorbed less albumin and IgG than PEG-Lips and Lips. Notably, CP$_1$-LVs exhibited little albumin and IgG adsorption in plasma and liver TIF, with 2.4-fold albumin adsorption in plasma and 2.1-fold IgG binding in liver TIF lower than that of PEG-Lips. In general, CP$_1$-LVs with an amino/hydroxyl ratio of approximately 0.4 adsorbed the lowest immune-related proteins, including albumin and IgG.

To illustrate the mechanism in dynamic albumin and IgG adsorption, isothermal titration calorimetry (ITC) was used to directly measure the affinity of proteins and nanovesicles, and fluorescence-labeled proteins were incubated with nanovesicles to visualize the protein-nanovesicle interactions. Nanovesicles were titrated into the protein solutions, and the changes in the heat for every titration were measured and integrated. The adsorption of albumin and IgG on the surfaces of the nanovesicles is exothermic. The calculated binding affinity of albumin to the nanovesicles was significantly less than that of IgG, resulting in IgG replacing some albumin adsorbed on the nanovesicles, as observed through confocal laser scanning microscopy (CLSM) with stimulated emission depletion (STED) mode (Fig. 4f, g and Supplementary Fig. 9). PEG-Lips showed lower binding affinity with albumin and IgG than Lips, resulting in decreased amount of albumin and IgG adsorption on the nanovesicles. CP-LVs with amino modification decreased the binding affinity with proteins compared to P-LVs and PEG-Lips. Notably, CP$_1$-LVs showed 1.6- and 1.7-fold binding affinity with albumin and IgG lower than PEG-Lips, respectively, leading to the lowest fluorescence intensity of albumin and IgG during the incubation process.

Taken together, the protein corona formed on the nanovesicle surface is kinetically unstable and undergoes dynamic change as nanovesicles are transported from plasma to livers. CP$_1$-LVs with an amino/hydroxyl ratio of approximately 0.4 maintained little protein adsorption during transportation, especially low IgG binding, which may reduce the activation of the immune system.

### Dynamic protein adsorption in vivo and protein opsonization in macrophages

To investigate the amount of protein corona and test whether adsorbed proteins change dynamically and result in ABC effect in vivo, we recovered the intravenously injected nanovesicles from plasma and livers (Fig. 5a). The amount and variety of protein coronas formed in plasma and livers were greatly different, especially the adsorption of albumin in plasma and IgG in livers (Fig. 5b–g), suggesting the dynamic behavior of protein adsorption while nanovesicles were transported in vivo. Notably, CP$_1$-LVs with an appropriate amino/hydroxyl ratio showed the lowest adsorption of plasma and liver proteins, especially albumin and IgG (Fig. 5f, g), which was consistent with the results of protein corona analysis in vitro. More importantly, PEG-Lips adsorbed nonnegligible IgG when they were transported to livers, indicating that the PEGylated nanovesicles retain some degree of immunogenicity, which leads to the production of anti-PEG antibody and the ABC effect with repeated administration.

Then, we investigated the anti-PEG antibody level induced by CP-LVs with PEG-like structures and subsequent protein adsorption after the second injection of nanovesicles. As shown in Fig. 5h, the anti-PEG

antibody level in plasma increased obviously at 7 days postinjection with PEG-Lips or P-LVs, which is consistent with the literature report[42,43]. With the addition of amino groups, CP-LVs showed the decreased anti-PEG induction compared to P-LVs, especially, CP$_1$-LVs with the highest amino density induced the lowest anti-PEG antibody level, with 4.6 times anti-PEG antibody level lower than that induced by PEG-Lips, which may be due to the low IgG adsorption decreasing the immune response. We further analyzed the composition of the protein corona on nanovesicles with repeated administrations (Fig. 5a). PEG-Lips and P-LVs adsorbed a number of proteins after repeated administration in vivo, with similar IgG/IgM adsorption to Lips (Fig. 5i, j and Supplementary Fig. 10), which was attributed to the specific binding of anti-PEG antibodies. In contrast, CP-LVs with amino modification showed less protein adsorption than PEG-Lips and P-LVs. In particular, CP$_1$-LVs with high amino density exhibited the lowest level of IgG/IgM. In general, CP$_1$-LVs could reduce protein adsorption in plasma and livers after single and repeated administrations, especially albumin and IgG/IgM, which may be constructive to avoiding macrophage recognition and phagocytosis.

In order to study the effect of the protein corona on macrophage phagocytosis, J774 cells and Kupffer cells separately representing mononuclear macrophages in the blood and special macrophages in the liver, were used to evaluate the interaction of nanovesicles and cells. As shown in Fig. 5k, polyhydroxy polymer-modified nanovesicles pretreated with plasma, including PEG-Lips, P-LVs, and CP-LVs, showed little change in cellular uptake within J774 cells compared to those without pretreatment (Supplementary Figs. 11 and 12). In contrast, the cellular uptake of nanovesicles adsorbing liver TIF proteins increased in liver Kupffer cells compared with that of nanovesicles without protein corona (Fig. 5m), indicating that the dynamic adsorption of liver TIF proteins, especially IgG binding, promoted the clearance of nanovesicles in livers. Notably, the fluorescence intensity of all nanovesicles within Kupffer cells was generally higher than that in J774 cells, suggesting that special macrophages in livers played a more important role in nanocarrier clearance than mononuclear macrophages in blood. Then, we further studied the effect of protein coronas on macrophage phagocytosis against nanovesicles with repeated administration. Compared to nanovesicles preincubated with blank plasma, PEG-Lips and P-LVs with protein coronas from plasma preinjected with nanovesicles showed obvious enhancement in internalization within J774 cells (Fig. 5k, l). Meanwhile, the cellular uptake of PEG-Lips and P-LVs precoated with protein corona from pretreated plasma and liver TIF significantly increased within Kupffer cells (Fig. 5m). This may be owing to the specific binding of anti-PEG antibodies, activating complement system and promoting macrophage phagocytosis. Benefiting from the low adsorption of IgG and IgM, CP$_1$-LVs displayed the reduced macrophage phagocytosis regardless of single and repeated administrations, with approximately 2- and 8-fold fluorescence intensity in J774 and Kupffer cells lower than that of PEG-Lips, respectively, which may contribute to the prolonged circulation in vivo.

These results indicated that PEGylated nanovesicles adsorbed a large number of liver TIF proteins, which promoted phagocytosis of local Kupffer cells in the liver, as well as boosted phagocytosis caused by the binding of anti-PEG antibodies after repeated administration, thus limiting blood circulation. CP$_1$-LVs possessed a low binding affinity with IgG and IgM, resulting in decreased internalization in Kupffer cells and the weakened ABC effect caused by anti-PEG antibody binding. This suggested that CP$_1$-LVs may prolong blood circulation in mice with single and repeated administrations.

### Blood circulation and biodistribution of CP-LVs in vivo

We further studied the effect of nanocarrier surface protein coronas on blood circulation in vivo. The pharmacokinetics of the nanovesicles and liposomes were evaluated with a single injection of nanocarriers. As shown in Fig. 6a, compared with non-PEG modified group Lips,

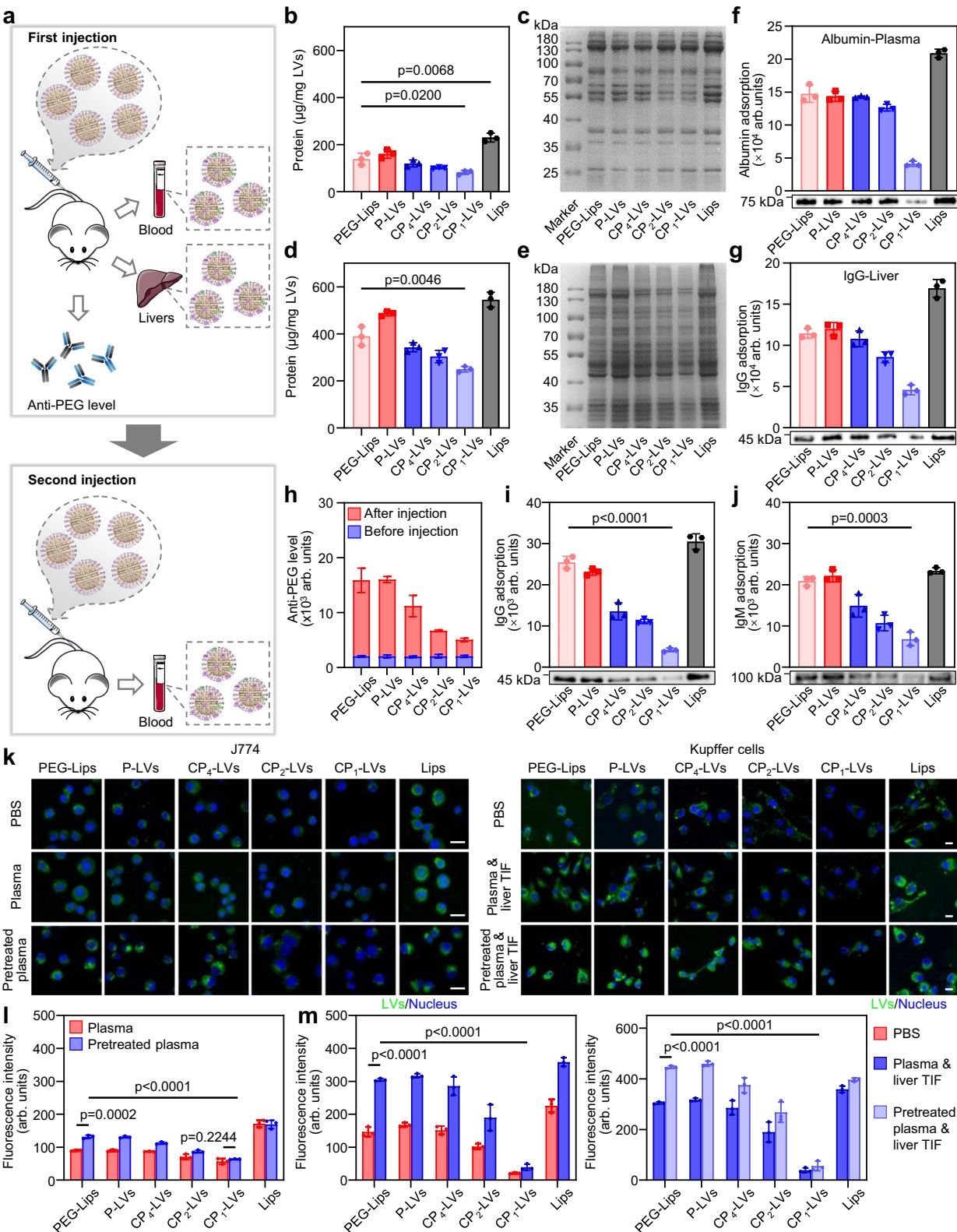

P-LVs with polyhydroxy polymer PEO-PPO-PEO modification showed a slower elimination from the circulation system-close to the positive control group PEG-Lips. As amino density increased, CP-LVs showed the enhanced retention in blood circulation compared with P-LVs. CP$_1$-LVs with the highest amino/hydroxyl ratio showed remarkably prolonged circulation with an elimination half-life of 44.6 h, which was 2.2-fold longer than that of PEG-Lips (Fig. 6b). This may be attributed to the minimum dynamic adsorption of proteins on CP$_1$-LVs as they are

transported in plasma and liver, avoiding opsonization in macrophages. PEGylated nanocarriers have been reported to induce the ABC effect with repeated administration[44,45]. We evaluated the pharmacokinetics of nanocarriers with repeated injections to investigate the effect of the protein corona on the ABC phenomenon. Obviously, PEG-Lips exhibited an ABC phenomenon caused by anti-PEG antibody binding, with an elimination half-life of 11.0 h, similar to that of Lips. In contrast, CP$_1$-LVs maintained extended blood circulation with or

**Fig. 5 | The effect of protein corona composition on macrophage uptake.**
**a** Schematic diagram describing the characterization of the protein corona in mice with single and repeated administrations. **b** Quantification of proteins adsorbed to the nanovesicles recovered from plasma. The data are displayed as the mean ± SD (two-tailed Student's *t*-test. *n* = 3 independent samples). **c** Qualitative molecular composition of the adsorbed protein layer on nanovesicles recovered from plasma. **d** Quantification of proteins adsorbed to the nanovesicles recovered from the liver. The data are displayed as the mean ± SD (two-tailed Student's *t*-test. *n* = 3 independent samples). **e** Qualitative molecular composition of the adsorbed protein layer on nanovesicles recovered from the liver. **f** Western blotting assay of albumin and **g** IgG adsorbed on nanovesicles. The data are displayed as the mean ± SD (*n* = 3 independent samples). **h** Expression level of anti-PEG antibodies in plasma collected from mice pretreated with nanovesicles. The data are displayed as the mean ± SD (*n* = 3 independent samples). **i** IgG and **j** IgM adsorption on nanovesicles

recovered from plasma in mice pretreated with nanovesicles. The data are displayed as the mean ± SD (two-tailed Student's *t*-test. *n* = 3 independent samples). **k** CLSM images of nanovesicles with different protein coronas internalized by J774 cells and Kupffer cells. Nanovesicles (PBS), nanovesicles with plasma protein corona (Plasma), and nanovesicles adsorbing proteins in plasma collected from mice preinjected with nanovesicles (Pretreated plasma) were incubated with J774 cells. Nanovesicles (PBS), nanovesicles with protein coronas formed in plasma and liver TIF (Plasma & liver TIF), and nanovesicles adsorbing proteins in plasma and liver TIF collected from mice preinjected with nanovesicles (Pretreated plasma & liver TIF) were incubated with Kupffer cells. Scale bar: 20 μm. **l** Fluorescence intensity of nanovesicles with different protein coronas within J774 cells and **m** Kupffer cells. The data are displayed as the mean ± SD (two-tailed Student's *t*-test. *n* = 3 independent samples). Source data are provided as a Source Data file.

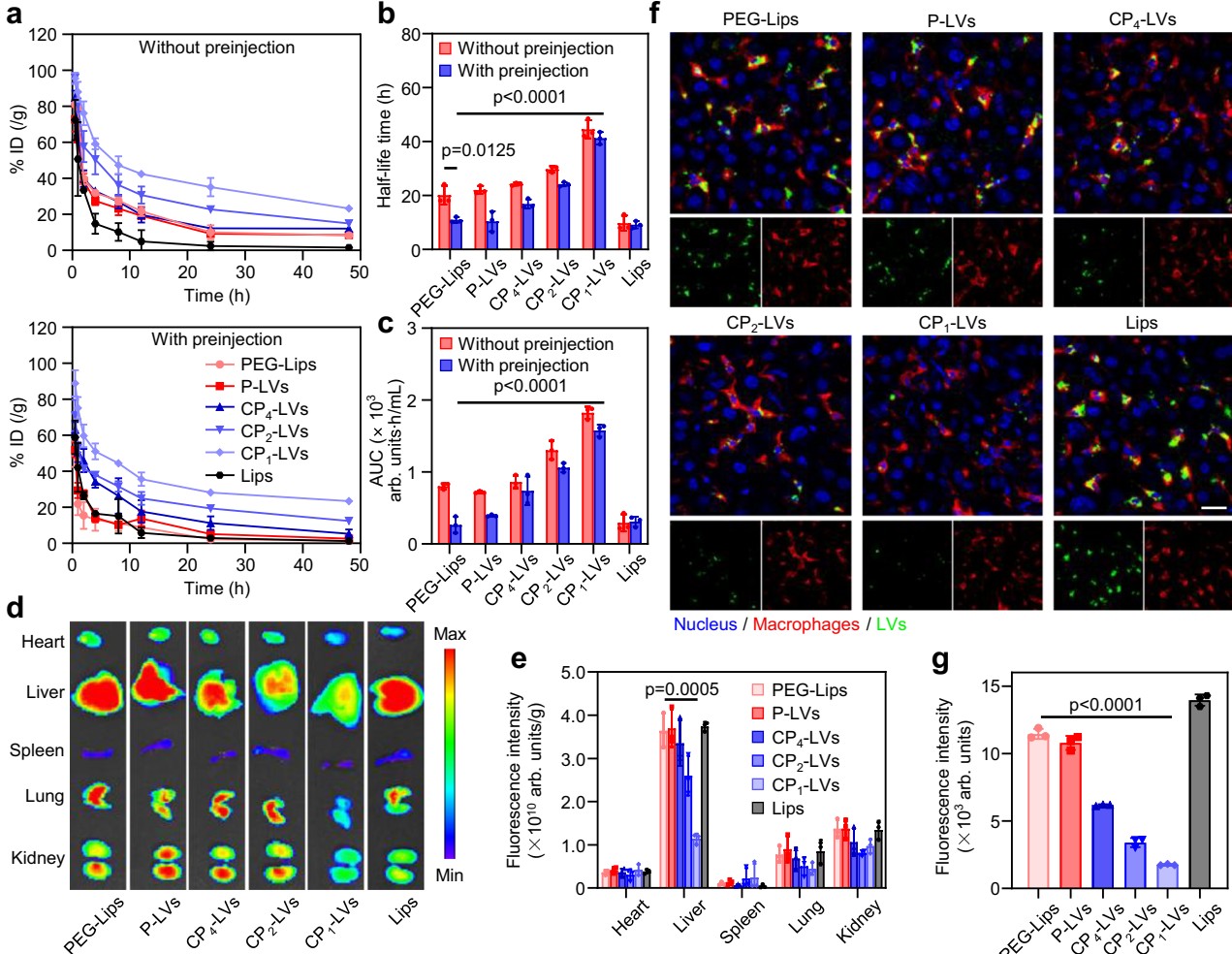

**Fig. 6 | The protein corona composition, circulation, and biodistribution of nanovesicles in vivo. a** Blood circulation curves, **b** related elimination half-life, and **c** AUC of nanovesicles in mice with or without preinjection. The data are displayed as the mean ± SD (two-tailed Student's *t*-test. *n* = 3 rats). **d** Ex vivo fluorescence imaging of major organs and **e** corresponding fluorescence intensities of the mice 12 h after i.v. injection. The data are displayed as the mean ± SD (two-tailed

Student's *t*-test. *n* = 3 mice). **f** CLSM images of liver sections showing the distribution of FITC-labeled nanovesicles (green) in liver tissues at 24 h postinjection. The Kupffer cells were stained with F4/80 (red), and the nuclei were stained with DAPI (blue). Scale bar = 20 μm. **g** Fluorescence intensity of nanovesicles in liver sections. The data are displayed as the mean ± SD (two-tailed Student's *t*-test. *n* = 3 independent experiments). Source data are provided as a Source Data file.

without preinjection, with a half-life of 41.3 h at the second injection, which was 3.8-fold longer than that of PEG-Lips (Fig. 6a, b). Moreover, CP₁-LVs possessed a significantly higher area under the curve (AUC$_{0-48h}$) than PEG-Lips (Fig. 6c). This was due to the low anti-PEG antibody level induced by CP₁-LVs weakening the ABC effect, and minimum adsorption of IgG/IgM on CP₁-LVs (Fig. 5h–j) further

decreasing the recognition and phagocytosis of macrophages. In general, CP₁-LVs with an appropriate amino/hydroxyl ratio could circulate in the blood for the long term regardless of single and repeated administration.

The biodistribution of IR783-labeled nanovesicles with repeated administration was evaluated in male ICR mice preinjected with each

nanovesicle. A large number of PEG-Lips were observed in livers at 12 h after injection, while $CP_1$-LVs were found to exhibit low distribution in livers, which was attributed to the low induction of anti-PEG antibodies and decreased IgG/IgM adsorption on $CP_1$-LVs (Fig. 6d, e). To further evaluate the prolonged in vivo blockage effect, the colocalization of nanocarriers and macrophages was investigated in liver tissue sections. A large number of PEG-Lips distributed in liver tissue sections and were colocalized with macrophages (Fig. 6f, g), indicating that most of the PEG-Lips were engulfed by Kupffer cells in the liver due to the anti-PEG antibody binding-activated complement system which could promote the phagocytosis of Kupffer cells. In contrast, profiting from low liver mesenchymal protein and IgG/IgM adsorption, few $CP_1$-LVs accumulated in the liver, avoiding Kupffer cell uptake. This finding was in accordance with the results of nanovesicle internalization in Kupffer cells. Taken together, $CP_1$-LVs exhibited the lowest binding affinity with plasma and liver TIF proteins, especially IgG and IgM, resulting in extended circulation.

## Dynamic protein coronas on nanovesicles in the transportation from blood to tumor

The overall therapeutic effect of nanocarriers in a pathological tissue depends not only on blood circulation but also on their internalization into target cells. It has been reported that protein coronas play an important role in nanocarrier internalization within tumor cells[18]. We then analyzed the dynamic composition of protein coronas on nanovesicles when they entered tumor tissues from blood (Fig. 7a). After incubated with the plasma and PBS, all the nanovesicles adsorbed the increased proteins in the TIF of A549 and HeLa tumors, with at least 1.7-fold higher amount of proteins than that in plasma (Fig. 7b). PEG-Lips showed the decreased adsorption of tumor TIF proteins compared to Lips due to the steric hindrance and hydrophilicity of PEG chains, while nanovesicles with increasing amino density showed the enhanced tumor stromal protein adsorption; in particular, $CP_1$-LVs exhibited the highest adsorption capacity. Meanwhile, the protein coronas formed in tumor TIF contained more types of proteins than those formed in plasma (Fig. 7c). Then, the detailed protein compositions were analyzed by LC-MS/MS, and were roughly divided into four categories, including cytoskeletal proteins, cell membrane proteins, extracellular matrix (ECM) proteins and metabolic enzymes. The proteins in the ECM and on the tumor cell membranes occupied a high proportion in the protein corona formed both in A549 and HeLa tumors, which may affect the subsequent interaction of nanovesicles and tumor cells (Fig. 7d). The abundant proteins classified as ECM proteins and cell membrane proteins in the protein corona formed in tumor TIF were further analyzed and found that the amount of OPN as an ECM component and transmembrane protein CD44 were higher than other proteins (Fig. 7e). The expression of OPN and CD44 in the protein corona was verified by Western blotting (Fig. 7f). CD44 and OPN were adsorbed on Lips in a large amount, and the adsorption amount of proteins on PEG-Lips and P-LVs was obviously reduced due to the polyhydroxy polymer modification. More importantly, the enhanced OPN and CD44 adsorption was observed on nanovesicles with increasing amino/hydroxyl ratios, and $CP_1$-LVs adsorbed the most OPN and CD44, similar to the results of LC-MS/MS.

During the transportation into liver and tumor from blood, $CP_1$-LVs exhibited different adsorption behaviors of proteins in liver and tumor TIF, especially the adsorption of IgG, OPN, and CD44. It is reported that electrostatic effect plays an important role in nonspecific protein adsorption, which is affected by the isoelectric point (pI) of proteins[46]. The pIs of IgG, CD44, and OPN are significantly different reported in the literature[47]. The pI of IgG is approximately 8.0, while those of CD44 and OPN are below 5.5. We speculated that the different isoelectric points of IgG, OPN, and CD44 resulted in the opposite adsorption behaviors on the $CP_1$-LVs, and verified it through isoelectric

focusing electrophoresis (IEF). Free proteins (IgG, OPN, and CD44), nanovesicles (P-LVs and $CP_1$-LVs) incubated with proteins were loaded at pH of about 7.4 in the electrophoresis gel with pH range of 5.0–8.0 (Fig. 7g). During electrophoresis, free IgG migrated to the high pH side, free CD44 and OPN moved in the opposite direction. The proteins in the nanovesicle-protein complexes showed different protein band patterns compared to free proteins. P-LVs obviously adsorbed IgG, but were rarely overlap with CD44 and OPN signals. In contrast, $CP_1$-LVs with amino modification were colocalized with more CD44 and OPN and less IgG than P-LVs (Fig. 7h). It may be due to the fact of that proteins with low pI are negatively charged in normal physiological environment (pH 7.4), tending to bind to positively charged groups (such as amino group), while IgG with high pI is positively charged and easily repelled by positive groups. Hence, $CP_1$-LVs with an appropriate amino/hydroxyl ratio selectively adsorbed proteins with low pI, including CD44 and OPN, resulting in the adsorption of many tumor TIF proteins, which may obviously affect the subsequent interaction with tumor cells.

## Effects of protein corona formed in tumor environment on tumor cell internalization

OPN and CD44 as important components of protein coronas formed in tumor environment, may play important roles in cellular uptake of nanovesicle in tumor cells. It has been reported that OPN overexpressed in a variety of cancer cells, contains an RGD domain that has strong affinity for cells overexpressing integrin receptors ($\alpha v\beta 3$ and $\alpha v\beta 5$) and is often designed for tumor targeting[48–50]. CD44 overexpressed in many tumors is a cell-surface glycoprotein involved in cell adhesion and migration, and has been reported to be the major hyaluronan (HA)-receptor to target tumor sites[51,52]. To explore the effect of protein corona on tumor cell internalization, the expression level of integrin receptor for OPN binding and transmembrane CD44 on A549, HeLa, and human umbilical vein endothelial cells (HUVEC) was determined, and the results showed that integrin and CD44 were highly expressed on the surface of A549 and HeLa cells, but were low in normal HUVEC cells (Fig. 8a). We then explored the cellular uptake of nanovesicles with tumor TIF protein coronas in tumor cells. As shown in Fig. 8b, PEG-Lips showed a weak fluorescence signal in A549 and HeLa cells, indicating low cellular uptake in target cells caused by steric hindrance of PEG chains and poor protein adsorption. With the enhanced protein adsorption in tumor TIF, $CP_1$-LVs showed the highest internalization into A549 and HeLa cells among all nanovesicle groups (Fig. 8c). On the contrary, $CP_1$-LVs with protein corona formed in tumor TIF exhibited little advantage in uptake in the HUVEC cells with low expression of integrin and CD44, compared with other nanovesicle groups (Fig. 8b). It suggested that OPN and CD44 in protein coronas, as typical proteins related to adhesion and uptake of tumor cells, may promote nanovesicle internalization through OPN-integrin and CD44-mediated ligand-receptor binding[53]. To investigate the role of OPN and CD44 opsonization in cellular uptake, cilengitide (CL) was used to inhibit the interaction between OPN and integrins[54,55], and free hyaluronic acid (HA) was pretreated with tumor cells for competitive binding with CD44 to impede the nanovesicle-CD44 interaction[56,57]. The uptake efficiency in both A549 and HeLa cells was reduced after CL or HA preincubation with tumor cells (Fig. 8d), suggesting the important roles of OPN and CD44 in the tumor cell uptake of nanovesicles in the tumor environment. The uptake inhibition efficiency of PEG-Lips and P-LVs caused by CL or HA was less than 10%, while the cellular uptake of CP-LVs with amino modification was more obviously inhibited by CL or HA. $CP_1$-LVs showed the sharpest decline in cellular uptake after CL or HA incubation, with an inhibition rate of approximately 25% which was the highest among three CP-LVs groups. It should be noted that adsorption of single protein exhibited a limited effect on nanovesicle internalization; thus, the fluorescence intensity of $CP_1$-LVs showed less than 30% reduction after preincubation with CL

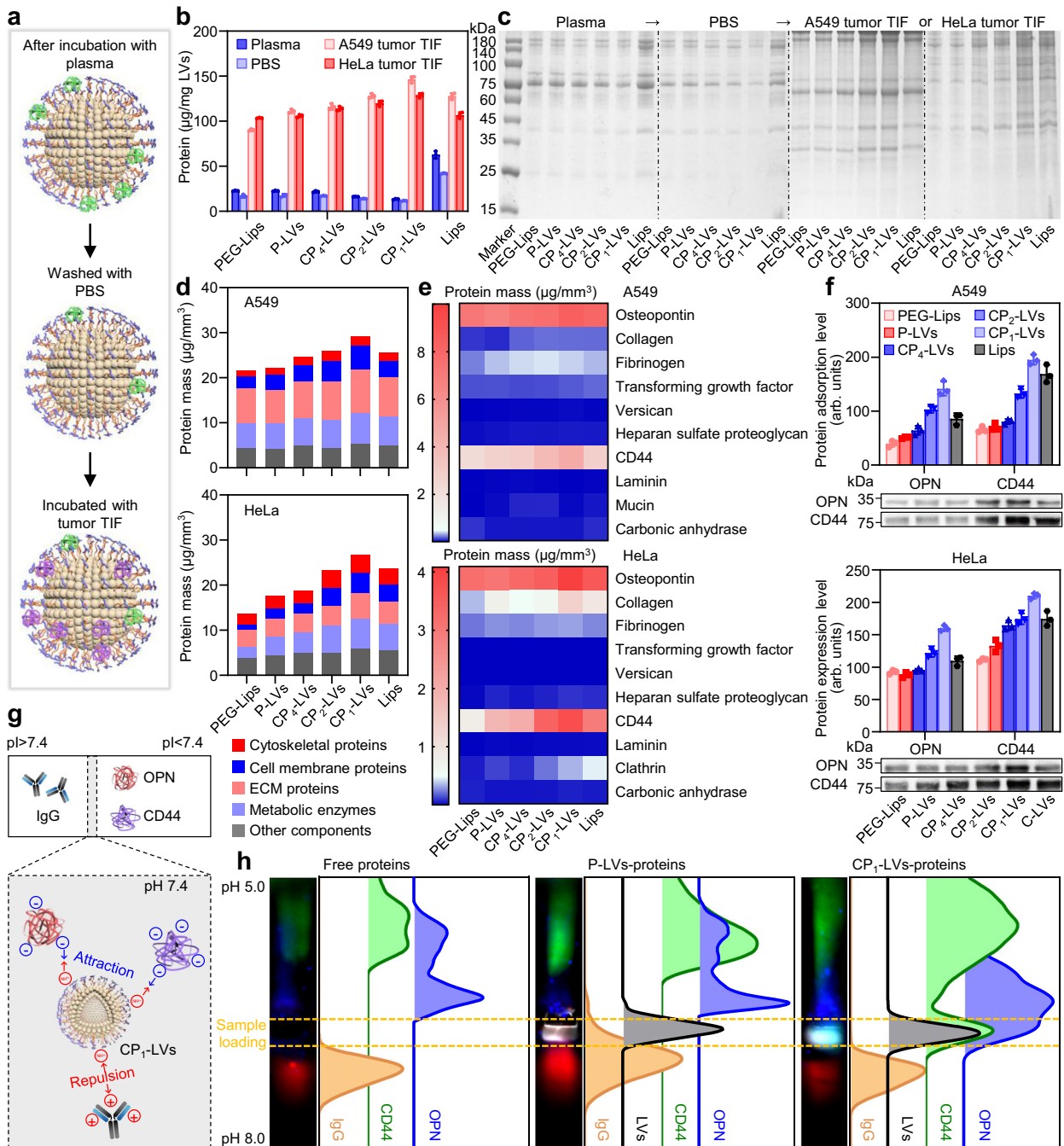

**Fig. 7 | Characterization of protein dynamic adsorption on nanovesicles.**
**a** Schematic diagram of nanovesicles dynamically adsorbing proteins. Nanovesicles were incubated with plasma, PBS, and tumor TIF in turn. **b** Quantification of proteins adsorbed to the nanovesicles (PEG-Lips, P-LVs, CP4-LVs, CP2-LVs, CP1-LVs, Lips) with incubation of plasma, PBS, and tumor TIF in turn. The data are displayed as the mean ± SD ($n = 3$ independent samples). **c** Qualitative molecular composition of the adsorbed protein layer on nanovesicles by SDS-PAGE in (**b**). **d** Classification of protein corona components identified by quantitative LC-MS/MS. **e** Heatmap of the most abundant proteins (top 10) in the protein corona of the nanovesicles with tumor TIF incubation determined by proteomic mass spectrometry. **f** OPN and CD44 bound to nanovesicles were determined by Western blotting assay. The data are displayed as the mean ± SD ($n = 3$ independent samples). **g** Schematic diagram of CP1-LVs adsorbing IgG, CD44, and OPN affected by protein pI. **h** IEF band images and intensity profile of IgG, CD44, OPN, and nanovesicles analyzed by ImageJ software. Red, green, and blue fluorescence signals represent IgG, CD44, and OPN, respectively. Gray fluorescence signal represents nanovesicles. Source data are provided as a Source Data file.

or HA, indicating that the high cellular uptake of CP1-LVs was caused by the synergism of various proteins, such as CD44, OPN and other proteins overexpressed in tumors.

Based on the high uptake in tumor cells, CP1-LVs are expected to exhibit superior antitumor efficacy. Paclitaxel (PTX), a classic chemotherapy drug, was selected as the model drug to be encapsulated into nanovesicles with a loading efficiency of approximately 10%

(Supplementary Fig. 13). The antitumor effects of PTX formulations against A549 and HeLa cells were then evaluated in vitro. As shown in Fig. 8e, all PTX formulations exhibited concentration-dependent cytotoxicity against A549 and HeLa cells. The IC50 value of PTX@CP1-LVs was 9.2 and 4.2 times lower than that of PTX@PEG-Lips in A549 and HeLa cells, respectively (Fig. 8f). In general, CP1-LVs adsorbed a large amount of protein in tumor TIF, which was conducive

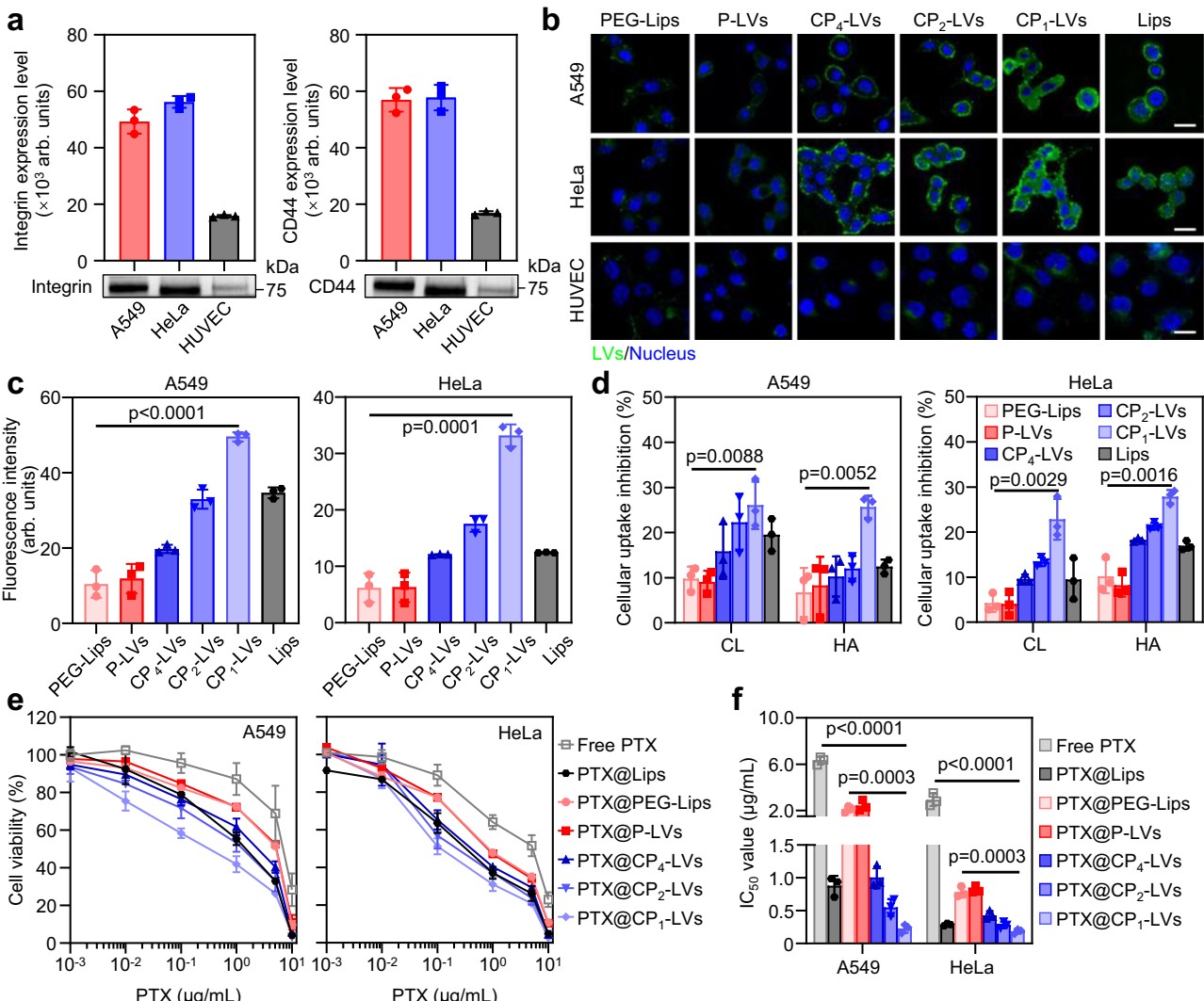

**Fig. 8 | The effect of the tumor protein corona on the internalization and efficacy of NPs in vitro. a** The expression levels of integrin and CD44 on A549, HeLa, and HUVEC cells determined by Western blotting. The data are displayed as the mean ± SD ($n = 3$ independent samples). **b** CLSM images of A549, HeLa, and HUVEC cells incubated with nanovesicles adsorbing tumor protein coronas. Scale bar: 20 μm. **c** Flow cytometric analysis of nanovesicle cellular uptake in A549 and HeLa cells. The data are displayed as the mean ± SD (two-tailed Student's $t$-test. $n = 3$ independent samples). **d** The inhibition rate of nanovesicle cellular uptake in A549

and HeLa cells pretreated with cilengitide (CL) or hyaluronic acid (HA). The data are displayed as the mean ± SD (two-tailed Student's $t$-test. $n = 3$ independent samples). **e** Cell viability of A549 and HeLa cells after incubation with PTX formulations. The data are displayed as the mean ± SD ($n = 3$ independent experiments). **f** The $IC_{50}$ values of PTX formulations in panel (**e**). The data are displayed as the mean ± SD (two-tailed Student's $t$-test. $n = 3$ independent experiments). Source data are provided as a Source Data file.

to enhancing their selectivity to tumor cells, thus enhancing the uptake of tumor cells and achieving good antitumor effect in vitro.

### Protein corona analysis in xenograft tumors and in vivo tumor accumulation

The protein corona formed in tumor interstitial fluid in vivo was further investigated in A549 or HeLa tumor-bearing mice (Fig. 9a). $CP_1$-LVs adsorbed the most proteins in tumor tissues, especially the adsorption of OPN and CD44 (Fig. 9b–d), similar to the results of in vitro incubation experiments, which could promote tumor cell internalization and may be beneficial to the delivery of nanovesicles to tumors. To investigate the biodistribution and tumor accumulation of the nanovesicles, IR783-labeled nanovesicles were intravenously injected into tumor-bearing mice and monitored using an IVIS Spectrum System. $CP_1$-LVs exhibited the highest tumor accumulation, with 2.6- and 2.2-fold higher fluorescence intensity than PEG-Lips in A549 and HeLa tumors, respectively (Fig. 9e–g). This was due to the avoidance of rapid

clearance by the reticular endothelial system and selective uptake in tumor cells.

The biodistribution of nanovesicles in tumors was further observed using CLSM. The images of tumor sections treated with PEG-Lips showed green fluorescent dots scattered far from the nucleus, while the fluorescence signal of $CP_1$-LVs was enriched in the tumor sections and distributed around the nucleus (Fig. 9h, i). This was attributed to the fact that $CP_1$-LVs adsorbed a large number of tumor stromal proteins, including OPN and CD44, which could increase the internalization efficiency of nanovesicles into tumor cells. In general, $CP_1$-LVs could avoid rapid clearance by the reticular endothelial system and be selectively internalized into tumor cells, thus may achieving optimal antitumor efficacy.

### Antitumor efficacy of PTX@CP-LVs in tumor-bearing mice

Based on the investigation of protein corona and nanovesicle accumulation in tumors, we then explored the antitumor potency of the

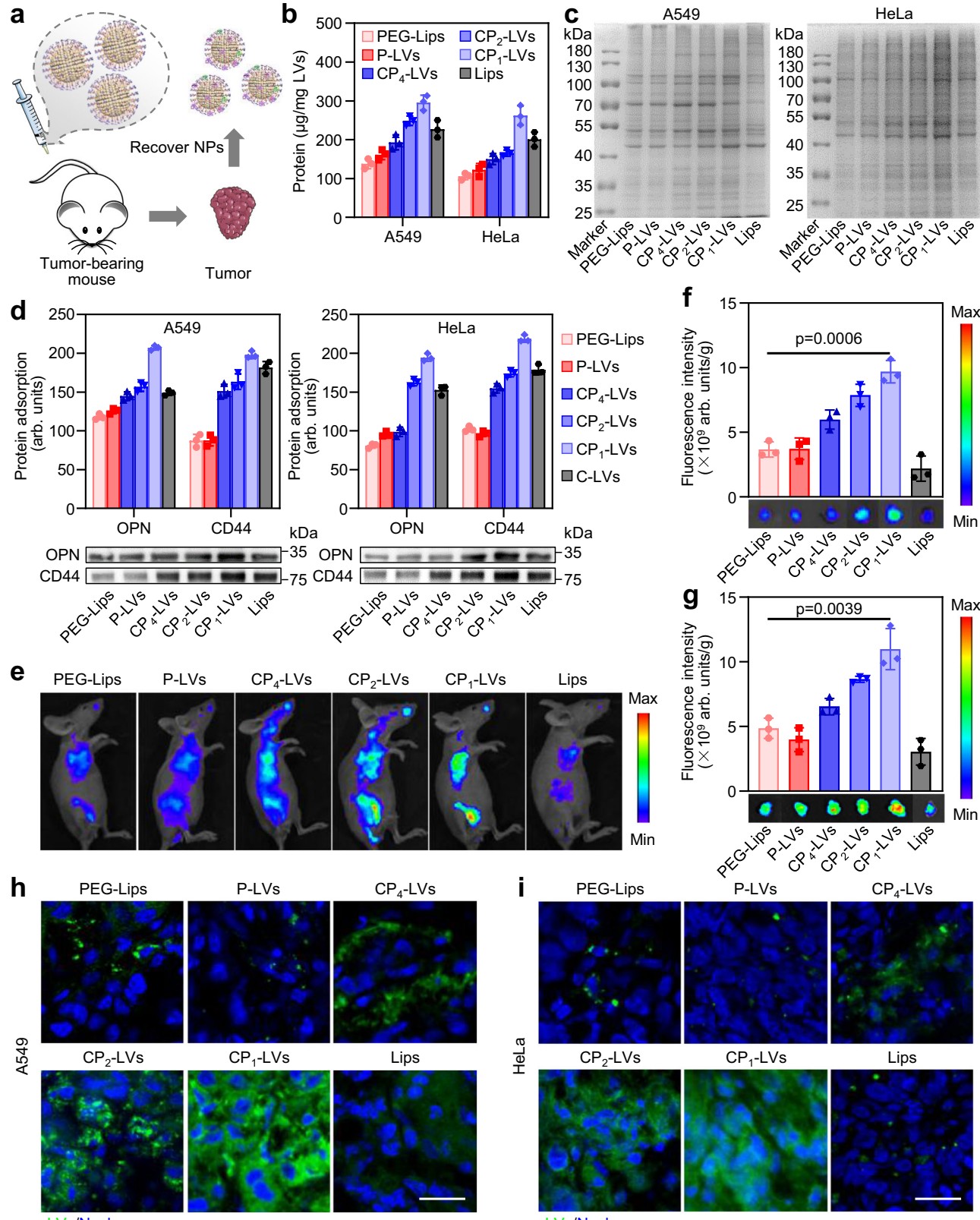

**Fig. 9 | The protein corona composition and accumulation of nanovesicles in tumors. a** Schematic illustration describing the characterization of protein coronas on nanovesicles recovered from tumors in mice injected with different nanovesicles. **b** Quantification of proteins adsorbed to the nanovesicles recovered from A549 and HeLa tumors. The data are displayed as the mean ± SD ($n = 3$ independent samples). **c** Qualitative molecular composition of the adsorbed protein layer on nanovesicles recovered from A549 and HeLa tumors by SDS-PAGE. **d** Western blotting assay of CD44 and OPN adsorbed on nanovesicles recovered from A549 and HeLa tumors. The data are displayed as the mean ± SD (two-tailed Student's $t$-test. $n = 3$ independent samples). **e** In vivo fluorescence images of tumor-bearing mice after injection of nanovesicles. **f** Ex vivo fluorescence images and quantitative analysis of A549 and **g** HeLa tumors at 48 h postinjection. The data are displayed as the mean ± SD (two-tailed Student's $t$-test. $n = 3$ mice). **h** Biodistribution of DiO-labeled nanovesicles in A549 and **i** HeLa tumor sections. Nuclei were stained with DAPI. Scale bar: 10 μm ($n = 3$ independent samples). Source data are provided as a Source Data file.

nanovesicles in A549 and HeLa carcinoma-bearing BALB/c nude mice. Mice were randomly allocated into 8 groups and intravenously injected with PBS, PTX-loaded nanovesicles, and commercially available paclitaxel injection (Free PTX) (Fig. 10a). The tumor size and body weight of each mouse were recorded once every 3 days after the first injection. Compared with the control groups, all the PTX formulations showed obvious tumor growth inhibition (Fig. 10b, c). The inhibition value of mice injected with PTX@CP$_1$-LVs reached more than 90%, which was significantly higher than that of PTX@Lips and PTX@PEG-Lips (Fig. 10d, e). Also, PTX@CP$_1$-LVs prolonged the overall survival of mice to at least 57 days without weight loss compared with PBS and Free PTX groups (Fig. 10f–h). Moreover, H&E staining images showed that tumor tissues from the PTX@CP$_1$-LVs group underwent substantial amounts of apoptosis and obvious tissue loss in a large area (Fig. 10i). Regarding safety, PTX@CP$_1$-LVs treatment did not cause any weight loss in mice (Fig. 10h). Histological examination of the main organs from the treated mice also revealed that PTX@CP$_1$-LVs exhibited no obvious alteration compared to the PBS group, indicating their good biological safety (Fig. 10j and Supplementary Fig. 14).

Taken together, PTX@CP$_1$-LVs had good antitumor effects and biocompatibility in both tumor models, significantly inhibiting tumor growth and prolonging the survival period of mice without obvious toxicity, and thus were promising for potential use in the clinical application of cancer therapy. The reasons can be explained as follows: Proteins absorbed on liposomes underwent dynamic change in vivo. Through surface modification, PTX@CP$_1$-LVs with an appropriate amino/hydroxyl ratio of approximately 0.4 absorbed the lowest proteins and immunoglobulins in plasma and maintained little protein adsorption during transportation from plasma to livers. Therefore, PTX@CP$_1$-LVs can avoid the capture of the mononuclear phagocyte system, enhance retention in the blood circulation, and accumulate more in tumor tissues. In addition, PTX@CP$_1$-LVs adsorbed high amounts of proteins in tumor TIF, enhancing the cellular uptake in tumor cells.

## Discussion

Protein coronas formed on nanocarrier surfaces significantly affect drug delivery. PEGylation is one of the most popular approaches for the preparation of antifouling nanosurfaces, thus yielding long-circulating nanocarriers. Unfortunately, protein absorption on the liposomal surface is inevitable even in the presence of PEG. Additionally, PEGylated nanocarriers bind to specific anti-PEG antibodies as they are injected repeatedly, leading to the ABC effect. Meanwhile, PEGylation hinders the cellular uptake of nanocarriers in the target cells, limiting the overall therapeutic efficacy of PEGylated nanocarriers. In this study, glycosylated polyhydroxy polymers were successfully synthesized to fabricate nanovesicles modified with different ratios of amino and hydroxyl groups. The zeta potential of CP-LVs with or without protein coronas were neutral (Supplementary Fig. 15), suggesting the differences in protein corona formation and cellular uptake were not caused by zeta potential. Hence, the different surface functional groups were the main variable to explore the protein corona composition on the nanovesicle surface and their delivery fate, particularly blood circulation and tumor cell internalization.

PEG-Lips as a control group could reduce plasma albumin adsorption and avoid phagocytosis of macrophages in blood, while the dynamic adsorption of liver TIF proteins on nanovesicles promoted liver macrophage recognition and phagocytosis, limiting the prolonged circulation of PEGylated nanovesicles. More importantly, the production and binding of anti-PEG antibody induced the enhancement of PEG-Lip internalization within blood macrophage J774 cells and liver Kupffer cells, shortening the half-life time in blood circulation and increasing the distribution in livers. In contrast, CP-LVs with amino modification exhibited decreased binding affinity with albumin and IgG in plasma and liver TIF, respectively. CP$_1$-LVs with the highest

amino/hydroxyl ratios dynamically adsorbed few proteins during transport from plasma to the liver, especially IgG/IgM, contributing to the prolonged circulation. Notably, CP$_1$-LVs induced extremely low anti-PEG antibody level due to the low immunogenicity shielding opsonin binding, and adsorbed little IgG/IgM after repeated administration, weakening the consequential ABC effect. Benefiting from the low adsorption of IgG/IgM, CP$_1$-LVs extended the blood circulation time, whether administered in single or multiple injections, and may increase the delivery efficiency of nanovesicles in target tissues.

Protein adsorption in the tumor microenvironment further affects nanocarrier accumulation in tumors. CP$_1$-LVs selectively adsorbed a large amount of tumor microenvironment-related protein and showed the enhanced internalization in tumor cells and tumor accumulation, which was attributed to synergism of various proteins, such as CD44, OPN, and other proteins overexpressed in tumors. The opposite phenomena of protein adsorption on CP$_1$-LVs in tumor TIF and liver TIF, which may be due to the different pIs of proteins. The functional groups on the surface of CP$_1$-LVs and other nanovesicles (P-LVs, CP$_2$-LVs, and CP$_4$-LVs) were the main variable, which can affect the nanovesicle-protein interaction through electrostatic effect. The protein pI plays an important role in the electrostatic effect between nanovesicles and different proteins. It is reported that the proteins with pI below 5.5 tend to bind to positively charged groups (such as amino group)[58]. Tumor TIF proteins with pI below 5.5, including CD44 and OPN, were easily to be adsorbed on CP$_1$-LVs with high amino/hydroxyl ratio, promoting nanovesicle internalization in tumor cells through CD44- and OPN-mediated pathways. In contrast, proteins in liver TIF that possess pIs higher than the environment pH, such as IgG, are positively charged and electrostatically repels the amino groups on CP$_1$-LVs. These resulted in the low adsorption of IgG and high binding of CD44 and OPN on CP$_1$-LVs.

Ultimately, benefiting from protein corona regulation mediated by surface modification, PTX-loaded CP$_1$-LVs significantly increased anticancer therapeutic efficacy in A549 or HeLa tumor-bearing mice compared to classical PTX-loaded PEGylated liposomes while exhibiting good biocompatibility. Therefore, this study may pave the way for the development of a highly innovative and translational approach for tumor treatment. More importantly, this work proposed the concept of subtly adjusting the functional group ratios and spatial arrangement for the rational design of nanocarrier surface engineering to control the protein corona composition on nanocarrier surfaces for efficient drug delivery.

## Methods

### Animals

The study involving animals was approved by the Institutional Animal Care and Use Committee at Shanghai Institute of Materia Medica (IACUC code: 2017-04-GY-31). Animal experiments were carried out in compliance with internationally recognized guidelines, and there were no instances of misuse or ethical violations.

### Chemical and reagents

Pluronic P123 (PEO$_{20}$-PPO$_{70}$-PEO$_{20}$, ~5800 kDa) was obtained from Sigma Aldrich. Chitosan oligosaccharide (1162 kDa, deacetylation degree: >90%) was provided by Shandong Weikang Biomedical Technology. N, N′-disuccinimidyl carbonate, 4-dimethylaminopyridine, acetonitrile, and ethanol were obtained from Sinopharm Chemical Reagent. Paclitaxel (98%) and 4′, 6-diamidino-2-phenylindole (DAPI) were obtained from Dalian Meilun Biotechnology. Dipalmitoyl phosphatidylcholine (DPPC) was purchased from A.V.T. Pharmaceutical.

### Synthesis of CSO-g-TCP

CSO-g-TCP was synthesized through a two-step route. First, PEO$_{20}$-PPO$_{70}$-PEO$_{20}$ (0.25 mmol) was dissolved in 5–10 mL acetonitrile and treated with N, N′-disuccinimidyl carbonate (DSC, 0.3 mmol) and

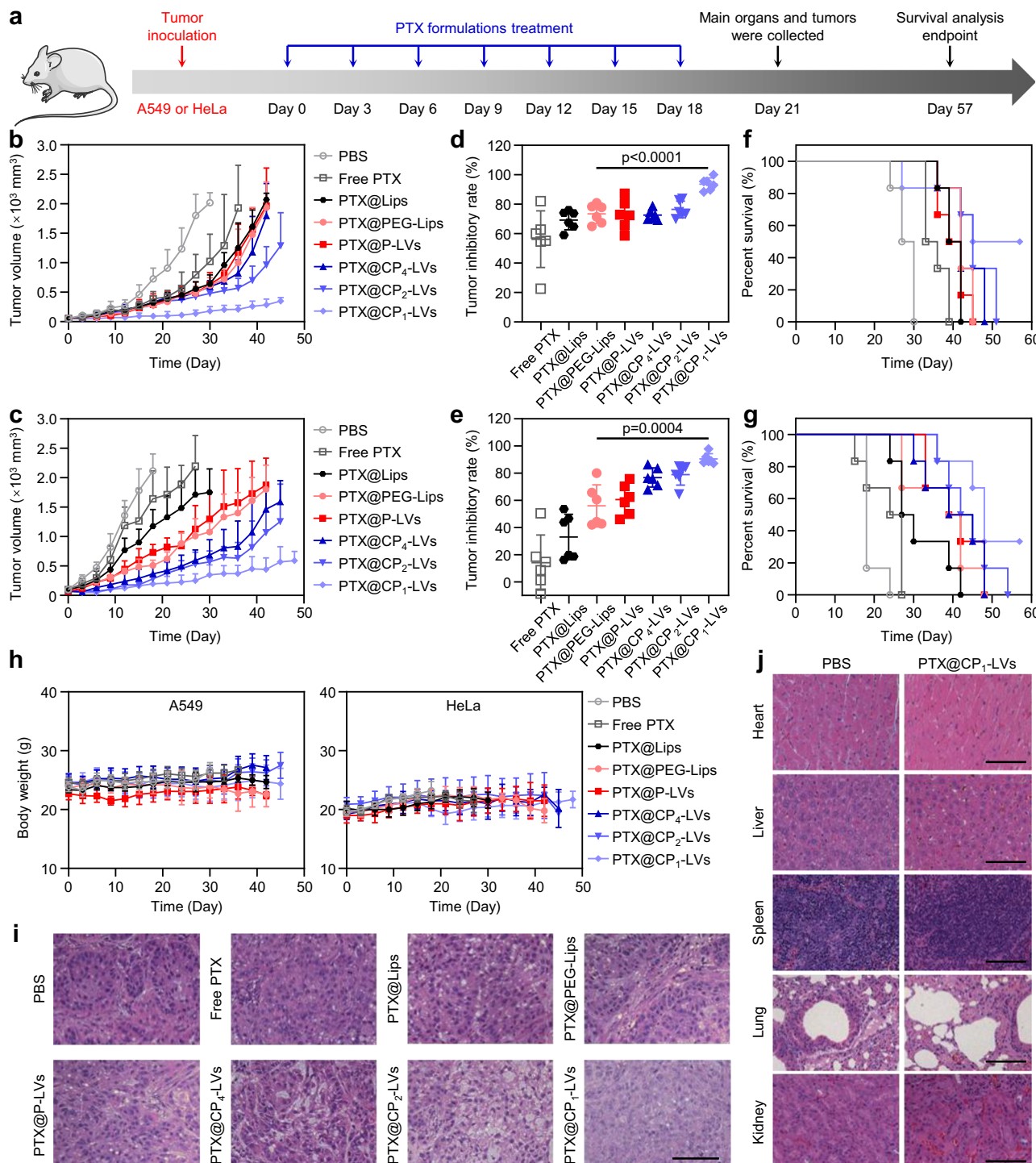

**Fig. 10 | Antitumor efficacy of PTX-loaded nanovesicles in vivo. a** Schematic diagram of PTX formulation therapy in A549 or HeLa tumor models. **b** Tumor growth kinetics of A549 and **c** HeLa xenograft tumors in BALB/c mice treated with PTX formulations. The data are displayed as the mean ± SD (*n* = 6 mice). **d** A549 and **e** HeLa tumor growth inhibition rate of PTX formulation treatments in comparison to the PBS group. The data are displayed as the mean ± SD (two-tailed Student's *t*-test. *n* = 6 mice). **f** Survival profiles of A549 and **g** HeLa tumor-bearing mice in various treatment groups. The data are displayed as the mean ± SD (*n* = 6 mice). **h** Body weight of treated A549 and HeLa tumor-bearing mice over time. The data are displayed as the mean ± SD (*n* = 6 mice). **i** Representative H&E-stained histological sections of A549 tumors after treatment. Scale bar: 100 μm. **j** Representative H&E-stained histological sections of the main organs after treatment. Scale bar: 100 μm. Source data are provided as a Source Data file.

4-dimethylaminopyridine (DMAP, 1.25 mmol) overnight to synthesize TCP-SC. The product was precipitated and dried under vacuum. CSO-g-TCP was prepared by adding different molar amounts of chitosan oligosaccharide to TCP-SC in deionized water. The resulting solution was stirred for 24 h to react. The product was concentrated and dialyzed in water by dialysis tubes (molecular weight cut off 3500 Da) to remove unreacted chitosan oligosaccharide (thin-layer chromatography confirmed that the unreacted chitosan oligosaccharide was removed completely), evaporated, and freeze-dried. The final product was stored at −20 °C.

## Preparation and characterization of CSO-g-TCP-modified lipid vesicles (CP-LVs)

CP-LVs were prepared by thin-film hydration. CSO-g-TCP with 1 μmol of TCP, DPPC (4 mg) and cholesterol (1 mg) were dissolved in ethanol and dried to form a thin lipid film. The lipid films were then hydrated with 1 mL of PBS (pH = 7.4) at 55 °C to obtain crude nanovesicle suspensions. The suspensions were extruded through a 100-nm polycarbonate membrane using a microextruder device to obtain the CP-LVs. To prepare the control formulations P-LVs, 1 μmol TCP was dissolved in ethanol, dried, and then hydrated with PBS. PEGylated liposomes (PEG-Lips) were prepared by hydration of a thin lipid film composed of 7.2 mg HSPC, 2.4 mg DSPE-PEG 2000, and 2.4 mg cholesterol. Common liposomes (Lips) were prepared by hydration of a lipid film containing DPPC (10 mg) and cholesterol (2 mg).

The particle size distribution and zeta potential of the nanoparticles were determined by a Malvern Zetasizer Nano ZS analyzer (Worcestershire, UK). Nanoparticles were evaluated for shape by a transmission electron microscope (TEM, Tecnai G2 Spirit Twin, FEI, USA).

The content of amino and hydroxyl groups was determined by Boehm titration. For amino content determination, 10 mL of CP-LVs was acidified using an equal volume of HCl (0.1 M) and then titrated with 0.05 M NaOH. At the end of the titration, the molar of HCl minus the molar of NaOH for titration is the amino group content. For hydroxyl content determination, samples were alkalinized with NaOH and titrated with HCl. The molar of NaOH minus the molar of HCl for titration is the content of hydroxy groups. The amino and hydroxyl contents in liposomes containing DPPC and cholesterol were determined as background values. The spatial arrangement of functional groups was analyzed using time-of-flight secondary ion mass spectrometry (IONTOF Tof-SIMS 5-100, Germany). CP-LVs were thinly coated on silicon substrates and freeze-dried. Ion images were acquired with Tof-SIMS using a 30-kV $Bi_1^+$ beam as the primary ion beam and a 5-kV Ar-Cluster as the sputtering beam. The analysis area of chemical mapping images was $35 \times 35$ μm$^2$ with the pixel size of about 120 nm for the maximal spatial resolution. The signals of amino and hydroxyl groups were detected by Tof-SIMS at different depths. The density of the amino group was recorded during the whole scan.

## Proteomic identification of proteins adsorbed on nanovesicles in plasma and liver TIF

To measure the level of protein adsorbed to the CP-LVs or liposomes, the CP-LVs or liposomes were incubated with plasma (total protein concentration 23 mg/mL) at 37 °C for 1 h. Then, the nanovesicle-protein complexes were isolated by centrifugation at $100,000 \times g$ at 4 °C using an ultracentrifuge (Hitachi, Japan), and resuspended in 50 μL RIPA lysates for the BCA protein assay and SDS-PAGE characterization. To ensure the similar nanovesicle amount used in the process of nanovesicle-protein incubation and characterization, CP-LVs or liposomes were labeled with DiO in the experiment to control nanovesicle concentration through the measurement of fluorescence intensity by a microplate reader (BioTek, USA). For SDS-PAGE, the nanovesicle-protein samples loaded in the gels were fluorescently imaged through a biomolecular imager (Typhoon 9500, GE, USA) to ensure the consistent nanovesicle amounts in the samples.

The proteomics of absorbed protein was determined by LC-MS/MS. The proteins loaded in the SDS-PAGE gels were handled through reduction, alkylation, and trypsinization in turn at 37 °C overnight after gel electrophoresis. Then, the protein samples were desalted, followed by freeze-drying. Finally, the proteins were resuspended in 0.1% formic acid and analyzed by LC-MS/MS.

Immunoglobulins IgG and IgM were measured by Western blotting. The adsorbed proteins were transferred from SDS-PAGE gels to polyvinylidene fluoride (PVDF) membranes, and blocked with skimmed milk. The target proteins were recognized through the incubation

of anti-IgG (Abcam, ab133470, 1:5000 dilution) and anti-IgM primary antibodies (Abcam, ab170492, 1:2000 dilution) overnight at 4 °C. Then, the PVDF membrane was incubated with HRP-conjugated anti-rabbit second antibody (Beyotime Biotechnology, A0208, 1:1000 dilution), followed by imaging through a ChemiDoc MP™ Imaging System (Bio-Rad Hercules, CA, USA).

Liver TIF was isolated through the method reported[59]. Briefly, fresh liver tissue was cut into small pieces in PBS with a protease inhibitor cocktail on ice. Then, the tissue pieces were washed with PBS and immersed in an equal volume of PBS followed by incubation at 37 °C for 1 h. The sample was centrifuged at 1000, 2000, and $20,000 \times g$ in turn, and the supernatant was collected to obtain the final liver TIF. The nanovesicles were incubated with liver TIF with similar protein concentration to plasma at 37 °C for 1 h. The nanovesicle-protein complexes were collected by ultracentrifugation and detected through LC-MS/MS.

## Dynamic protein adsorption on nanovesicles in plasma-PBS-liver TIF incubation system

To observe the dynamic behavior of the proteins adsorbed on the nanovesicles moved from plasma to the liver, we incubated nanovesicles in plasma, PBS and liver TIF in turn. Briefly, nanovesicles were incubated with plasma for 1 h. After washing three times with PBS, the nanovesicle-protein complexes were incubated in PBS for 1 h, and the nanovesicles were pelleted by ultracentrifugation. The pellets were resuspended in PBS and incubated with liver TIF for 1 h. Then, the nanovesicle-protein complexes were collected by centrifugation after incubation with plasma, PBS, and liver TIF, respectively. The pellets were resuspended in 50 μL RIPA lysates for BCA protein assay, SDS-PAGE, and dot blotting.

For quantification of adsorbed albumin and IgG, the adsorbed proteins were quantified with a dot-blot assay as follows. 2 μL aliquots were applied onto a 0.45-μm-pore PVDF membrane. The membranes were blocked with skimmed milk for 1 h at room temperature, probed with anti-IgG and anti-albumin primary antibodies (Abcam, ab207327, 1:2000 dilution) at 4 °C overnight, and then incubated with the HRP-conjugated anti-rabbit secondary antibodies (Beyotime Biotechnology, A0208, 1:1000 dilution) against the primary antibody species. The signal was detected by a Bio-Rad imaging system and quantified using ImageJ software.

## Mechanistic study of dynamic protein adsorption

Isothermal titration calorimetry was used to determine the binding affinity constant of protein and nanovesicles[60,61]. Albumin and IgG were dissolved in PBS and dialyzed in PBS overnight. The nanovesicles were hydrated with the medium after protein dialysis to avoid the endothermic or exothermic reaction caused by different solvents during titration. The ITC tests were performed in a microcalorimeter (MicroCal ITC200) at 25 °C. Nanovesicles were injected into the sample cell containing protein solutions with a stirring speed of 750 rpm. Titration of nanovesicles was performed as a first injection of 0.4 μL, followed by a series of 2 μL injections. The ITC measurements were analyzed and fitted into a one-site binding model in Origin Software.

To observe the dynamic changes in protein adsorption on the nanovesicle surface, albumin and IgG were labeled with FITC and RITC, respectively. Briefly, the proteins were dispersed in PBS, followed by the addition of FITC (or RITC), and stirred for 24 h. The proteins were then dialyzed with a 10 kDa pore size in PBS for 24 h and freeze-dried for storage. The DiD-labeled nanovesicles were incubated with FITC-albumin and RITC-IgG in turn and then separated from residual proteins by centrifugation. The nanovesicle-protein complexes were observed by confocal laser scanning microscopy (Olympus FV1000, Japan) at a magnification of 100×. The single nanovesicle adsorbing proteins were zoomed in for further observation through CLSM with stimulated emission depletion (STED) mode (Leica, Germany).

## Protein Corona Analysis in plasma and livers in vivo

To analyze the protein corona composition formed in vivo, the DiO-labeled nanovesicles were intravenously injected into male BALB/c nude mice. Blood sample was collected for nanovesicle-protein complex isolation at 2 h after injection. Plasma containing nanovesicles was loaded onto a Sepharose CL-4B column and eluted by HEPES buffer solution (150 mM NaCl and 20 mM HEPES). Fractions containing nanovesicles were concentrated and washed using ultrafiltration centrifugal tubes (10000 and 1000000 MWCO, Sartorious)[62] to remove unadsorbed proteins. The UV absorption at 280 nm determined by a microplate reader was used to verify the successful recovery of nanovesicles in plasma. The nanovesicle amount was standardized through the nanovesicle fluorescence intensity. At 24 h postinjection, the liver TIF-containing nanovesicles were collected using the same procedure of liver TIF isolation, and nanovesicle recovery from liver TIF was performed using the same procedure of nanovesicles isolated from plasma. The composition of protein coronas on nanovesicles was studied by BCA protein assay kit and SDS-PAGE. The adsorption level of albumin and IgG on the nanovesicles were analyzed by Western blotting.

To study the anti-PEG expression in plasma from mice with nanovesicle injection, the nanovesicles were intravenously injected into male ICR mice (4–6 weeks), and the plasma was collected at day 7 post-first injection for the detection of anti-PEG expression using an anti-PEG IgM ELISA Kit.

To analyze the protein corona formed on nanovesicles in plasma after repeated administration, DiO-labeled nanovesicles were intravenously injected again into male ICR mice 7 days after the first injection. The plasma was collected to recover the nanovesicles at 2 h after injection. The amount and type of proteins adsorbed on nanovesicles were studied by BCA protein assay kit and SDS-PAGE. The adsorption levels of IgM and IgG on the nanovesicles were analyzed by Western blotting.

## Cellular uptake of nanovesicles in macrophages

To investigate the effect of the protein corona on the circulation of nanoparticles at the cellular level, the J774 cell line and rat primary Kupffer cells isolated from rat livers were used for uptake experiments. The Kupffer cells were isolated as reported[43,63]. Briefly, after anesthesia of the rat, the liver was perfused with HBSS solution to expel blood. The hepatic cells were fully digested with collagenase solution (0.5%) and collected by cutting the liver lobes. Then, the Kupffer cells were purified by differential centrifugation and selected by cell adhesion on 12-well plates.

For the cellular uptake experiments, J774 cells and Kupffer cells were pretreated with fresh serum-free medium to eliminate the interference of serum on the cellular uptake, and then incubated with DiO-labeled nanovesicles. Before the addition of nanovesicles, the nanovesicles preadsorbed the proteins in plasma (or plasma & liver TIF). The nanovesicles without protein coronas or with different protein coronas were added to the macrophages. After incubation, cells were detached with 2.5% trypsin followed by detection with BD FACScalibur flow cytometer. For imaging, cells cultured on glass coverslips were incubated with DiO-labeled nanovesicles. After washing and fixation, the cells were stained with DAPI and imaged using CLSM.

To explore the effect of the protein corona on cellular uptake of nanovesicles with repeated administration, plasma, and liver TIF were isolated from mice preinjected with nanovesicles. The nanovesicles with protein coronas were then added to J774 and Kupffer cells. The nanovesicle contents within cells were evaluated by flow cytometry and confocal imaging.

## Pharmacokinetics and biodistribution studies

For pharmacokinetics and biodistribution studies, FITC or the near-infrared fluorescent dye IR783 was covalently labeled with lipids (DPPC or HSPC). Male SD rats (200–220 g) and male BALB/c nude mice (4–6 weeks) were obtained from Shanghai Lab. Animal Research Center. For the pharmacokinetics study, eighteen SD rats were randomly divided into six groups: $CP_1$-LVs, $CP_2$-LVs, $CP_4$-LVs, P-LVs, Lips, and PEG-Lips. Then, the FITC-labeled nanovesicles were injected into the rats intravenously. At the indicated time points postinjection, blood samples were collected and centrifuged to obtain the plasma. The plasma sample was mixed with acetonitrile in equal volumes and vortexed for 5 min. Then, the solution was centrifuged at 8000 rpm for 10 min, and the supernatant was recovered for detection through a microplate reader. For the pharmacokinetic study of nanovesicles after repeated administration, FITC-labeled nanovesicles were intravenously injected into the male SD rats at 7 days after preinjection with each nanovesicle. Blood collection and nanovesicle fluorescence measurement were performed using the same procedure of pharmacokinetics study of single dose. The pharmacokinetic profiles were analyzed by DAS software.

For the biodistribution study, IR783-labeled nanovesicles were injected into male BALB/c nude mice intravenously. At 12 h post-injection, the mice were sacrificed, and tissues (hearts, livers, spleens, lungs, kidneys) were collected at necropsy for imaging. The fluorescence intensity of organs was quantified by the ROI measurement tool in IVIS software.

To investigate the distribution of nanovesicles in livers, the livers were collected at 24 h after administration of FITC-labeled nanovesicles into mice, and cryo-sectioned using a Leica CM 1950 Cryostat (Leica, Germany). Then, the Kupffer cells in the liver slices were labeled with anti-F4/80 antibody (Abcam, ab300421, 1:100 dilution), and the nucleus was stained with DAPI, followed by imaging through a Leica microscope (DMi8).

## Dynamic protein adsorption on nanovesicles in plasma-PBS-tumor TIF incubation system

Lung cancer and cervical cancer were selected as tumor models to investigate the effect of the protein corona in tumor TIF on nanovesicle cellular uptake in tumor cells. A549 cells ($5 \times 10^6$ cells) and HeLa cells ($2 \times 10^6$ cells) were separately implanted subcutaneously into the armpits of male and female BALB/c nude mice (4–6 weeks) to establish xenograft tumors. The isolation of tumor TIF was performed using the same method as that used for liver TIF. The nanovesicles were incubated with plasma, PBS, and tumor TIF in turn, and centrifuged for BCA and SDS-PAGE analysis. The composition of protein coronas formed in A549 or HeLa tumor TIF was analyzed through LC-MS/MS. The OPN and CD44 adsorbed on the nanovesicles were identified and determined by Western blotting. The proteins transferred to PVDF membranes were probed with anti-OPN (Abcam, ab283656, 1:1000 dilution) and anti-CD44 (Abcam, ab189524, 1:1000 dilution), and were detected after incubated with the HRP-conjugated anti-rabbit secondary antibody (Beyotime Biotechnology, A0208, 1:1000 dilution).

The mechanism study of $CP_1$-LVs adsorbing IgG, CD44, and OPN in different trends was explored by IEF electrophoresis. DiO-labeled nanovesicles ($CP_1$-LVs and P-LVs) were incubated with liver TIF and tumor TIF for 1 h. The samples including free proteins and nanovesicle-protein complexes were loaded to the electrophoresis gel with 1% agarose and 2% ampholyte (a pH range of 3–10). 5% phosphoric acid and 2% sodium hydroxide were added in the positive and negative electrophoretic tanks, respectively. IEF electrophoresis was performed at 160 V voltage for 20 min. The proteins were transferred from gel to PVDF membranes, and then labeled with anti-IgG, anti-OPN, or anti-CD44. Finally, the proteins on PVDF membranes were recognized by Alexa Fluor 647-labeled second antibody (Yeasen Biotechnology, 34213ES60, 1:200 dilution), and imaged together with gels containing nanovesicles through a biomolecular imager (Typhoon 9500, GE, USA). The fluorescence signals of IgG, CD44, OPN, and nanovesicles were fitted together by ImageJ software, subsequently quantifying the fluorescence intensity.

## Tumor protein adsorption affects nanovesicle internalization in tumor cells

The expression of integrin and CD44 on tumor cells or normal HUVEC cells were determined by Western blotting. The lysates of A549, HeLa, and HUVEC cells with total protein content of 30 μg were loaded on SDS-PAGE gels, and transferred to PVDF membranes. The integrin and CD44 were labeled with anti-integrin beta 1 (Abcam, ab179472, 1:1000 dilution) and anti-CD44 (Abcam, ab189524, 1:1000 dilution), and were detected using a Bio-Rad imaging system.

To investigate the effect of tumor protein adsorption on nanovesicle internalization in tumor cells, A549 or HeLa or HUVEC cells were seeded on glass coverslips and cultured overnight. DiO-labeled nanovesicles were incubated with tumor TIF for 1 h, followed by incubated with cells for 2 h. The cells were stained with DAPI and imaged using CLSM. For quantitative analysis, A549 and HeLa cells were detached and collected for flow cytometry after incubated with nanovesicles adsorbing tumor TIF proteins.

For the mechanistic study of tumor protein adsorption affecting nanovesicle internalization in tumor cells, cilengitide, and hyaluronic acid were pretreated with A549 or HeLa cells for 1 h. Then, the cells were incubated with DiO-labeled nanovesicles adsorbing tumor proteins for 2 h and detached for detection using a BD FACS flow cytometer.

To study the antitumor activity of PTX-loaded nanovesicles, A549 or HeLa cells were seeded in 96-well plates ($1 \times 10^4$ cells/well) and treated with free PTX, PTX@Lips, PTX@PEG-Lips, PTX@P-LVs, PTX@CP$_4$-LVs, PTX@CP$_2$-LVs and PTX@CP$_1$-LVs at different concentrations for 48 h. Then, the MTT assay was performed.

## Protein corona analysis and nanovesicle distribution in tumors in vivo

To establish xenograft tumor models, A549 cells ($5 \times 10^6$ cells) were implanted subcutaneously into the right flanks of 4- to 6-week-old male mice, and HeLa cells ($2 \times 10^6$ cells) were implanted subcutaneously into female BALB/c nude mice. The DiO-labeled nanovesicles were intravenously injected into A549 or HeLa tumor-bearing BALB/c nude mice. At 24 h postinjection, the tumor TIF-containing nanovesicles were collected using the same procedure of tumor TIF isolation, and nanovesicle recovery from tumor TIF was performed using the same method of nanovesicles isolated from liver TIF. The composition of proteins adsorbed on nanovesicles was analyzed by BCA assay and SDS-PAGE. The OPN and CD44 adsorbed on the nanovesicles were determined by Western blotting.

For the biodistribution study, A549 cells ($5 \times 10^6$ cells) were implanted subcutaneously into the right flanks of 4- to 6-week-old male BALB/c nude mice to establish xenograft tumors. When the tumor volume reached approximately 200 mm³, IR783-labeled nanovesicles were injected into the mice intravenously, and monitored using the IVIS system at 48 h. At 48 h postinjection, the mice were sacrificed, and tumors were collected at necropsy for imaging. The fluorescence intensity of organs was quantified by the ROI measurement tool in IVIS software. The nanovesicles accumulated in HeLa tumors were performed using the same procedure as those in A549 tumors.

To study the distribution of nanovesicles in tumor tissues, tumors were isolated from mice at 12 h after injection of DiO-labeled nanovesicles, and cryo-sectioned into 20-μm thick slices. After fixation and DAPI staining, the fluorescent signals of nanovesicles were observed by CLSM (Olympus FV1000, Japan).

## In vivo antitumor efficacy

A549 cell suspension ($5 \times 10^6$ cells) was implanted into the axilla of male BALB/c nude mice to establish subcutaneous xenografts. When tumor volumes reached 50-100 mm³, the mice were randomly divided into 8 groups for the following treatments: (1) PBS, (2) free PTX, (3) PTX@Lips, (4) PTX@PEG-Lips, (5) PTX@P-LVs, (6) PTX@CP$_4$-LVs, (7) PTX@CP$_2$-LVs, and (8) PTX@CP$_1$-LVs. Tumor-bearing mice were intravenously administrated with different formulations at a PTX dosage of 10 mg/kg once every three days for 7 times. The tumor volume and mice body weight ($n = 6$ for each group) were monitored over time, and survival was defined as natural death or tumor volume > 2000 mm³. The tumor volume was calculated as follows: volume = (length × width²)/2. After treatment for 21 days, the major organs including heart, liver, spleen, lung, and kidney, and the tumors were dissected for H&E staining. All histological analysis was carried out by the Center for Drug Safety Evaluation and Research (CDSER, SIMM CAS). The antitumor efficacy of PTX formulations in HeLa tumors implanted in the female BALB/c nude mice was performed using the same method as that in A549 tumors.

## Statistical analysis

Significant differences were determined using unpaired Student's *t*-test for two group comparisons and ANOVA for comparisons of multiple treatment groups within individual experiments. $P < 0.05$ was considered significant. All values are displayed as the mean ± standard deviation (SD).

## Reporting summary

Further information on research design is available in the Nature Portfolio Reporting Summary linked to this article.

## Data availability

The authors declare that all data needed to support the findings of this study are presented in the article, Supplementary information, and Source data file. A reporting summary for this article is available as a Supplementary Information file. Source data are provided with this paper (figshare https://doi.org/10.6084/m9.figshare.24612237).

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

## Acknowledgements

This study was sponsored by the National Science Fund for Distinguished Young Scholars (Grant No. 82025032), the National Natural Science Foundation of China (Grant No. 82104113, 82222066 and 81973250), the China Postdoctoral Science Foundation (No. 2022T150675), the Natural Science Foundation of Shanghai (No. 21ZR1475800) and the State Key Laboratory of Drug Research (No. SIMM2103ZZ-01). We thank the staff members of the Integrated Laser Microscopy System at the National Facility for Protein Science in Shanghai (NFPS, ZJLab) for their help in collecting confocal microscopy and flow cytometry data. We also thank the staff members of the Electron Microscopy System and the Large-scale Protein Preparation System at the National Facility for Protein Science in Shanghai (NFPS), Shanghai Advanced Research Institute, Chinese Academy of Sciences, China for providing technical support and assistance in data collection and analysis.

## Author contributions

Y.G. and X.Z. conceived and designed the experiments. Y.M., L.L., L.G., QY.Z., and D.N. performed the experiments, and Y.M. analyzed the data. Y.M. prepared the original manuscript. All authors edited, reviewed, and approved the manuscript.

## Competing interests

The authors declare no competing interests.
