## [Peer Review File · Nature Communications]

Reviewers' Comments:

Reviewer #1:

Remarks to the Author:

In this study, lipid nanovesicles modified by glycosylated polyhydroxy polymers with different amino/hydroxyl group ratios (CP1-LVs, CP2-LVs, and CP4-LVs) were investigated on the composition of protein coronas formed. The results showed that CP-LVs with higher amino/hydroxyle ratio had lower non-specific protein binding in plasma and liver interstitial fluid, while the higher binding capability to tumor interstitial fluid, making them circulate longer in blood, and more prone to accumulation in tumor tissues after injection. A higher tumor-killing efficacy was achieved when used as an anti-tumor therapeutic carrier.

Major issues:

The authors reported a CP-LV nanostructure (CP1-LVs) with high amino density on the surface, showing decreased adsorption of proteins in plasma and livers. Still, they sharply increased the adsorption of proteins in tumor TIF, compared with the other nanostructures like CP2-LVs, CP4-LVs, P-LVs, and PEG-Lips (fig. 4c vs. fig. 7b). A significant issue is how this opposite protein adsorption profile occurred for the same nanovesicles interacting with different TIFs. The difference in protein composition in tumor and liver TIF should be characterized and compared. Protein composition in protein corona formed on nanovesicles in these media should also be carefully characterized and compared.

To be more convincing, in vitro protein corona analyses should be performed to mimic the in vivo characterization. The nanovesicles should incubate with, in the order of plasma, PBS, and tumor TIF, to mimic the in vivo translocation process.

Other minor issues:

1. In Fig 5 and other places, the Anti-PEG antibody level in protein coronas on the nanovesicle surface was analyzed or discussed. It does not make much sense to me because PEG is not a component of CP-LVs nanovesicle surface molecules. It is not unexpected to see no or low PEG antibodies after applying these nanovesicles.
2. Some discussion should be included in the text to postulate possible mechanisms or reasons contributing to CP1-LVs's superior protein corona formation property in plasma and tumor interstitial fluid compared to other nanoparticles.
3. y-axis titles of Fig. 3a,4b,5b,5d,7a, and 8b should be protein (μg)/ μg nanovesicles since normalization is mentioned in the method section. Otherwise, data are not comparable, e.g., in Fig 5b and 5d.
4. I strongly suggest including a brief method description in both the main text and captions of the figures.

Reviewer #2:

Remarks to the Author:

This manuscript is interesting and well-organized. Most conclusions have been proved by the authors. However, there are still some issues should be modified and clarified before the further consideration.

1. The position of the tri-block polymer in liposomes was not clearly. Was it coating on the surface? Was It properly to use PEG-liposomes as the control? The PEG molecules was inserted in liposomes using DSPE part.
2. Absence of PEG, why CP1-LVs with the highest amino density induced the lowest anti-PEG antibody level. It doesn't make sense. To reveal the reasons, those antibodies such as anti-CP1 antibody should be evaluated.

3. What is the full name of OPN? Osteopontin?
4. Amino and hydroxyl groups are not bioselective, why glycosylated polyhydroxy polymer can offer nanocarriers bioselectivity?

Reviewer #3:

Remarks to the Author:

Miao et al. have synthesized so-called lipid "nanovesicles" modified with glycosylated polyhydroxy polymers as an alternative to PEG, and determined the impact of subtle differences in surface functional groups with respect to protein adsorption.

The study is comprehensive, but the authors need to be much more specific in terms of describing the experiments (for instance, what do they mean by "plasma"), and they also need to be more stringent in terms of conclusions; the manuscript is replete with unsubstantiated "conclusions" (speculative statements). The authors are advised to take the following into consideration:

Abstract and Introduction read reasonably well and the authors cover relevant background literature with respect to nanoparticles and the protein corona, and tumor targeting/uptake. However, the Introduction ends with the statement that the present study has shown "striking therapeutic benefits in tumor treatment". This is not an appropriate statement. First, the word "striking" is not appropriate and should be deleted. Second, the authors have not evaluated any anti-tumor treatment in patients, only in conventional subcutaneous xenograft models in nude mice.

The schematic diagram (Figure 1) is hardly useful, though it may certainly serve as a "graphical abstract". As it stands, this schematic figure contains almost no information on the nanoparticles (except "amino/hydroxyl"). The figure (and the subsequent data reported in the Results section) does, however, raise an unanswered question: if the "nanovesicles" with the highest amino content (designated as CP1-LVs) display the lowest protein adsorption in plasma (also lower than the PEGylated control particles), why do the very same particles or vesicles display the highest adsorption of proteins in tumor-derived tissue interstitial fluid (TIF)? What is the mechanism? The authors have not addressed this question anywhere in the manuscript not even in the Discussion, but this is a crucial point that needs to be addressed in order to make sense of the present work. As it stands, it is difficult to understand the mechanism since the authors have not performed unbiased investigations of every plasma-derived, or liver TIF-derived, or tumor TIF-derived protein. Instead, the authors have decided to focus on a few representative proteins, e.g., albumin and IgG (in plasma) and osteopontin and CD44 (in tumor TIF). Therefore, further studies are required to address this conundrum (that is, why does a high amino content repel plasma proteins such as albumin yet attract tumor interstitial fluid-derived proteins?).

Results: synthesis and characterization of the nanoparticles is good, but the authors also need to establish whether the particles are endotoxin-free (or not) prior to i.v. injection in mice. Furthermore, some basic toxicity assessment is also required, not using tumor cell lines, but using primary (mouse or human) cells, to determine the dose-response (from low dose to high dose) of each of the particles and whether varying the surface functional groups or amino content will make a difference. The size and zeta potential of the 3 nanovesicles is similar (100 nm, and a zeta potential that is "nearly neutral" (it is slightly negative). What is the zeta potential of these nanovesicles following their immersion in plasma, or liver/tumor TIF?

The authors incubated the various particles in "plasma", but they do not specify anywhere in the manuscript (neither in Results nor in the Methods section) whether this means "human plasma" or "mouse plasma" which means that one cannot properly evaluate the results. Moreover, was any anti-coagulant used? if human plasma, then how many human donors were used (the samples are presumably pooled samples of plasma from several donors)? If human donors, then this information (including an ethical board approval) has to be included in the Methods section. Or did the authors use mouse plasma? Minor comment: no need to say "multitudinous" proteins; this is also ambiguous (a high protein content overall, or many different types of plasma proteins?). In this section, the authors spend a great deal of time discussing some of the findings, including the

finding that albumin, fibrinogen and "various immune-associated proteins" are present in the corona. They seem to focus their attention in particular on albumin, an abundant albeit low-affinity protein, and on IgG (and IgM). However, the results show that several apolipoproteins are present in the corona (including apolipoprotein A-I, apolipoprotein A-IV, apolipoprotein C, and clusterin, also known as apolipoprotein J) and these proteins are widely known to bind to nanoparticles; why are these protein corona constituents not mentioned or discussed? In particles, clusterin, which has been shown to bind (specifically) to bind to polymer-modified nanocarriers, leading to a reduced (not enhanced) uptake (Nat Nanotechnol. 2016;11(4):372-7). In fact, previous work has shown that for nanoparticles with low poly(ethylene glycol) (PEG) coverage, adsorption of apolipoproteins can prolong circulation times, whereas the low-density-lipoprotein receptor plays a predominant role in the clearance of nanoparticles, irrespective of PEG density (Nat Commun. 2017;8(1):777). To summarize, the rationale for choosing to focus on albumin and IgG is not clear.

Furthermore, here (and in several other places) the authors insert "conclusions" while such conclusions are not relevant. For instance, "possibly avoiding the interaction with the immune system and displaying low immunogenicity" (page 10); similar statements are found on almost every page of the Results section ("which may be beneficial to avoiding macrophage phagocytosis", "which may be constructive to avoiding macrophage recognition and phagocytosis", and so on). Such "conclusions" should be avoided; this is speculation and it is also not obvious a priori that a rich protein corona would necessarily lead to uptake as some corona proteins may also act as dysopsonins and not as opsonins.

Further down, the authors claim that they injected the nanoparticles intravenously and then retrieved the particles "from plasma and liver". How were the particles retrieved from the liver? This seems non-trivial and it is not explained in the text. The authors then studied the interactions of the particles with "J774 cells and Kupffer cells". Some clarification is needed (i.e., the murine macrophage-like cell line J774, and rat primary Kupffer cells) (and clarify whether these cells were maintained in serum-free medium, which seems to be the case according to the Methods section, as this is important for the interpretation of the results). The authors have to be more stringent in describing these experiments; the use of isolated Kupffer cells does not mean that "local phagocytosis of local Kupffer cells in the liver" has been investigated (this belongs to the subsequent section of the paper). In other words, authors need to be clear whether they are performing or discussing in vitro results or in vivo results.

In reference to the in vivo biodistribution studies using ICR mice: the authors, again, draw conclusions regarding the role of individual corona proteins (e.g., role of IgG/IgM) without having any evidence. The fact that different particles are taken up to a different extent by macrophages in the liver does not prove a role for specific proteins; this was not studied and it is merely extrapolation. To prove a role of specific ligands, authors have to knock down the corresponding receptors (Nat Commun. 2017;8(1):777). It is very likely that apolipoproteins also play a role. In the subsequent section, the authors investigate whether the nanoparticles are taken up by A549 lung adenocarcinoma cells or HeLa cervical carcinoma cells. However, they refer to these models as "A59 and HeLa tumors" which is wrong; they are just cell lines. Here, the authors select OPN and CD44 as typical cancer overexpressed proteins, but in order to prove a role for these proteins (in the corona of the nanoparticles), the authors must knock down the corresponding receptors in the cancer cells. The presence of a particular protein in the corona does not necessarily mean that it is a functional epitope for receptor binding, nor does it prove that it is required for "targeting".

Finally, authors investigate uptake (targeting) and anti-tumor efficacy in a conventional subcutaneous xenograft model (in nude mice). Again, uptake is not "targeted" unless uptake is reduced in mice carrying tumors with low or silenced receptor expression. Furthermore, the authors should test whether the anti-tumor effects of nanoparticles delivering paclitaxel are mitigated when "targeting" is reduced (by silencing or blocking relevant receptors). This is a key point if the authors want to argue that the tuning of the protein corona is important for efficient drug delivery, as implied by the title of the present manuscript. There is no formal proof to support the suggestion (on page 30) that the presence of OPN or CD44 in the corona have anything to do with tumor cell uptake; this is, again, merely speculation. There is also little proof that the

particles are "selectively" internalized by tumor cells. One question that has not been addressed is the uptake of these particles in normal cells; it is possible that particles are taken up and then released by exocytosis while the particles may be retained in cancer cells.

Overall, a comprehensive, preclinical study, but the reasons for the reduced coronation of particles with high amine content with plasma proteins and the simultaneous enhanced corona formation in tumor-derived fluids remains highly perplexing. The long-term stability of these "nanovesicles" also remains to be investigated.

Reviewer #4:

Remarks to the Author:

Summary

The present work describes a new design of nanoparticles, mainly vesicles, which are made of a polymer with different surface modifications to modulate the interaction of the formed vesicles with proteins from the in vivo environment, such as plasma proteins or from a liver extract. They study whether these new surface properties give an added value to the plain vesicles in order to probe their usefulness in vivo.

Language:

The language of the text includes too much technical jargon, is very difficult to read and the English is complicated, which makes the work very difficult to follow for people who are not experts in the field or in the type of techniques used by these researchers. It is my opinion that a multidisciplinary journal like Nat Com should naturally present a much more accessible language for researchers from other disciplines.

General comments:

My general impression is that there are many results collected with little lack of analysis. This means that, in my opinion, the hypotheses established are poorly supported by empirical evidence. I will now cite some of the examples I have observed in the first part of the work I have evaluated.

Abstract:

The summary of the article is complicated to understand due to the complicated use of the language and the excess of jargon and technical language that complicates the reading for those of us who are not experts in the field.

In the abstract (lanes 22-23) on one side about the evolution of the corona, it also says that the uptake of proteins is prevented (so, is there a corona?) but on the other side it says that albumin and immunoglobulin adsorption is efficiently suppressed, but on the other hand it says that proteins typical of tumors are adsorbed. To me, this seems to be a contradiction since this means there is a protein corona.

On the other hand, says (lanes 25-26) "Moreover, CP1-LVs adsorbed abundant tumor distinctive proteins containing CD44 and osteopontin ... mediating specific tumor cell internalization." On the one hand, CD44 is apparently an 85-200 KDa transmembrane glycoprotein that is present in most cells or tissues, so it does not appear to be specific. And, on the other hand, it is not understood why a tumor membrane protein (i.e. CD44) would cause the vesicles to have more affinity for the tumor cells or tissue, since it is not a specific ligand of any tumor receptor, but is part of the repertoire of tumor membrane proteins. so I think this argument is not valid and should not be in the abstract.

Introduction.

The authors should use references on how peptide- or protein-coated nanoparticles are able to interact with receptors in a specific way and not focus exclusively on the coatings they use. Figure 1 is not clear and is not really necessary, I think it is totally dispensable.

Some of the sentences are rather pretentious in the introduction and I think that as they are part of the results should be left for discussion or conclusions: i.e. "...Due to their strong 84 ability to prolong blood circulation and promote tumor cell internalization, CP1-LVs loaded with 85 paclitaxel (PTX) showed striking therapeutic benefits in tumor treatment...".

Or "... The present study 86 emphasizes the promising application of protein corona regulation to improve delivery efficiency 87 of nanocarriers, and promotes the rational design of functional groups modified nanocarriers."

Results.

As mentioned above, there are sentences that are excessively difficult to follow, such as those that follow from lines 98 to 104.

"... The amino-modified hydrophilic polymers were synthesized by grafting PEO20-PPO70-PEO20 triblock copolymers to the backbone of CSO through a diimide-activated amidation CSO-g-TCP4, CSO-g-TCP2 and CSO-g-TCP1 (Supplementary Fig. 2), respectively. CSO-g-TCP and DPPC were hydrated together to obtain a series of lipid nanovesicles with surface modification of CSO-g-TCP, structure of a spherical vesicle and an average size of approximately 100 nm, called CP4-LVs, CP2-LVs and CP1-LVs..."

Some sentences are repeated i.e. ... (lane 109) Surface modification of amino and hydroxyl groups determines the interaction of nanovesicles and proteins...

Figure 2: Characterization of CP-LVs

Fig 2a (lane 576) is this TEM or CryoTEM?;

Fig 2e, it says "...Representative images of amino and hydroxyl distribution at the top and middle sections observed by Tof-SIMS. Scale bar = 2 μ m..."

The authors should explain the rationale (fundamentals and calculations) for this technique, what it evaluates, and by what method, as I could not find it anywhere in the manuscript.

Fig2h. The statistics (t values) are not shown (only the n=3 and the SD), I guess that there are significant differences in the zeta potential of the different engineered particles. So if there are no statistically significant differences, I cannot understand how this could influence the adsorption of the three different nanoparticles in the three different nanoparticles. This makes me think that the difference in the composition of the nanoparticles, in the end, is not so important from a surface point of view. Hence, it is questionable whether this may have an effect on the differential adsorption of proteins that ultimately adhere to the nanoparticle surface because of electrostatic or hydrophobic forces.

Lanes 147: To study the composition of protein coronas on CP-LVs and the influence by the surface modification according to the amino density on the composition in a "physiological environment" authors incubate the particles in a protein solution of 23 mg/mL concentration of plasma used to analyze the protein corona formed on CP-LVs incubated with plasma. In fact, the actual plasma protein concentration is approximately 60–80 mg/mL of which about 50–60% are albumins and 40% globulins (10–20% immunoglobulin) so the environment in which the particles are incubated is not physiological.

On the other hand, they also say that they use a liver extract (lane 608), which does not give its protein concentration and only gives very rough data on how it is prepared. Actually, this extract could have any protein concentration, thus it may not even remotely mimic a physiological liver extract.

Here is a major point of the study. Coomassie-stained SDS:PAGE gels or western blots can only be semi-quantified when there are appropriate loading controls in the same sample, dot blots are not quantifiable and are not comparable among each other where the starting material is not identically quantified at the protein level.

Figure 3

The Coomassie-stained gels in Figure 3b, e, f, and the accompanying quantification in Figure 3d are unreliable without internal controls. For example, a standard protein could be added to the mix to ensure that the sample treatment and loading of each well of the gel have been identical, so that they can be semi-quantified.

I guess the need to do a BCA would be to put identical amounts of protein per well, so it makes no sense that the different lanes of the gel show difference in total protein (and I don't mean differences in types of protein). Since this is what has been measured, there should be no differences in the total amount of protein but in the distribution of the proteins.

But if doing this on an acrylamide gel transfer, where proteins separate neatly by size, is already inaccurate, it is much less imprecise to quantify the result of a western "dot" blot of a protein drop. (lane 620: "... For quantification of adsorbed albumin and IgG, the protein corona was quantified with a dot-blot assay ... with the corresponding HRP-conjugated secondary antibodies against... signal was detected using a Bio-Rad imaging system..."

Therefore, statements such as this (lane 158): "...CP1-LVs showed the lowest protein adsorption among all groups and weaker intensity of most bands compared to CP2-LVs and CP4-LVs...." are not credible on the data provided and should be changed. This can result from a defect in the gel well charge and can be remedied by placing internal load markers (which there are none in this case).

Figure 3b, in results (lane 155) states: "... Most intriguingly, increasing the density of amino groups on the surface of CP-LVs reduced total protein adsorption. Analysis of the protein corona by SDS-PAGE was consistent with the result of the amount of protein adsorbed on the nanovesicles (Fig. 3b)....." I really do not understand how this statement can be made in this manner when no controls are used for the quantification of the starting protein, and not a standard quantification of the initial protein or the adsorbed protein has been done.

Figure 4

The nanovesicles were incubated in plasma, PBS, and liver tissue interstitial fluid, and the proteins on the surface are compared using SDS:PAGE (Fig4c). Again, no loading controls are shown, so there could be many factors interfering with this result, from faulty loading or lack of adequate particle stripping.

Figure 4d: "...albumin and IgG bound to nanovesicles was determined by dot blotting assay..." As already mentioned, dot blots are not quantifiable and are not comparable to each other where the starting material is not identically quantified at the protein level.

Figure 4g: "...confocal images of nanovesicles incubated with albumin (upper panel) and IgG solution..." NPs are in the range of 100 nm and this is under the optical resolution (200nm x 500nm). A high-resolution confocal could have been used but only an Olympus FV1000 is mentioned so the images are merely representative. The settings of the microscope should be identical

(lane 198) Regarding the study where the dynamic exchange of proteins in the corona is done, it seems quite obvious, and according to the hundreds of studies in the literature that when a nanoparticle is coated with proteins, it always exchanges proteins when the particle is exposed to a different medium.

When this occurs the particle captures the proteins in the new medium and this process is dynamic. Therefore, it is expected that the finding of the article.

Up to this point I have been able to evaluate this work. I believe that the absence of controls at the biochemical level in each of the steps performed makes questionable the majority of the results shown at the protein level and, therefore, I believe that these design flaws detected will surely condition all the results and the conclusions that can be drawn from these data.

Reviewer #1 (Remarks to the Author):

In this study, lipid nanovesicles modified by glycosylated polyhydroxy polymers with different amino/hydroxyl group ratios (CP1-LVs, CP2-LVs, and CP4-LVs) were investigated on the composition of protein coronas formed. The results showed that CP-LVs with higher amino/hydroxyl ratio had lower non-specific protein binding in plasma and liver interstitial fluid, while the higher binding capability to tumor interstitial fluid, making them circulate longer in blood, and more prone to accumulation in tumor tissues after injection. A higher tumor-killing efficacy was achieved when used as an anti-tumor therapeutic carrier.

Major issues:

The authors reported a CP-LV nanostructure (CP1-LVs) with high amino density on the surface, showing decreased adsorption of proteins in plasma and livers. Still, they sharply increased the adsorption of proteins in tumor TIF, compared with the other nanostructures like CP2-LVs, CP4-LVs, P-LVs, and PEG-Lips (fig. 4c vs. fig. 7b). A significant issue is how this opposite protein adsorption profile occurred for the same nanovesicles interacting with different TIFs. The difference in protein composition in tumor and liver TIF should be characterized and compared. Protein composition in protein corona formed on nanovesicles in these media should also be carefully characterized and compared.

Response: The different protein adsorption on nanovesicles in liver TIF and

tumor TIF was due to the isoelectric point (pI) of proteins. To investigate the effect of protein pI on adsorption behaviors, IgG in liver TIF and CD44 (or OPN) in tumor TIF were used as representative proteins to performed isoelectric focusing electrophoresis. Free proteins (IgG, OPN and CD44), nanovesicles (P-LVs and CP₁-LVs) incubated with proteins were loaded at pH of about 7.4 in the electrophoresis gel with pH range of 5.0-8.0 (Fig. 7g). During electrophoresis, free IgG migrated to the high pH side, free CD44 and OPN move in the opposite direction, indicating the different pI of representative proteins in liver and tumor TIF. CP₁-LVs with amino modification were colocalized with more CD44 and OPN and less IgG than P-LVs (Fig. 7h). It may be due to the fact of that proteins with low pI are negatively charged in normal physiological environment (pH 7.4), tending to bind to positively charged groups (such as amino group), while IgG with high pI is positively charged and easily repelled by positive groups. We have added the data in the Fig. 7 and updated the results in the revised manuscript.

Fig. 7 g Schematic diagram of CP₁-LVs adsorbing IgG, CD44 and OPN affected by protein pl. **h** IEF band images and intensity profile of IgG, CD44, OPN and nanovesicles analyzed by Image J software. Red, green and blue fluorescence signals represent IgG, CD44 and OPN, respectively. Gray fluorescence signal represents nanovesicles.

The composition of protein corona in liver TIF analyzed by LC-MS/MS has been showed in Supplementary Fig. 5 in the submitted Supplementary information, and described in the section of “Dynamic exchange of the protein corona on nanovesicles” in the submitted manuscript. To clearly compare the compositions of protein coronas formed in plasma and liver TIF, we reprocessed the data and arranged the responding diagrams in Fig. 3g-i. The description of results has been added in the section of “The composition of protein coronas on CP-LVs influenced by surface modification” in the revised manuscript.

The liver contains thousands of proteins differing from plasma, which may obviously influence the protein corona compositions. Albumin, IgG and metabolic enzymes were adsorbed to the nanovesicle surface in large quantities. CP₁-LVs adsorbed the lowest IgG and liver metabolic enzymes among all nanovesicles, possibly avoiding the interaction with the immune system and displaying low immunogenicity.

Fig. 3 g Classification of protein corona components formed in the liver TIF identified by quantitative LC-MS/MS. **h** Heatmap of immunoglobulin and albumin in the protein corona of the nanovesicles with liver TIF incubation. **i** Heatmap of the most abundant proteins (top 10) in the adsorbed metabolic enzymes and heat shock proteins on nanovesicles with liver TIF incubation.

We have supplemented the characterization of protein corona composition in tumor TIF through LC-MS/MS. The data was shown in Fig. 7d, e. The adsorbed proteins contained cytoskeletal proteins, cell membrane proteins, extracellular matrix (ECM) proteins and metabolic enzymes. The proteins in the ECM and on the tumor cell membranes occupied a high proportion in the protein corona formed both in A549 and Hela tumors, which may affect the sequent interaction of nanovesicles and tumor cells. The most abundant proteins classified as ECM proteins and cell membrane proteins in the protein corona formed in tumor TIF

were further analyzed and found that the amount of osteopontin (OPN) as an ECM component and transmembrane protein CD44 were higher than other proteins. The description of results were added in the section of “Dynamic protein coronas on nanovesicles in the transportation from blood to tumor” in the revised manuscript.

Fig. 7 d Classification of protein corona components identified by quantitative LC-MS/MS. **e** Heatmap of the most abundant proteins (top 10) in the protein corona of the nanovesicles with tumor TIF incubation determined by proteomic mass spectrometry.

To be more convincing, in vitro protein corona analyses should be performed to mimic the in vivo characterization. The nanovesicles should incubate with, in the order of plasma, PBS, and tumor TIF, to mimic the in vivo translocation

process.

Response: According to the reviewer's comments, we have supplemented the analysis of protein corona on nanovesicles incubated with plasma, PBS and tumor TIF in turn, to mimic the in vivo translocation process. After incubated with the plasma and PBS, all the nanovesicles adsorbed increased proteins in the TIF of A549 and HeLa tumors, with at least 1.7-fold higher amounts of proteins than that in plasma (Fig. 7b). PEG-Lips showed decreased adsorption of tumor TIF proteins compared to Lips due to the steric hindrance and hydrophilicity of PEG chains, while nanovesicles with increasing amino density showed enhanced tumor stromal protein adsorption; in particular, CP₁-LVs exhibited the highest adsorption capacity. Meanwhile, the protein coronas formed in tumor TIF contained more types of proteins than those formed in plasma (Fig. 7c). The data was supplemented in Fig.7 in the revised manuscript.

Fig. 7 **b** Quantification of proteins adsorbed to the nanovesicles (PEG-Lips, P-LVs, CP₄-LVs, CP₂-LVs, CP₁-LVs, Lips) with incubation of plasma, PBS, and tumor TIF in turn. **c** Qualitative molecular composition of the adsorbed protein layer on nanovesicles by SDS-PAGE in panel **b**.

Other minor issues:

1. In Fig 5 and other places, the Anti-PEG antibody level in protein coronas on the nanovesicle surface was analyzed or discussed. It does not make much sense to me because PEG is not a component of CP-LVs nanovesicle surface molecules. It is not unexpected to see no or low PEG antibodies after applying these nanovesicles.

Response: Anti-PEG antibody induced by PEGylated nanocarriers specifically bind to the repeated units of $-O-CH_2-CH_2-$ which are also the main composition of PEO-PPO-PEO triblock copolymers (TCP) used in the synthesis of CSO-g-TCP. P-LVs as control groups were mainly composed of TCP, and induced high expression level of anti-PEG antibody (Fig. 5h), indicated that the repeated units of $-O-CH_2-CH_2-$ in the PEO segments of TCP could induce similar antibodies to PEG induced. CSO-g-TCP used in CP-LVs are synthesized through CSO and TCP. CSO has not been proved to induce specific antibodies. Therefore, anti-PEG as a marker of CP-LVs induced in vivo to evaluate the accelerate blood clearance is reasonable.

Fig. 5 h Expression level of anti-PEG antibodies in plasma collected from mice pretreated with nanovesicles.

2. Some discussion should be included in the text to postulate possible mechanisms or reasons contributing to CP₁-LVs'superior protein corona formation property in plasma and tumor interstitial fluid compared to other nanoparticles.

Response: According to the reviewer's comments, we have supplemented responding content in the section of "Discussion". The opposite phenomena of protein adsorption on CP₁-LVs in tumor TIF and liver TIF, which may be due to the different pI of proteins. The functional groups on the surface of CP₁-LVs and other nanovesicles (P-LVs, CP₂-LVs, and CP₄-LVs) was the main variable, which can affect the nanovesicle-protein interaction through electrostatic effect. The protein pI plays an important role in the electrostatic effect between nanovesicles and different proteins. It is reported that the proteins with pI below 5.5 tend to bind to positively charged groups (such as amino group). Tumor TIF proteins with pI below 5.5, including CD44 and OPN, were easily to be adsorbed on CP₁-LVs with high amino/hydroxyl ratio, promoting nanovesicle internalization in tumor cells through CD44- and OPN-mediated pathways. On contrast, proteins in liver TIF that possesses a pI higher than the environment pH, such as IgG, are positively charged and electrostatically repels the amino groups on CP₁-LVs. These resulted in the low adsorption of IgG and high

binding of CD44 and OPN on CP₁-LVs.

3. y-axis titles of Fig. 3a,4b,5b,5d,7a, and 8b should be protein (μg)/ μg nanovesicles since normalization is mentioned in the method section. Otherwise, data are not comparable, e.g., in Fig 5b and 5d.

Response: P-LVs and CP-LVs with similar molar TCP were incubated with plasma, liver and tumor TIF in vitro, PEG-Lips with similar PEG density to P-LVs and Lips with similar mass to CP-LVs were used as control groups. Then, the amount of proteins adsorbed on nanovesicles was determined by BCA assay. In fact, the nanovesicle mass used in the protein corona characterization was similar. Additionally, the protein corona analysis in vivo was performed through fluorescence-labeled nanovesicles. The nanovesicle content in nanovesicle-protein complex recovered from plasma, liver and tumor TIF were standardized through fluorescence intensity of fluorescence-labeled nanovesicle. Then, the BCA results of protein corona in vivo represented the content of proteins adsorbed by the same amount of nanovesicles. According to the reviewer's comments, we have recalculated the results of protein corona contents through conversion of protein amount divided by nanovesicle mass, and revised the units of y-axis.

Fig. 3 **a** Quantification of plasma proteins adsorbed to the nanovesicle surface determined by BCA assay.

Fig. 4 **b** Quantification of proteins adsorbed to the nanovesicles (PEG-Lips, P-LVs, CP₄-LVs, CP₂-LVs, CP₁-LVs, Lips) with incubation of plasma, PBS, and liver TIF in turn.

Fig. 5 **b** Quantification of proteins adsorbed to the nanovesicles recovered from plasma. **d** Quantification of proteins adsorbed to the nanovesicles recovered from the liver.

Fig. 7 **b** Quantification of proteins adsorbed to the nanovesicles (PEG-Lips, P-LVs, CP₄-LVs, CP₂-LVs, CP₁-LVs, Lips) with incubation of plasma, PBS, and tumor TIF in turn.

Fig. 9 **b** Quantification of proteins adsorbed to the nanovesicles recovered from A549 and HeLa tumors.

4. I strongly suggest including a brief method description in both the main text and captions of the figures.

Response: According to the reviewer's comments, we have supplemented a brief method description in both the main text and captions of the figures.

Reviewer #2 (Remarks to the Author):

This manuscript is interesting and well-organized. Most conclusions have been proved by the authors. However, there are still some issues should be modified and clarified before the further consideration.

1. The position of the tri-block polymer in liposomes was not clearly. Was it coating on the surface? Was It properly to use PEG-liposomes as the control? The PEG molecules was inserted in liposomes using DSPE part.

Response: The PPO segments of tri-block polymer were mixed with lipids in nanovesicles, and the PEO segments grafted with CSO were coated on the nanovesicle surface. The position of CSO-g-TCP in nanovesicles was verified through time-of-flight secondary ion mass spectrometry (ToF-SIMS) in Fig. 2d-g in the submitted manuscript.

To further investigate the spatial arrangement of functional groups, time-of-flight secondary ion mass spectrometry (ToF-SIMS) was used to scan the signals of the two functional groups. The images clearly illustrated the presence of amino and hydroxyl ions on CP-LVs at the top section (at the beginning of the scan) and the middle section (after a certain period of sputtering), respectively. CP-LVs exhibited a higher intensity of amino ions at the top section than that at the middle section, while hydroxyl ions showed a similar intensity at the top and middle sections. These results indicated that hydroxyl groups were distributed throughout the modified layers and amino groups were exposed to the outside of the PEO layers. Additionally, the signal intensity of amino on CP₁-LVs was

higher than that on CP₂-LVs and CP₄-LVs, similar to the result of Boehm titration. These results demonstrated that CP₁-LVs have the highest amino intensity covering the surface.

Fig. 2 **d** Schematic diagram describing the spatial distribution of amino and hydroxyl groups in CP-LVs determined by ToF-SIMS. **e** Representative images of amino and hydroxyl distribution at the top and middle sections observed by

Tof-SIMS. CP-LVs were coated on silicon substrates and freeze-dried to form a thin film. The signals of amino and hydroxyl groups were detected by Tof-SIMS at different depths in an area of $35 \times 35 \mu\text{m}^2$ with the pixel size of about 120 nm. Scale bar = 2 μm . **f** Real-time signal of amino during a scanning time of 800 s. **g** The decreased rate of relative intensity of amino signal during scanning.

PEGylation have been proved to prolong blood circulation of nanocarriers and widely used in the marketed nanomedicines, such as Doxil. It is necessary and proper to use PEG-liposome as a positive control in the study of long-circulated nanocarrier design. Meanwhile, P-LVs modified with TCP that contains similar -O-CH₂-CH₂- segments to PEG, was used as a control group, comparing with CP-LVs modified with CSO-g-TCP to emphasize the effect of CSO modification on protein adsorption and in vivo behaviors. Additionally, the structure of DSPE-PEG2000 and TCP (PEO-PPO-PEO) is similar, which both possess hydrophilic segment and lipophilic segment, and were inserted in nanovesicles using lipophilic part. To make the PEG-Lips and CP-LVs comparable, the molar of -O-CH₂-CH₂- segments in PEG2000 and CSO-g-TCP was similar in different nanovesicles and liposomes. Hence, PEG-Lips was properly to be used as the control.

2. Absence of PEG, why CP1-LVs with the highest amino density induced the lowest anti-PEG antibody level. It doesn't make sense. To reveal the reasons,

those antibodies such as anti-CP1 antibody should be evaluated.

Response: Anti-PEG antibody induced by PEGylated nanocarriers specifically bind to the repeated units of -O-CH₂-CH₂- which are also the main composition of TCP used in the synthesis of CSO-g-TCP. P-LVs as control groups were mainly composed of TCP, and induced high expression level of anti-PEG antibody, indicated that the repeated units of -O-CH₂-CH₂- in the PEO segments of TCP could induce similar antibodies to PEG induced. CSO-g-TCP used in CP-LVs are synthesized through CSO and TCP. CSO has not been proved to induce specific antibodies. Therefore, anti-PEG as a marker of CP-LVs induced in vivo to evaluate the accelerate blood clearance is reasonable.

3. What is the full name of OPN? Osteopontin?

Response: The full name of OPN is osteopontin. We have noted it in the revised manuscript.

4. Amino and hydroxyl groups are not bioselective, why glycosylated polyhydroxy polymer can offer nanocarriers bioselectivity?

Response: Amino and hydroxyl groups could not affect the interaction between nanovesicles and cells directly, but regulate the compositions of protein coronas on nanovesicles to influence the nanovesicle internalization in different cells. Amino and hydroxyl groups affected nanovesicle-protein interaction through electrostatic effect, which is influenced by the isoelectric point (pI) of proteins.

Proteins with low pI are negatively charged in normal physiological environment (pH 7.4), tending to bind to positively charged groups (such as amino group), while protein with high pI is positively charged and easily repelled by positive groups. CP₁-LVs with an appropriate amino/hydroxyl ratio selectively adsorbed proteins with low pI, including CD44 and OPN, resulting in the adsorption of many tumor TIF proteins, promoting the selective uptake in tumor cells. On contrast, IgG in liver TIF possesses a pI higher than the environment pH, which is positively charged and electrostatically repels the amino groups on CP₁-LVs, resulting low adsorption on CP₁-LVs and little uptake by macrophages in livers.

Reviewer #3 (Remarks to the Author):

Miao et al. have synthesized so-called lipid "nanovesicles" modified with glycosylated polyhydroxy polymers as an alternative to PEG, and determined the impact of subtle differences in surface functional groups with respect to protein adsorption.

The study is comprehensive, but the authors need to be much more specific in terms of describing the experiments (for instance, what do they mean by "plasma"), and they also need to be more stringent in terms of conclusions; the manuscript is replete with unsubstantiated "conclusions" (speculative statements). The authors are advised to take the following into consideration:

Response: According to the reviewer's comments, we have revised the responding methods and results. The detailed contents were shown as follows.

Abstract and Introduction read reasonably well and the authors cover relevant background literature with respect to nanoparticles and the protein corona, and tumor targeting/uptake. However, the Introduction ends with the statement that the present study has shown "striking therapeutic benefits in tumor treatment". This is not an appropriate statement. First, the word "striking" is not appropriate and should be deleted. Second, the authors have not evaluated any anti-tumor treatment in patients, only in conventional subcutaneous xenograft models in nude mice.

Response: We have revised the ending of introduction. The word "striking" has been replaced in the revised manuscript. "Due to their strong ability to prolong blood circulation and promote tumor cell internalization, CP₁-LVs loaded with paclitaxel (PTX) showed superior antitumor efficacy in tumor-bearing mice."

The schematic diagram (Figure 1) is hardly useful, though it may certainly serve as a "graphical abstract". As it stands, this schematic figure contains almost no information on the nanoparticles (except "amino/hydroxyl"). The figure (and the subsequent data reported in the Results section) does, however, raise an unanswered question: if the "nanovesicles" with the highest amino content (designated as CP₁-LVs) display the lowest protein adsorption in plasma (also

lower than the PEGylated control particles), why do the very same particles or vesicles display the highest adsorption of proteins in tumor-derived tissue interstitial fluid (TIF)? What is the mechanism? The authors have not addressed this question anywhere in the manuscript not even in the Discussion, but this is a crucial point that needs to be addressed in order to make sense of the present work. As it stands, it is difficult to understand the mechanism since the authors have not performed unbiased investigations of every plasma-derived, or liver TIF-derived, or tumor TIF-derived protein. Instead, the authors have decided to focus on a few representative proteins, e.g., albumin and IgG (in plasma) and osteopontin and CD44 (in tumor TIF). Therefore, further studies are required to address this conundrum (that is, why does a high amino content repel plasma proteins such as albumin yet attract tumor interstitial fluid-derived proteins?).

Response: We have revised Fig. 1 to clearly present the main content of the manuscript.

Fig. 1 Schematic diagram describing the design of nanovesicles modified with different ratios of amino and hydroxyl groups to explore the dynamic protein corona compositions on nanovesicle surfaces and protein corona-affected in vivo behaviors. CP₁-LVs with an appropriate amino/hydroxyl ratio were expected to display extended circulation by evading protein adsorption to avoid macrophage phagocytosis and enhance internalization in tumor cells through binding tumor-related proteins.

There are often hundreds or even thousands of different types of proteins in plasma and liver TIF. It is difficult to explore the effect of each protein on in vivo behaviors of nanovesicles. The proteomics of plasma protein corona by LC-

MS/MS showed that albumin as one of the most abundant proteins in plasma, was observed to be enriched on the nanovesicle surface and exhibited a reduced adsorption amount on CP-LVs compared with that on PEG-Lips and P-LVs (Fig. 3c, d). Immunoglobulins, including IgG and IgM, are common proteins studied in the field of plasma protein adsorption-affected blood circulation. (Nat Commun. 2018, 9: 2982. Nat Commun. 2020, 11: 3048.) Meanwhile, albumin and IgG were still contained in the protein coronas formed in liver TIF analyzed by LC-MS/MS (Fig. 3g, h). Additionally, liver TIF contains a large amount of liver metabolic enzymes, including dehydrogenases and glutathione transferases, which could cooperate with immunoglobulins with catalase activity to metabolize nanovesicles (Fig. 3g, i). IgG plays an important role in nanovesicle metabolism and clearance in livers, although it has a small proportion of protein corona in liver TIF. Hence, albumin and IgG were chosen as representative proteins in dynamic protein coronas formed in the nanovesicle transportation from plasma to livers.

The protein corona formed in tumor TIF were roughly divided into four categories, including cytoskeletal proteins, cell membrane proteins, extracellular matrix (ECM) proteins and metabolic enzymes determined by LC-MS/MS. The proteins in the ECM and on the tumor cell membranes occupied a high proportion in the protein corona formed both in A549 and Hela tumors (Fig. 7d). The most abundant proteins classified as ECM proteins and cell membrane proteins in the protein corona formed in tumor TIF were further analyzed and

found that the amount of osteopontin (OPN) as an ECM component and transmembrane protein CD44 were higher than other proteins (Fig. 7e). Hence, CD44 and OPN were chosen as representative proteins in protein coronas formed in tumor TIF.

Fig. 3 g Classification of protein corona components formed in the liver TIF identified by quantitative LC-MS/MS. **h** Heatmap of immunoglobulin and albumin in the protein corona of the nanovesicles with liver TIF incubation. **i** Heatmap of the most abundant proteins (top 10) in the adsorbed metabolic enzymes and heat shock proteins on nanovesicles with liver TIF incubation.

Fig. 7 d Classification of protein corona components identified by quantitative LC-MS/MS. **e** Heatmap of the most abundant proteins (top 10) in the protein corona of the nanovesicles with tumor TIF incubation determined by proteomic mass spectrometry.

IgG, CD44 and OPN were used as representative proteins of protein coronas formed in liver and tumor TIF, to investigate the reason of opposite adsorption behaviors of CP₁-LVs in liver and tumor TIF. It is reported that electrostatic effect plays an important role in non-specific protein adsorption, which is affected by the isoelectric point (pI) of proteins. We speculated that the different isoelectric points of IgG, OPN and CD44 resulted in the opposite adsorption behaviors on the CP₁-LVs, and verified it through isoelectric focusing electrophoresis (IEF). During electrophoresis, free IgG migrated to the high pH side, free CD44 and

OPN move in the opposite direction. This was consistent with the previous literatures reported that the pI of IgG is approximately 8.0, while that of CD44 and OPN are both below 5.5. Meanwhile, the fluorescence signal of CP₁-LVs with amino groups was colocalized with more CD44 and OPN signals and less IgG signals, compared to that of P-LVs (Fig. 7h). It may be due to the fact of that proteins with low pI are negatively charged in normal physiological environment (pH 7.4), tending to bind to positively charged groups (such as amino group), while IgG with high pI is positively charged and easily repelled by positive groups. The above results suggested that CP₁-LVs adsorbed a large amount of tumor TIF proteins with low pI and few liver TIF proteins with high pI.

Fig. 7 g Schematic diagram of CP₁-LVs adsorbing IgG, CD44 and OPN affected by protein pI. **h** IEF band images and intensity profile of IgG, CD44, OPN and nanovesicles analyzed by Image J software. Red, green and blue fluorescence signals represent IgG, CD44 and OPN, respectively. Gray fluorescence signal represents nanovesicles.

Results: synthesis and characterization of the nanoparticles is good, but the authors also need to establish whether the particles are endotoxin-free (or not) prior to i.v. injection in mice. Furthermore, some basic toxicity assessment is also required, not using tumor cell lines, but using primary (mouse or human) cells, to determine the dose-response (from low dose to high dose) of each of the particles and whether varying the surface functional groups or amino content will make a difference. The size and zeta potential of the 3 nanovesicles is similar (100 nm, and a zeta potential that is "nearly neutral" (it is slightly negative). What is the zeta potential of these nanovesicles following their immersion in plasma, or liver/tumor TIF?

Response: The safety evaluation of nanovesicles has been supplemented, including hemolysis assay and basic toxicity assessment in normal cells. The results showed that CP-LVs possessed good biosafety.

Supplementary Fig. 4 a The cell viability of HUVECs treated with CP-LVs at different concentrations. **b** In vitro hemolysis assay of CP-LVs. **c** The hemolysis

ratio of CP-LVs. Data are displayed as the mean \pm SD (n = 3).

The zeta potential of nanovesicles incubated with plasma, liver TIF or tumor TIF has been detected and shown as follows. Nanovesicles with protein coronas were electrically neutral. The data has been added in the revised supplementary information.

Supplementary Fig. 12 The zeta potential of nanovesicles incubating with plasma, liver TIF and Hela tumor TIF, compared with that of nanovesicles dispersed in water. Data are presented as mean \pm SD (n=3).

The authors incubated the various particles in "plasma", but they do not specify anywhere in the manuscript (neither in Results nor in the Methods section) whether this means "human plasma" or "mouse plasma" which means that one cannot properly evaluate the results. Moreover, was any anti-coagulant used? if human plasma, then how many human donors were used (the samples are presumably pooled samples of plasma from several donors)? If human donors, then this information (including an ethical board approval) has to be included in the Methods section. Or did the authors use mouse plasma? Minor comment:

no need to say "multitudinous" proteins; this is also ambiguous (a high protein content overall, or many different types of plasma proteins?). In this section, the authors spend a great deal of time discussing some of the findings, including the finding that albumin, fibrinogen and "various immune-associated proteins" are present in the corona. They seem to focus their attention in particular on albumin, an abundant albeit low-affinity protein, and on IgG (and IgM). However, the results show that several apolipoproteins are present in the corona (including apolipoprotein A-I, apolipoprotein A-IV, apolipoprotein C, and clusterin, also known as apolipoprotein J) and these proteins are widely known to bind to nanoparticles; why are these protein corona constituents not mentioned or discussed? In particles, clusterin, which has been shown to bind (specifically) to bind to polymer-modified nanocarriers, leading to a reduced (not enhanced) uptake (Nat Nanotechnol. 2016;11(4):372-7). In fact, previous work has shown that for nanoparticles with low poly(ethylene glycol) (PEG) coverage, adsorption of apolipoproteins can prolong circulation times, whereas the low-density-lipoprotein receptor plays a predominant role in the clearance of nanoparticles, irrespective of PEG density (Nat Commun. 2017;8(1):777). To summarize, the rationale for choosing to focus on albumin and IgG is not clear.

Response: The plasma used in this study was of mouse or rat origin.

The proteomics of plasma protein corona by LC-MS/MS showed that albumin as one of the most abundant proteins in plasma, was observed to be enriched on the nanovesicle surface and exhibited a reduced adsorption amount on CP-

LVs compared with that on PEG-Lips and P-LVs (Fig. 3c, d). Immunoglobulins, including IgG and IgM, are common proteins studied in the field of plasma protein adsorption-affected blood circulation. (Nat Commun. 2018, 9: 2982. Nat Commun. 2020, 11: 3048.) Meanwhile, albumin and IgG were still contained in the protein coronas formed in liver TIF analyzed by LC-MS/MS (Fig. 3g, h). Additionally, liver TIF contains a large amount of liver metabolic enzymes, including dehydrogenases and glutathione transferases, which could cooperate with immunoglobulins with catalase activity to metabolize nanovesicles (Fig. 3g, i). IgG plays an important role in nanovesicle metabolism and clearance in livers, although it has a small proportion of protein corona in liver TIF. Hence, albumin and IgG were chosen as representative proteins in dynamic protein coronas formed in the nanovesicle transportation from plasma to livers.

Fig. 3 g Classification of protein corona components formed in the liver TIF identified by quantitative LC-MS/MS. **h** Heatmap of immunoglobulin and

albumin in the protein corona of the nanovesicles with liver TIF incubation. i
Heatmap of the most abundant proteins (top 10) in the adsorbed metabolic
enzymes and heat shock proteins on nanovesicles with liver TIF incubation.

Furthermore, here (and in several other places) the authors insert "conclusions" while such conclusions are not relevant. For instance, "possibly avoiding the interaction with the immune system and displaying low immunogenicity" (page 10); similar statements are found on almost every page of the Results section ("which may be beneficial to avoiding macrophage phagocytosis", "which may be constructive to avoiding macrophage recognition and phagocytosis", and so on). Such "conclusions" should be avoided; this is speculation and it is also not obvious a priori that a rich protein corona would necessarily lead to uptake as some corona proteins may also act as dysopsonins and not as opsonins.

Response: We have revised the responding description of results to summarize rigorously. "possibly avoiding the interaction with the immune system and displaying low immunogenicity" has been revised as "possibly avoiding the immune response activated by protein coronas". "which may be beneficial to avoiding macrophage phagocytosis" has been revised as "which may reduce the activation of the immune system".

Further down, the authors claim that they injected the nanoparticles intravenously and then retrieved the particles "from plasma and liver". How

where the particles retrieved from the liver? This seems non-trivial and it is not explained in the text. The authors then studied the interactions of the particles with "J774 cells and Kupffer cells". Some clarification is needed (i.e., the murine macrophage-like cell line J774, and rat primary Kupffer cells) (and clarify whether these cells were maintained in serum-free medium, which seems to be the case according to the Methods section, as this is important for the interpretation of the results). The authors have to be more stringent in describing these experiments; the use of isolated Kupffer cells does not mean that "local phagocytosis of local Kupffer cells in the liver" has been investigated (this belongs to the subsequent section of the paper). In other words, authors need to be clear whether they are performing or discussing in vitro results or in vivo results.

Response: The nanovesicles in the liver were recovered as follows. Firstly, liver TIF was collected using the same procedure of liver TIF isolation in the section of "Proteomic identification of proteins adsorbed on nanovesicles in plasma and liver TIF", after intravenous injection of nanovesicles into mice. Secondly, the nanovesicles in the liver TIF were collected using the same procedure of nanovesicle recovery from plasma, including agarose gel column chromatography and ultrafiltration.

The cellular uptake of nanovesicles adsorbing proteins was performed in J774 and Kupffer cells with pre-incubation of serum-free medium for 1 h. The responding content was written in the section of "Cellular uptake of

nanovesicles in macrophages”.

In reference to the in vivo biodistribution studies using ICR mice: the authors, again, draw conclusions regarding the role of individual corona proteins (e.g., role of IgG/IgM) without having any evidence. The fact that different particles are taken up to a different extent by macrophages in the liver does not prove a role for specific proteins; this was not studied and it is merely extrapolation. To prove a role of specific ligands, authors have to knock down the corresponding receptors (Nat Commun. 2017;8(1):777). It is very likely that apolipoproteins also play a role. In the subsequent section, the authors investigate whether the nanoparticles are taken up by A549 lung adenocarcinoma cells or HeLa cervical carcinoma cells. However, they refer to these models as "A549 and HeLa tumors" which is wrong; they are just cell lines. Here, the authors select OPN and CD44 as typical cancer overexpressed proteins, but in order to prove a role for these proteins (in the corona of the nanoparticles), the authors must knock down the corresponding receptors in the cancer cells. the presence of a particular protein in the corona does not necessarily mean that it is a functional epitope for receptor binding, nor does it prove that it is required for "targeting".

Response: This manuscript aims at investigating the common proteins in plasma, liver and tumor dynamically adsorbed on nanovesicles to affect the blood circulation and tumor cellular uptake, rather than exploring the targeting ability caused by nanovesicles adsorbing special proteins bound to tumor

receptor. Albumin and IgG are common immune-related proteins in plasma and liver that were verified through analysis of LC-MS/MS and dot blot in the manuscript, which have been proved to promote the macrophage phagocytosis in many literatures¹. There are hundreds of proteins in protein corona of nanovesicles after incubation with tumor cells, however, OPN and CD44 are representative proteins overexpressed in various tumors, including A549 and Hela tumors. It is universal in the effect of OPN and CD44 on nanovesicle cellular uptake in tumor cells. Through inhibition of protein-mediated internalization pathways, it was proved that the adsorption of OPN and CD44 could enhance nanovesicle internalization in tumor cells (Fig. 8d), which cannot be called as “targeting”. Overall, CP₁-LVs designed in this manuscript prolonged the blood circulation through decreasing the adsorption of common immune-related proteins, such as albumin and IgG, promoting the biodistribution of nanovesicles into tumors. Further, CP₁-LVs adsorbed a large number of proteins overexpressed in various tumors, including OPN and CD44, to enhance the nanovesicle internalization into tumors, improving the delivery efficiency of nanovesicles. The nanovesicles were designed to be used in a variety of tumor treatments.

Fig. 8 d The inhibition rate of nanovesicle cellular uptake in A549 and HeLa cells pretreated with cilengitide (CL) or hyaluronic acid (HA). The data are displayed as the mean \pm SD (n = 3). **p < 0.01, ***p < 0.001 and ****p < 0.0001.

References:

1. Bournazos S, Gupta A, Ravetch JV. The role of IgG Fc receptors in antibody-dependent enhancement. *Nat Rev Immunol* 20, 633-643 (2020).

Finally, authors investigate uptake (targeting) and anti-tumor efficacy in a conventional subcutaneous xenograft model (in nude mice). Again, uptake is not "targeted" unless uptake is reduced in mice carrying tumors with low or silenced receptor expression. Furthermore, the authors should test whether the anti-tumor effects of nanoparticles delivering paclitaxel are mitigated when "targeting" is reduced (by silencing or blocking relevant receptors). This is a key point if the authors want to argue that the tuning of the protein corona is important for efficient drug delivery, as implied by the title of the present manuscript. There is no formal proof to support the suggestion (on page 30) that the presence of OPN or CD44 in the corona have anything to do with tumor cell uptake; this is, again, merely speculation. There is also little proof that the particles are "selectively" internalized by tumor cells. One question that has not been addressed is the uptake of these particles in normal cells; it is possible that particles are taken up and then released by exocytosis while the particles may be retained in cancer cells.

Response: To explore the effect of CD44 and OPN on tumor cell internalization, the expression level of integrin receptor for OPN binding and transmembrane CD44 on A549, HeLa and HUVEC cells was determined, and the results showed that integrin and CD44 were highly expressed on the surface of A549 and HeLa cells, but were low on HUVEC cells (Fig. 8a). We then explored the cellular uptake of nanovesicles with tumor TIF protein coronas in tumor cells. As shown in Fig. 8b, PEG-Lips showed a weak fluorescence signal in A549 and HeLa cells, indicating low cellular uptake in target cells caused by steric hindrance of PEG chains and poor protein adsorption. With the enhanced protein adsorption in tumor TIF, CP₁-LVs showed the highest internalization into A549 and HeLa cells among all nanovesicle groups (Fig. 8c). On the contrary, CP₁-LVs with protein corona formed in tumor TIF exhibited little advantage in uptake in the HUVEC cells with low expression of integrin and CD44, compared with other nanovesicle groups (Fig. 8b). This suggested that OPN and CD44 in protein coronas, as typical proteins related to adhesion and uptake of tumor cells, may contribute to the tumor cellular uptake of nanovesicles. To investigate the role of OPN and CD44 opsonization in cellular uptake, cilengitide (CL) was used to inhibit the interaction between OPN and integrins, and free hyaluronic acid (HA) was pretreated with tumor cells for competitive binding with CD44 to impede the nanovesicle-CD44 interaction. The uptake efficiency in both A549 and HeLa cells was reduced after CL or HA preincubation with tumor cells (Fig. 8d), suggesting the role of OPN and CD44 in the tumor cell uptake of nanovesicles

in the tumor environment. CP₁-LVs showed the sharpest decline in cellular uptake after CL or HA incubation, with an inhibition rate of approximately 25%, while no significant difference was observed in the PEG-Lips group. It should be noted that adsorption of a single protein exhibited a limited effect on nanovesicle internalization; thus, the fluorescence intensity of CP₁-LVs showed less than 30% reduction after preincubation with CL or HA, indicating that the high cellular uptake of CP₁-LVs was caused by the synergism of various proteins, such as CD44, OPN and other proteins overexpressed in tumors.

Fig. 8 a The expression level of integrin and CD44 on A549, HeLa and HUVEC cells determined by Western blotting. **b** CLSM images of A549, HeLa and HUVEC cells incubated with nanovesicles adsorbing tumor protein corona. Scale bar = 20 μ m. **c** Flow cytometric analysis of nanovesicle cellular uptake in A549 and HeLa cells. **d** The inhibition rate of nanovesicle cellular uptake in

A549 and HeLa cells pretreated with cilengitide (CL) or hyaluronic acid (HA).

Overall, a comprehensive, preclinical study, but the reasons for the reduced coronation of particles with high amine content with plasma proteins and the simultaneous enhanced corona formation in tumor-derived fluids remains highly perplexing. The long-term stability of these "nanovesicles" also remains to be investigated.

Response: The data of nanovesicle long-term stability has been supplemented in the supporting information.

Supplementary Fig. 3 The size of CP-LVs in simulated physiological environment over 7 days.

Reviewer #4 (Remarks to the Author):

Summary

The present work describes a new design of nanoparticles, mainly vesicles, which are made of a polymer with different surface modifications to modulate the interaction of the formed vesicles with proteins from the in vivo environment,

such as plasma proteins or from a liver extract. They study whether these new surface properties give an added value to the plain vesicles in order to probe their usefulness in vivo.

Language:

The language of the text includes too much technical jargon, is very difficult to read and the English is complicated, which makes the work very difficult to follow for people who are not experts in the field or in the type of techniques used by these researchers. It is my opinion that a multidisciplinary journal like Nat Com should naturally present a much more accessible language for researchers from other disciplines.

Response: To accurately express the research content of this manuscript, we used specialized words and sentences in the field of protein coronas to write this paper. In order to enhance the readability of this paper, we have revised some description of abstract, introduction and results in the revised manuscript.

General comments:

My general impression is that there are many results collected with little lack of analysis. This means that, in my opinion, the hypotheses established are poorly supported by empirical evidence. I will now cite some of the examples I have observed in the first part of the work I have evaluated.

Abstract:

The summary of the article is complicated to understand due to the complicated use of the language and the excess of jargon and technical language that complicates the reading for those of us who are not experts in the field.

In the abstract (lines 22-23) on one side about the evolution of the corona, it also says that the uptake of proteins is prevented (so, is there a corona?) but on the other side it says that albumin and immunoglobulin absorption is efficiently suppressed, but on the other hand it says that proteins typical of tumors are adsorbed. To me, this seems to be a contradiction since this means there is a protein corona.

On the other hand, says (lines 25-26) "Moreover, CP1-LVs adsorbed abundant tumor distinctive proteins containing CD44 and osteopontin ... mediating specific tumor cell internalization." On the one hand, CD44 is apparently an 85-200 KDa transmembrane glycoprotein that is present in most cells or tissues, so it does not appear to be specific. And, on the other hand, it is not understood why a tumor membrane protein (i.e. CD44) would cause the vesicles to have more affinity for the tumor cells or tissue, since it is not a specific ligand of any tumor receptor, but is part of the repertoire of tumor membrane proteins. so I think this argument is not valid and should not be in the abstract.

Introduction.

The authors should use references on how peptide- or protein-coated nanoparticles are able to interact with receptors in a specific way and not focus exclusively on the coatings they use. Figure 1 is not clear and is not really

necessary, I think it is totally dispensable.

Some of the sentences are rather pretentious in the introduction and I think that as they are part of the results should be left for discussion or conclusions: i.e. "...Due to their strong 84 ability to prolong blood circulation and promote tumor cell internalization, CP1-LVs loaded with 85 paclitaxel (PTX) showed striking therapeutic benefits in tumor treatment...".

Or "... The present study 86 emphasizes the promising application of protein corona regulation to improve delivery efficiency 87 of nanocarriers, and promotes the rational design of functional groups modified nanocarriers."

Response: Lane 22-23 "CP-LVs with the highest amino density (CP₁-LVs) efficiently suppressed albumin and immunoglobulin adsorption in plasma and livers" means that the protein coronas on CP₁-LVs formed in plasma and livers were less than other nanovesicles (such as PEG-Lips, Lips, P-LVs).

During the transportation into liver and tumor from blood, CP₁-LVs exhibited different adsorption behaviors of proteins in liver and tumor TIF, especially the adsorption of IgG, OPN and CD44. It is reported that electrostatic effect plays an important role in non-specific protein adsorption, which is affected by the isoelectric point (pI) of proteins. The pI of IgG, CD44 and OPN is significantly different reported in the literature. The pI of IgG is approximately 8.0, while that of CD44 and OPN is below 5.5. We speculated that the different isoelectric points of IgG, OPN and CD44 resulted in the opposite adsorption behaviors on the CP₁-LVs, and verified it through isoelectric focusing electrophoresis (IEF).

Free proteins (IgG, OPN and CD44), nanovesicles (P-LVs and CP₁-LVs) incubated with proteins were loaded at pH of about 7.4 in the electrophoresis gel with pH range of 5.0-8.0 (Fig. 7g). During electrophoresis, free IgG migrated to the high pH side, free CD44 and OPN move in the opposite direction. The proteins in the nanovesicle-protein complexes showed different protein band patterns compared to free proteins. P-LVs obviously adsorbed IgG, but rarely overlap with CD44 and OPN signals. In contrast, CP₁-LVs with amino modification were colocalized with more CD44 and OPN and less IgG than P-LVs (Fig. 7h). It may be due to the fact of that proteins with low pI are negatively charged in normal physiological environment (pH 7.4), tending to bind to positively charged groups (such as amino group), while IgG with high pI is positively charged and easily repelled by positive groups. Hence, CP₁-LVs with an appropriate amino/hydroxyl ratio selectively adsorbed proteins with low pI, including CD44 and OPN, resulting in the adsorption of many tumor TIF proteins.

Fig. 7 g Schematic diagram of CP₁-LVs adsorbing IgG, CD44 and OPN affected by protein pl. **h** IEF band images and intensity profile of IgG, CD44, OPN and nanovesicles analyzed by Image J software. Red, green and blue fluorescence signals represent IgG, CD44 and OPN, respectively. Gray fluorescence signal represents nanovesicles.

CD44 overexpressed in many tumors is a cell-surface glycoprotein involved in cell adhesion and migration and has been reported to be the major hyaluronan (HA)-receptor to target tumor sites. CP₁-LVs selectively bound to CD44 to enhance the interaction with tumor cell membranes, promoting the nanovesicle internalization in tumor cells.

To clearly present the research content of this paper, we have revised the Fig. 1 in the revised manuscript.

Fig. 1 Schematic diagram describing the design of nanovesicles modified with different ratios of amino and hydroxyl groups to explore the dynamic protein corona compositions on nanovesicle surfaces and protein corona-affected in vivo behaviors. CP₁-LVs with an appropriate amino/hydroxyl ratio were expected to display extended circulation by evading protein adsorption to avoid macrophage phagocytosis and enhance internalization in tumor cells through binding tumor-related proteins.

The introduction has been revised to rigorously describe the expected results of this research. “Due to their strong ability to prolong blood circulation and promote tumor cell internalization, CP₁-LVs loaded with paclitaxel (PTX) were expected to exert superior antitumor efficacy in tumor-bearing mice. The

present study emphasizes the promising application of protein corona regulation to improve delivery efficiency of nanocarriers, and may promote the rational design of functional groups modified nanocarriers.”

Results.

As mentioned above, there are sentences that are excessively difficult to follow, such as those that follow from lines 98 to 104.

"... The amino-modified hydrophilic polymers were synthesized by grafting PEO₂₀-PPO₇₀-PEO₂₀ triblock copolymers to the backbone of CSO through a diimide-activated amidation CSO-g-TCP₄, CSO-g-TCP₂ and CSO-g-TCP₁ (Supplementary Fig. 2), respectively. CSO-g-TCP and DPPC were hydrated together to obtain a series of lipid nanovesicles with surface modification of CSO-g-TCP, structure of a spherical vesicle and an average size of approximately 100 nm, called CP₄-LVs, CP₂-LVs and CP₁-LVs..."

Some sentences are repeated i.e. ... (lane 109) Surface modification of amino and hydroxyl groups determines the interaction of nanovesicles and proteins...

Response: We have revised some description of results. “The amino-modified hydrophilic polymers were synthesized by grafting PEO₂₀-PPO₇₀-PEO₂₀ triblock copolymers to the backbone of CSO through amidation (Fig. 2a and Supplementary Fig. 1). The molar ratios of TCP and CSO in polymers were 4:1, 2:1 and 1:1, which were calculated based on proton nuclear magnetic resonance (¹H-NMR), and named as CSO-g-TCP₄, CSO-g-TCP₂ and CSO-g-

TCP₁ (Supplementary Fig. 2), respectively. CSO-g-TCP and DPPC were hydrated together to obtain a series of lipid nanovesicles, possessing the structure of a spherical vesicle and an average size of approximately 100 nm, called CP₄-LVs, CP₂-LVs and CP₁-LVs (Fig. 2b).”

Figure 2: Characterization of CP-LVs

Fig 2a (lane 576) is this TEM or CryoTEM?;

Fig 2e, it says “...Representative images of amino and hydroxyl distribution at the top and middle sections observed by Tof-SIMS. Scale bar = 2 μm....”

The authors should explain the rationale (fundamentals and calculations) for this technique, what it evaluates, and by what method, as I could not find it anywhere in the manuscript.

Fig2h. The statistics (t values) are not shown (only the n=3 and the SD), I guess that there are significant differences in the zeta potential of the different engineered particles. So if there are no statistically significant differences, I cannot understand how this could influence the adsorption of the three different nanoparticles in the three different nanoparticles. This makes me think that the difference in the composition of the nanoparticles, in the end, is not so important from a surface point of view. Hence, it is questionable whether this may have an effect on the differential adsorption of proteins that ultimately adhere to the nanoparticle surface because of electrostatic or hydrophobic forces.

Lanes 147: To study the composition of protein coronas on CP-LVs and the

influence by the surface modification according to the amino density on the composition in a “physiological environment” authors incubate the particles in a protein solution of 23 mg/mL concentration of plasma used to analyze the protein corona formed on CP-LVs incubated with plasma. In fact, the actual plasma protein concentration is approximately 60–80 mg/mL of which about 50–60% are albumins and 40% globulins (10–20% immunoglobulin) so the environment in which the particles are incubated is not physiological.

On the other hand, they also say that they use a liver extract (lane 608), which does not give its protein concentration and only gives very rough data on how it is prepared. Actually, this extract could have any protein concentration, thus it may not even remotely mimic a physiological liver extract.

Response: Fig 2a is the images of CP-LVs imaged by Cryo-TEM.

We have supplemented the detailed method of Tof-SIMS in the method section of manuscript. “The spatial arrangement of functional groups was analyzed using time-of-flight secondary ion mass spectrometry (IONTOF Tof-SIMS 5-100, Germany). CP-LVs were thinly coated on silicon substrates and freeze-dried. Ion images were acquired with Tof-SIMS using a 30-kV Bi_1^+ beam as the primary ion beam and a 5-kV Ar-Cluster as the sputtering beam. The analysis area of chemical mapping images was $35 \times 35 \mu\text{m}^2$ with the pixel size of about 120 nm for the maximal spatial resolution. The signals of amino and hydroxyl groups were detected by Tof-SIMS at different depths. The density of the amino group was recorded during the whole scan.”

In Fig. 2h, one-way ANOVA was used to analyze the difference of nanovesicle zeta potential, with a P value of 0.945 and no statistical significance. It means that CP-LVs with different amino/hydroxyl ratios showed negligible change in zeta potential. The zeta potential represents the charging properties of the nanoparticle as a whole. CP-LVs with different amino contents showed no difference in zeta potential and were neutrally charged, which was due to the steric hindrance of PEO chains and neutralization of -OH charges. However, the different amino/hydroxyl ratios on CP-LVs could still produce different binding pattern with proteins through electrostatic effect.

To make plasma and Liver TIF protein coronas comparable, the protein concentration of plasma and liver TIF used in the experiments need to be consistent. Therefore, the concentration of plasma and liver TIF was 23 g/L. To further explore the composition of protein coronas in vivo, we recovered nanovesicle-protein complexes from plasma, liver and tumor, and analyzed the protein adsorption behaviors.

Here is a major point of the study. Coomassie-stained SDS:PAGE gels or western blots can only be semi-quantified when there are appropriate loading controls in the same sample, dot blots are not quantifiable and are not comparable among each other where the starting material is not identically quantified at the protein level.

Response: The methods of SDS-PAGE, western blot and dot blot were used

for the protein corona characterization, referring to the published high-impact papers¹⁻³. Conventional method of SDS-PAGE and western blot were used to explore the amount of specific protein in samples with the same amount of protein, while the SDS-PAGE, western blot and dot blot in our study were used to investigate the different protein adsorption of CP-LVs, including protein amounts and types. Therefore, the loading amount of protein onto gel cannot be artificially controlled. PEG-Lips as a control group showed similar results as literature shown. All these methods used in this manuscript are correct and can be used to confirm our hypothesis. The references are as follows:

1. Chen F, et al. Complement proteins bind to nanoparticle protein corona and undergo dynamic exchange in vivo. *Nat Nanotechnol* 12, 387-393 (2017).
2. Li M, et al. Nanoparticle elasticity affects systemic circulation lifetime by modulating adsorption of apolipoprotein A-I in corona formation. *Nat Commun* 13, 4137 (2022).
3. Schöttler S, et al. Protein adsorption is required for stealth effect of poly(ethylene glycol)- and poly(phosphoester)-coated nanocarriers. *Nat Nanotechnol* 11, 372-377 (2016).

Figure 3

The Coomassie-stained gels in Figure 3b, e, f, and the accompanying quantification in Figure 3d are unreliable without internal controls. For example, a standard protein could be added to the mix to ensure that the sample

treatment and loading of each well of the gel have been identical, so that they can be semi-quantified.

I guess the need to do a BCA would be to put identical amounts of protein per well, so it makes no sense that the different lanes of the gel show difference in total protein (and I don't mean differences in types of protein). Since this is what has been measured, there should be no differences in the total amount of protein but in the distribution of the proteins.

But if doing this on an acrylamide gel transfer, where proteins separate neatly by size, is already inaccurate, it is much less imprecise to quantify the result of a western "dot" blot of a protein drop. (lane 620: "... For quantification of adsorbed albumin and IgG, the protein corona was quantified with a dot-blot assay ... with the corresponding HRP-conjugated secondary antibodies against... signal was detected using a Bio-Rad imaging system...")

Therefore, statements such as this (lane 158): "...CP1-LVs showed the lowest protein adsorption among all groups and weaker intensity of most bands compared to CP2-LVs and CP4-LVs...." are not credible on the data provided and should be changed. This can result from a defect in the gel well charge and can be remedied by placing internal load markers (which there are none in this case).

Figure 3b, in results (lane 155) states: "-... Most intriguingly, increasing the density of amino groups on the surface of CP-LVs reduced total protein adsorption. Analysis of the protein corona by SDS-PAGE was consistent with

the result of the amount of protein adsorbed on the nanovesicles (Fig. 3b).....”

I really do not understand how this statement can be made in this manner when no controls are used for the quantification of the starting protein, and not a standard quantification of the initial protein or the adsorbed protein has been done.

Response: Similar to the question above, SDS-PAGE, western blot and dot blot were used to investigate protein corona compositions on nanovesicles. Different nanovesicles with similar mass adsorbed proteins with different amounts and types. Hence, there is a lack of clear internal standard with equal mass in the results. However, the methods used in this research were referred to the published high-impact papers, therefore, the results are valuable to draw the corresponding conclusions.

Figure 4

The nanovesicles were incubated in plasma, PBS, and liver tissue interstitial fluid, and the proteins on the surface are compared using SDS:PAGE (Fig4c). Again, no loading controls are shown, so there could be many factors interfering with this result, from faulty loading or lack of adequate particle stripping.

Figure 4d: “...albumin and IgG bound to nanovesicles was determined by dot blotting assay...” As already mentioned, dot blots are not quantifiable and are not comparable to each other where the starting material is not identically quantified at the protein level.

Response: The SDS-PAGE, western blot and dot blot in our study were used to investigate the different protein adsorption of CP-LVs, including protein amounts and types. Dot blot is a simplified version of western blot, and used for the detection of large samples. Dot blot avoids the errors of the process of electrophoresis and membrane transfer, so that the results are more intuitive and accurate. This method has been widely used in the protein corona characterization¹.

1. Chen F, et al. Complement proteins bind to nanoparticle protein corona and undergo dynamic exchange in vivo. *Nat Nanotechnol* 12, 387-393 (2017).

Figure 4g: “....confocal images of nanovesicles incubated with albumin (upper panel) and IgG solution...” NPs are in the range of 100 nm and this is under the optical resolution (200nm x 500nm). A high-resolution confocal could have been used but only an Olympus FV1000 is mentioned so the images are merely representative. The settings of the microscope should be identical

Response: The images of nanovesicles incubated with proteins were images through Olympus FV1000 and stimulated Emission Depletion (STED) with high-resolution which is mentioned in the manuscript. We have supplemented the responding content to clearly introduce the method. “The nanovesicle-protein complexes were observed by confocal laser scanning microscopy (Olympus FV1000, Japan) at a magnification of 100×. The single nanovesicle adsorbing proteins were zoomed in for further observation through CLSM with stimulated

emission depletion (STED) mode (Leica, Germany).”

(lane 198) Regarding the study where the dynamic exchange of proteins in the corona is done, it seems quite obvious, and according to the hundreds of studies in the literature that when a nanoparticle is coated with proteins, it always exchanges proteins when the particle is exposed to a different medium. When this occurs the particle captures the proteins in the new medium and this process is dynamic. Therefore, it is expected that the finding of the article.

Response: Thank you for your comments on our work.

Up to this point I have been able to evaluate this work. I believe that the absence of controls at the biochemical level in each of the steps performed makes questionable the majority of the results shown at the protein level and, therefore, I believe that these design flaws detected will surely condition all the results and the conclusions that can be drawn from these data.

Reviewers' Comments:

Reviewer #1:

Remarks to the Author:

The authors supplied new data and addressed all questions from this reviewer. The manuscript is now acceptable for publication.

Reviewer #2:

Remarks to the Author:

The authors answered all my questions.

Reviewer #3:

Remarks to the Author:

The authors have addressed several of the comments, and added some discussion on the possible importance of the isoelectric point of proteins. But they also seem to deflect some comments. For instance, why do the authors not comment on (and discuss) the fact that apolipoproteins are also found in the corona (a finding that resonates with several previous publications)? The question also remains: how can one extrapolate from a few selected proteins (osteopontin, CD44) to arrive at "efficient" (if not targeted) drug delivery? Granted, these proteins were selected as examples of proteins that are more abundant in the corona, but there are at least 2 issues that need to be addressed. First, while integrins may act as receptors for OPN, it is not clear what the experiments with HA are supposed to tell us; CD44 is a receptor (not a ligand) and it binds HA, but CD44 also binds OPN. Overall, OPN and CD44 are merely markers of "tumor-specific" (or tumor-enriched) proteins but it is not proven that these proteins drive the uptake of the nanoparticles. This brings us to the second issue namely that the *in vivo* results at the end of the manuscript do not seem to be well connected to the preceding *in vitro* results. That is, the role of specific proteins in the corona is not addressed. How, then, can the "regulation" of the protein corona explain the "efficient drug delivery" (cf. title)?

Reviewer #4:

Remarks to the Author:

The present work describes a new design of vesicles with different surface modifications to modulate the interaction of the formed vesicles with proteins from the *in vivo* environment. The study investigates the qualitative/ quantitative composition of the biocorona and the implication of this comparison to plain vesicles to probe their usefulness *in vivo*.

Similar to my initial review of this work, I am left with the impression that a significant amount of effort has been invested in this study, albeit with limited attention to controls. Since the foundation of this work relies on the distinct composition of the biocorona across various vesicle types and its implications in their interactions with cells and biodistribution, the initial hypothesis must be robustly demonstrated; otherwise, the subsequent work lacks meaningful context. A paramount requirement is for the authors to establish that an equal number of vesicles is loaded with the Acrylamide gels. Without these controls, a valid comparison of associated protein quantities between lanes becomes unfeasible, rendering any conclusive findings elusive.

I will now address some of the issues I previously identified, all of which were highlighted in my initial review and have yet to be attended to. To justify their inaction, the authors have referred me to another study by a different author where similar techniques were employed, purportedly to evade addressing the concerns I raised. Specifically, they assert that the utilization of these methods, as seen in another highly regarded study published in *Nature Nanotechnology*, may not align with the strengths of this journal, particularly in light of the complex biochemistry involved. Figures 3a, 3e, and 3f show the quantification of proteins on the vesicles. As it is known in nanotechnology, the amount of surface on the nanostructures is critical and, in this case, is crucial to understand how much protein adheres to the particles. Assuming they all have the same size and thus, similar surfaces, it is still not mentioned how they have standardized the amount of

vesicles placed in each lane of the gel. This is decisive, as more vesicles imply more protein. As I mentioned in my first review, there is no display of any loading control indicating if exactly the same load of vesicles has been added, which is necessary to directly compare the adsorbed protein. Therefore, this experiment, as well as those in Figure 4, 5, lack loading controls, and without them, one cannot compare one lane to another and quantify the data, which is a very serious biochemical error.

In addition to this, Figures 3c and 3g are histograms without errors (suggesting no experimental replicates), so they are statistically unreliable. In the caption of Figure 3, just as in my previous review, I advised using the BCA method for standard protein quantification. However, it seems the authors have not fully understood the purpose of this technique, as evidenced in the caption (line 203), where they include the phrase "Quantification of plasma proteins adsorbed to the nanovesicle surface determined by BCA assay." I understand that this may be an attempt to address the criticism from the first review, but clearly this cannot be correct, as these measurements only make sense for pre-gel loading protein quantification, while here it seems they are measured in the gel, after electrophoresis.

I also criticized in my previous review the type of protein immunodetection used in the dot blot in Figure 4d. Once again, the authors do not respond to my criticisms and simply mention that there is some article that uses it and that supposedly supports their experiment. Clearly, this technique can be used to determine the total amount of protein when we have a pure solution of a single protein. However, in cases like this where we have a solution of complex proteins and we add an antibody, if we don't know exactly which protein we are detecting and given that it is very common to have unspecific reactions, the signal we are seeing can be from any contaminant protein with cross-reactivity to the antibody. Hence, it is necessary to run an acrylamide gel and perform a Western blot, so if we get a band from a protein in the wrong place, we know it is an unspecific antibody reaction. In other words, although this technique has its value, in this case, it is being used incorrectly to demonstrate what the authors intend, and it is of no use that it has been used in other scientific manuscripts if it is in a different context.

In this same figure (Figure 4), protein quantifications are also presented in the same way as in Figures 3 and 4. As I have mentioned, this quantification makes no sense without an internal control for the exact loading in each gel. On the other hand, qualitative studies may not present any problem. From my point of view, Figure 4c should be removed, and Figure 4b should be repeated, indicating which vesicle loading controls have been used to demonstrate that there is an equal number of vesicles with different amounts of proteins.

I understand that repeating these experiments is very laborious, but in research that emphasizes the quantitative and qualitative importance of proteins on the surface of these nanostructures, it seems essential to clarify these concepts. By not doing so, the quantitative experiments make no sense from the biochemical point of view. I advise the authors to seek advice and a second opinion from a biochemist regarding the critique I am making.

Another criticism I made, which was not taken into account, is the following: In Figure 4g, confocal microscopy techniques are used to validate the presence of proteins on the particle surface by fluorescence. As these nanoparticles are below the optical resolution of light microscopy (200 to 500 nanometres), this technique does not seem to be the most appropriate to confirm the results obtained biochemically. I recommend that the authors use other fluorescence techniques, such as flow cytometry, where they can analyze around 10,000 particles instead of the few dozen shown in the image.

Figure 5 of the paper also includes several figures where proteins on the nanoparticles are quantified in the same way as in Figures 3 and 4. As I have argued previously from a biochemical standpoint, I refer to my previous comments on how protein should be quantified and what controls should be used. Likewise, the dot blot technique would also be inappropriate in this case. Given these form and control inaccuracies in the biochemical aspect, which the authors understandably emphasize, I have reservations about the suitability of this paper for publication in Nature Communications. The subsequent sections of the paper, as they heavily rely on the hypothesis presented in the initial figures, may lack coherence since the initial premise is not sufficiently validated and lacks a solid foundation.

Reviewer #1 (Remarks to the Author):

The authors supplied new data and addressed all questions from this reviewer.

The manuscript is now acceptable for publication.

Response: Thank you for your positive comments on our work.

Reviewer #2 (Remarks to the Author):

The authors answered all my questions.

Response: Thank you very much for your positive comments.

Reviewer #3 (Remarks to the Author):

The authors have addressed several of the comments, and added some discussion on the possible importance of the isoelectric point of proteins. But they also seem to deflect some comments. For instance, why do the authors not comment on (and discuss) the fact that apolipoproteins are also found in the corona (a finding that resonates with several previous publications)? The question also remains: how can one extrapolate from a few selected proteins (osteopontin, CD44) to arrive at "efficient" (if not targeted) drug delivery? Granted, these proteins were selected as examples of proteins that are more abundant in the corona, but there are at least 2 issues that need to be addressed. First, while integrins may act as receptors for OPN, it is not clear what the experiments with HA are supposed to tell us; CD44 is a receptor (not a ligand) and it binds HA, but CD44 also binds OPN. Overall, OPN and CD44 are merely markers of "tumor-specific" (or tumor-enriched) proteins but it is not proven that these proteins drive the uptake of the nanoparticles. This brings us to the second issue namely that the in vivo results at the end of the manuscript do not seem to be well connected to the preceding in vitro results. That is, the

role of specific proteins in the corona is not addressed. How, then, can the "regulation" of the protein corona explain the "efficient drug delivery" (cf. title)?

Response: As apolipoproteins have lipid-binding domains, their attraction to hydrophobic surfaces is readily conceivable. PEG is supposed to be a rather hydrophilic material. Hence, apolipoproteins were more attached to the surface of Lips than that of PEGylated liposomes and TCP-modified nanovesicles (Fig. 3c). Apolipoprotein A-I (ApoA1) was the predominant abundant protein among the corona lipoproteins in our study, and has been reported as the major apolipoprotein whose relative abundance in corona strongly correlated with nanoparticles' blood clearance lifetimes¹. We used ApoA1 as the representative apolipoprotein to explore the protein adsorption behavior and subsequent impact on macrophage phagocytosis. Lips adsorbed a large amount of ApoA1, while PEGylated liposomes and TCP-modified nanovesicles showed decreased ApoA1 adsorption. Meanwhile, there was little difference in ApoA1 adsorption on PEG-Lips, P-LVs and CP-LVs, indicating that the amino/hydroxyl ratio had little effect on ApoA1 adsorption (Supplementary Fig. 7a, b). Then, we investigated the effect of ApoA1 adsorption on uptake of nanovesicles in J774 cells. Lips adsorbing ApoA1 decreased the cellular uptake in J774 cells compared to Lips without ApoA1 adsorption, due to a large amount of ApoA1 adsorbed on the nanovesicle surface suppressing cellular uptake in macrophages. However, PEG-Lips, P-LVs and CP-LVs showed no obvious difference in cellular uptake within J774 cells in the absence and presence of ApoA1 adsorption, due to the negligible adsorption of ApoA1 (Supplementary Fig. 7c, d). The data has been supplemented in revised supplementary information.

Supplementary Fig. 7 a Quantification of ApoA1 adsorbed to the nanovesicle surface determined by BCA assay. **b** Qualitative molecular composition of the adsorbed protein layer on the nanovesicles by SDS-PAGE. **c** Representative flow cytometric histograms of nanovesicles with or without ApoA1 adsorption within J774 cells. **d** Fluorescence intensity of nanovesicles with different protein coronas within J774 cells. The data are displayed as the mean \pm SD ($n = 3$). n.s. $p > 0.05$.

Reference:

1. Li M, *et al.* Nanoparticle elasticity affects systemic circulation lifetime by modulating adsorption of apolipoprotein A-I in corona formation. *Nature Communications* **13**, 4137 (2022).

In the field of nano drug delivery system, the nanocarriers modified with ligands of small or biological macromolecules could bind to receptor proteins hyper expressed on the target cells and enhance the cell internalization^{1, 2}. For example, nanocarriers modified with hyaluronic acid could bind to CD44 overexpressed on the tumor cell membranes and promote the nanocarrier cellular uptake in tumor cells. Arginyl-glycyl-aspartate (RGD) peptides

conjugated- nanocarriers were internalized into tumor cells through RGD-integrin pathway. Folate modified-nanocarriers target the folate receptor on tumor cell membranes. After interacting with receptor, the drug nanocarriers are internalized into cells through clathrin, caveolin or macropinocytosis mediated cell membrane endocytosis. In our study, nanovesicles adsorbed a large number of tumor-overexpressed proteins in the tumor TIF, including OPN and CD44. Proteins adsorbed on the surface of nanovesicles could play a similar role as targeting ligands to bind receptors on tumor cell membranes, including OPN-integrin and CD44-mediated pathway, promoting the interaction between nanovesicles and cell membranes, followed by endocytosis and drug release. Hence, the protein coronas formed in the tumor TIF could enhance the delivery efficiency of nanovesicles. We have supplemented the related discussion and references in the revised manuscript.

References:

1. Mitchell MJ, Billingsley MM, Haley RM, Wechsler ME, Peppas NA, Langer R. Engineering precision nanoparticles for drug delivery. *Nature Reviews Drug Discovery* **20**, 101-124 (2021).
2. Dilliard SA, Siegwart DJ. Passive, active and endogenous organ-targeted lipid and polymer nanoparticles for delivery of genetic drugs. *Nature Reviews Materials* **8**, 282-300 (2023).

Reviewer #4 (Remarks to the Author):

The present work describes a new design of vesicles with different surface modifications to modulate the interaction of the formed vesicles with proteins from the in vivo environment. The study investigates the qualitative/ quantitative composition of the biocorona and the implication of this comparison to plain vesicles to probe their usefulness in vivo.

Similar to my initial review of this work, I am left with the impression that a significant amount of effort has been invested in this study, albeit with limited

attention to controls. Since the foundation of this work relies on the distinct composition of the biocorona across various vesicle types and its implications in their interactions with cells and biodistribution, the initial hypothesis must be robustly demonstrated; otherwise, the subsequent work lacks meaningful context. A paramount requirement is for the authors to establish that an equal number of vesicles is loaded with the Acrylamide gels. Without these controls, a valid comparison of associated protein quantities between lanes becomes unfeasible, rendering any conclusive findings elusive.

I will now address some of the issues I previously identified, all of which were highlighted in my initial review and have yet to be attended to. To justify their inaction, the authors have referred me to another study by a different author where similar techniques were employed, purportedly to evade addressing the concerns I raised. Specifically, they assert that the utilization of these methods, as seen in another highly regarded study published in Nature Nanotechnology, may not align with the strengths of this journal, particularly in light of the complex biochemistry involved.

Figures 3a, 3e, and 3f show the quantification of proteins on the vesicles. As it is known in nanotechnology, the amount of surface on the nanostructures is critical and, in this case, is crucial to understand how much protein adheres to the particles. Assuming they all have the same size and thus, similar surfaces, it is still not mentioned how they have standardized the amount of vesicles placed in each lane of the gel. This is decisive, as more vesicles imply more protein. As I mentioned in my first review, there is no display of any loading control indicating if exactly the same load of vesicles has been added, which is necessary to directly compare the adsorbed protein. Therefore, this experiment, as well as those in Figure 4, 5, lack loading controls, and without them, one cannot compare one lane to another and quantify the data, which is a very serious biochemical error.

Response: In Fig. 3-5 , the characterization of protein corona were analyzed by BCA assay and SDS-PAGE based on the similar nanovesicle amount. We

incubated the proteins with DiO-labeled nanovesicles, and measured nanovesicle fluorescence intensity to prove the similar nanovesicle amount in the process of nanovesicle-protein complex isolation and characterization. Take the plasma protein corona as an example, the detail method of protein corona characterization was as follows:

1. DiO-labeled nanovesicles were prepared and measured by a microplate reader to ensure the consistent fluorescence intensity of nanovesicles at the same concentration, to choose the fluorescence intensity as the detection index of nanoparticle concentration.
2. DiO-labeled nanovesicles were incubated with plasma. The nanovesicle fluorescence intensity and protein concentration in nanovesicle-protein mixtures were determined through a microplate reader and BCA assays, to ensure the same concentration of nanovesicles and proteins during the process of incubation.
3. The nanovesicle-protein complexes were isolated through ultracentrifugation and resuspended in RIPA lysates. The nanovesicle-protein complexes were diluted with PBS and their fluorescence intensity were measured by a microplate reader, to prove the same concentration of nanovesicles in the complexes. Then, the protein contents of nanovesicle-protein complexes were determined by BCA assays.
4. For SDS-PAGE, nanovesicle-protein samples with similar amounts of nanovesicles were loaded into the SDS-PAGE gels, and imaged by the biomolecular imager to prove the similar amounts of nanovesicles. After electrophoresis, the protein bands were stained with Coomassie blue and imaged by a gel documentation system.

The nanovesicle amount as the control variable was standardized through fluorescence intensity at each step. The protein corona characterization was evaluated on the basis of the consistent nanovesicle amounts, and the results in the manuscript could accurately reflect the protein adsorption behaviors and the difference in protein adsorption between groups.

1. DiO-labeled nanovesicle preparation

The fluorescence intensity of nanovesicles was determined through a microplate reader.

	PEG-Lips	P-LVs	CP ₄ -LVs	CP ₂ -LVs	CP ₁ -LVs	Lips
Fluorescence intensity	540425	534566	533702	545400	556889	586540

2. DiO-labeled nanovesicles incubated with plasma

The protein content and fluorescence intensity of nanovesicle-protein solution were determined.

	PEG-Lips	P-LVs	CP ₄ -LVs	CP ₂ -LVs	CP ₁ -LVs	Lips
Fluorescence intensity	295263	312102	300975	316425	301861	323678
Protein content (µg/mL)	11141.7	11079.2	11100.0	11558.3	11933.3	11113.9

3. Isolation of nanovesicle-protein complexes through ultracentrifugation

The protein content and fluorescence intensity of isolated nanovesicle-protein complexes were determined.

	PEG-Lips	P-LVs	CP ₄ -LVs	CP ₂ -LVs	CP ₁ -LVs	Lips
Fluorescence intensity	141394	141297	145287	146099	135437	151742
Protein content (µg)	44.6	49.3	42.0	31.7	8.5	151.3

4. Protein characterization by SDS-PAGE

The protein samples with similar fluorescence intensity of nanovesicles were loaded into the SDS-PAGE gels and imaged by the biomolecular imager.

After electrophoresis, the protein bands were stained with Coomassie blue and imaged by a gel documentation system.

I also criticized in my previous review the type of protein immunodetection used in the dot blot in Figure 4d. Once again, the authors do not respond to my criticisms and simply mention that there is some article that uses it and that supposedly supports their experiment. Clearly, this technique can be used to determine the total amount of protein when we have a pure solution of a single protein. However, in cases like this where we have a solution of complex proteins and we add an antibody, if we don't know exactly which protein we are detecting and given that it is very common to have unspecific reactions, the signal we are seeing can be from any contaminant protein with cross-reactivity to the antibody. Hence, it is necessary to run an acrylamide gel and perform a Western blot, so if we get a band from a protein in the wrong place, we know it is an unspecific antibody reaction. In other words, although this technique has its value, in this case, it is being used incorrectly to demonstrate what the

authors intend, and it is of no use that it has been used in other scientific manuscripts if it is in a different context.

Response: In Fig. 4d, the adsorption of albumin and IgG in different protein solutions were compared. The expression level of specific proteins analyzed by Western blotting was recognized by antibodies after the important step of protein transfer from the SDS-PAGE gels to the PVDF membranes. This step is easy to be interfered by multiple factors, including SDS-PAGE gels, transfer buffer, transfer time, and temperature, etc., resulting in different transfer efficiency of proteins. Western blotting is not suitable to analyze the difference in a large number of samples which cannot be loaded on the same piece of SDS-PAGE gel and transferred on the same piece of PVDF membrane.

To avoid the errors of the process of membrane transfer, Dot blotting was used for the detection of large samples, and are more intuitive and accurate. To avoid the cross-reactivity between any contaminant protein with antibody, we used western blotting to verify the specific reactions between antibody and detected protein. There is only a single band detected by anti-albumin or anti-IgG in the molecular weight range of 20-100KD, indicating that the antibodies used in Dot blotting have strong specificity, and the results of Dot blotting could well reflect the process and trend of adsorption and desorption, and exhibit the difference in protein adsorption between groups.

Supplementary Fig. 8 Western blotting of albumin and IgG adsorbed on nanovesicles.

In order to prove the viewpoint in this manuscript more precisely, the adsorption

level of albumin, IgG, OPN and CD44 in Fig. 5f, Fig. 7f in Fig. 5f, Fig. 7f and Fig. 9d was redetected by the Western blotting assay to accurately compare the level of proteins adsorbed on the different nanovesicles. The results of Western blotting and Dot blotting were consistent. The responding data of Western blotting has been updated in Fig. 5, 7 and 9 of the manuscript.

Fig. 5 f Western blotting assay of albumin and **g** IgG adsorbed on nanovesicles.

Fig. 7 f OPN and CD44 bound to nanovesicles were determined by Western blotting assay.

Fig. 9 d Western blotting assay of CD44 and OPN adsorbed on nanovesicles recovered from A549 and HeLa tumors.

In this same figure (Figure 4), protein quantifications are also presented in the same way as in Figures 3 and 4. As I have mentioned, this quantification makes no sense without an internal control for the exact loading in each gel. On the other hand, qualitative studies may not present any problem. From my point of view, Figure 4c should be removed, and Figure 4b should be repeated,

indicating which vesicle loading controls have been used to demonstrate that there is an equal number of vesicles with different amounts of proteins.

Response: During the study of protein corona characterization, we controlled the concentration of nanoparticles at each step. For SDS-PAGE, nanovesicle-protein samples with similar amounts of nanovesicles were loaded into the SDS-PAGE gels, and imaged by the biomolecular imager to prove the similar amounts of nanovesicles. After electrophoresis, the protein bands were stained with Coomassie blue and imaged by a gel documentation system.

I understand that repeating these experiments is very laborious, but in research that emphasizes the quantitative and qualitative importance of proteins on the surface of these nanostructures, it seems essential to clarify these concepts. By not doing so, the quantitative experiments make no sense from the biochemical point of view. I advise the authors to seek advice and a second opinion from a biochemist regarding the critique I am making.

Another criticism I made, which was not taken into account, is the following: In Figure 4g, confocal microscopy techniques are used to validate the presence of proteins on the particle surface by fluorescence. As these nanoparticles are below the optical resolution of light microscopy (200 to 500 nanometres), this technique does not seem to be the most appropriate to confirm the results obtained biochemically. I recommend that the authors use other fluorescence techniques, such as flow cytometry, where they can analyze around 10,000 particles instead of the few dozen shown in the image.

Response: The images in Fig. 4g matched the binding affinity constant of nanovesicle-protein in Fig. 4f. The interaction between nanovesicles and proteins (albumin and IgG) was explored by isothermal titration calorimetry (ITC) to explain the dynamic adsorption of proteins on nanovesicles. The calculated binding affinity of albumin to the nanovesicles was significantly less than that of IgG, which could be used to explain the dynamic adsorption of albumin and IgG

in plasma and liver TIF in Fig. 4 a-e. To visualize the protein-nanovesicle interactions, fluorescence-labeled proteins were incubated with nanovesicles and observed by confocal laser scanning microscopy (CLSM) with stimulated emission depletion (STED) mode. These images represented the visual information of nanovesicle-protein interaction. On the basis of the affinity constant quantified by ITC, we used the microscope images to visually display the dynamic adsorption of albumin and IgG on nanovesicle.

Figure 5 of the paper also includes several figures where proteins on the nanoparticles are quantified in the same way as in Figures 3 and 4. As I have argued previously from a biochemical standpoint, I refer to my previous comments on how protein should be quantified and what controls should be used. Likewise, the dot blot technique would also be inappropriate in this case. Given these form and control inaccuracies in the biochemical aspect, which the authors understandably emphasize, I have reservations about the suitability of this paper for publication in Nature Communications. The subsequent sections of the paper, as they heavily rely on the hypothesis presented in the initial figures, may lack coherence since the initial premise is not sufficiently validated and lacks a solid foundation.

Response: The protein corona characterization analyzed by BCA assays and SDS-PAGEs in Fig. 5 controlled the amounts of nanovesicles by their fluorescence intensity. For SDS-PAGE, nanovesicle-protein samples with similar amounts of nanovesicles were loaded into the SDS-PAGE gels, and imaged by the biomolecular imager to prove the similar amounts of nanovesicles. The nanovesicle amount as the only control variable was standardized through fluorescence intensity at each step, to ensure the accurate results of protein corona characterization.

The adsorption level of albumin and IgG was explored by Western blotting, and the responding data has been supplemented in Fig. 5 f and g.

Reviewers' Comments:

Reviewer #3:

Remarks to the Author:

In general, while the present study may serve as a nice example of the importance of surface modification-mediated protein corona regulation on nanocarriers for drug delivery, the authors seem to place too much faith in certain proteins; hence, distinctive proteins namely CD44 and osteopontin are suggested to mediate selective tumor cell internalization (see abstract). I think that these proteins may be seen as examples of relevant protein corona constituents but to prove their role in tumor cell targeting, the corresponding receptors would have to be ablated (on tumor cells). Therefore, it may be prudent to say "such as" CD44 and osteopontin as the role(s) of other corona proteins has not yet been ruled out.

Reviewer #4:

Remarks to the Author:

Having thoroughly examined the authors' responses to my critiques, I find that they have devised new experiments and technologies to address the issues raised. Consequently, I believe the article, from my perspective, is now suitable for publication.

REVIEWERS' COMMENTS

Reviewer #3 (Remarks to the Author):

In general, while the present study may serve as an nice example of the importance of surface modification-mediated protein corona regulation on nanocarriers for drug delivery, the authors seem to place too much faith in certain proteins; hence, distinctive proteins namely CD44 and osteopontin are suggested to mediate selective tumor cell internalization (see abstract). I think that these proteins may be seen as examples of relevant protein corona constituents but to prove their role in tumor cell targeting, the corresponding receptors would have to be ablated (on tumor cells). Therefore, it may be prudent to say "such as" CD44 and osteopontin as the role(s) of other corona proteins has not yet been ruled out.

Response: We have revised some descriptions in the section of "Abstract" and added some contents in the section of "Discussion", as follows.

Abstract: "Moreover, CP1-LVs adsorbed abundant tumor distinctive proteins, such as CD44 and osteopontin in tumor interstitial fluids, mediating selective tumor cell internalization."

Discussion: "CP₁-LVs selectively adsorbed a large amount of tumor microenvironment-related proteins and showed the enhanced internalization in tumor cells and tumor accumulation, which was attributed to synergism of various proteins, such as CD44, OPN and other proteins overexpressed in tumors."

Reviewer #4 (Remarks to the Author):

Having thoroughly examined the authors' responses to my critiques, I find that they have devised new experiments and technologies to address the issues raised. Consequently, I believe the article, from my perspective, is now suitable

for publication.

Response: Thank you for your positive comments on our work.

EDITOR'S COMMENTS

FEATURED IMAGE

If you wish, you can provide an interesting image (but not an illustration or schematic) for consideration as a Featured Image on the Nature Communications homepage. The file should be 1200x675 pixels in RGB format and should be uploaded as a Related Manuscript File with the title “featured image suggestion”. In addition to our home page, we may also use this image (with credit) in other journal-specific promotional material. If your featured image is chosen you will need to complete a Licence to Publish form which will be sent to you via DocuSign at a later stage.

Response: We have provided a featured image as follows.

AUTHOR CHANGES DURING PREVIOUS REVISIONS

If there have been any changes to the author list since your initial submission, please use this approval form www.nature.com/documents/nr-author-list-change-form.pdf, arranging for all authors on your paper to sign the statement confirming that they agree to the author list being changed, and add this document to your resubmission. Please also add an explanation of the changes to your cover letter.

Response: During the revision process, the author list changed due to an error. We have checked and corrected the author list. We confirm that the final author list is consistent with the author list since the initial submission.